

# Benthic alkalinity and DIC fluxes in the Rhône River prodelta

# generated by decoupled aerobic and anaerobic processes

Jens Rassmann[a,1] , Eryn M. Eitel[b, 1] , Cécile Cathalot[c], Christophe Brandily[c],
Bruno Lansard[a] , Martial Taillefert[b] , Christophe Rabouille[a, 2]

[a] *Laboratoire des Sciences du Climat et de l'Environnement, LSCE/IPSL,CEA-CNRS-UVSQ-Université Paris Saclay, 91198 Gif-sur-Yvette, France*
[b] *School of Earth and Atmospheric Sciences; Georgia Institute of Technology, GA 30332-0340 Atlanta, USA*
[c] *IFREMER, Laboratoire Environnement Profond, 29280 Plouzané, France*

[1]Jens Rassmann and Eryn M. Eitel contributed equally to this article
[2] Corresponding author
    Email adress: rabouill@lsce.ipsl.fr (Christophe Rabouille)
    ORCID: https://orcid.org/0000-0003-1211-717X



## Abstract

Estuarine regions are generally considered a net source of atmospheric $CO_2$ as a result of the high
organic carbon (OC) mineralization rates in the water column and their sediments. Yet, the intensity
of anaerobic respiration processes in the sediments tempered by the reoxidation of reduced
metabolites controls the net production of alkalinity from sediments that may partially buffer the
metabolic $CO_2$ generated by OC respiration. In this study, a benthic chamber was deployed in the
Rhône River prodelta and the adjacent continental shelf (Gulf of Lions, NW Mediterranean) to assess
the fluxes of total alkalinity (TA) and dissolved inorganic carbon (DIC) from the sediment.
Concurrently, *in situ* $O_2$ and pH microprofiles, electrochemical profiles, pore water and solid
composition were measured in surface sediments to identify the main biogeochemical processes
controlling the net production of alkalinity in these sediments. The benthic fluxes of TA and DIC,
ranging between 14 and 74 mmol $m^{-2}$ $d^{-1}$ and 18 and 78 mmol $m^{-2}$ $d^{-1}$, respectively, were up to 8 times
higher than the DOU fluxes ($10.4 \pm 0.9$ mmol $m^{-2}$ $d^{-1}$) close to the river mouth, but their intensity
decreased offshore, as a result of the decline in OC inputs. Low nitrate concentrations and strong pore
water sulfate gradients indicated that the majority of the TA and DIC was produced by sulfate and
iron reduction. Despite the complete removal of sulfate from the pore waters, dissolved sulfide
concentrations were low due to the precipitation and burial of iron sulfide minerals (12.5 mmol $m^{-2}$
$d^{-1}$ near the river mouth), while soluble organic-Fe(III) complexes were concurrently found
throughout the sediment column. The presence of organic-Fe(III) complexes together with low sulfide
concentrations and high sulfate consumption suggests a dynamic system driven by the variability of
the organic and inorganic particulate input originating from the river. By preventing reduced
substances from being reoxidized, the precipitation and burial of iron sulfide decouples the iron and
sulfur cycles from oxygen, therefore allowing a flux of alkalinity out of the sediments. In these
conditions, the sediment provides a source of alkalinity to the bottom waters which mitigates the
effect of the benthic DIC flux on the carbonate chemistry of coastal waters.



**Keywords**

Carbon cycle; alkalinity flux; iron reduction; sulfate reduction; coupled element cycles
**1. Introduction**
As a link between continental and marine environments, the coastal ocean plays a key role in the
global carbon cycle (Bauer et al., 2013). In particular, large fluxes of dissolved and particulate organic
carbon (POC) are delivered by rivers to neighbouring continental shelves (Bianchi and Allison, 2009).
In fact, even though shelf regions only occupy around 7 % of the global ocean surface area (Jahnke,
2010), they account for more than 40 % of POC burial in the oceans of which about half is buried in
river deltas and estuaries (Hedges and Keil, 1995; McKee et al., 2004; Muller-Karger et al., 2005;
Chen and Borges, 2009). River-dominated ocean margins receive substantial amounts of
allochthonous and authigenic POC that settle to the sea floor (Rabouille et al., 2001; Burdige, 2005;
Andersson et al., 2006), therefore increasing the organic carbon content of the sediments and
enhancing mineralization rates (Canfield et al., 1993a; Mckee et al., 2004; Muller-Karger et al., 2005;
Aller et al., 2008; Burdige, 2011). These processes allow estuarine and deltaic regions to constitute a
net source of $CO_2$ to the atmosphere (Chen and Borges, 2009, Cai, 2011). In these river-dominated
margins, high sedimentation rates of material containing large concentrations of POC decrease the
residence time of organic carbon in the oxic sediment layers (Hartnett et al., 1998) and increase the
relative contribution of anaerobic compared to aerobic degradation pathways of organic carbon
(Canfield et al., 1993a). Anaerobic respiration processes, including denitrification, dissimilatory
nitrate reduction to ammonium (DNRA), manganese reduction, iron reduction, and sulfate reduction
create total alkalinity (TA) (Berner, 1970; Dickson, 1981; Wolf-Gladrow et al., 2007, Table 1) that
increases the buffer capacity of pore waters (Ben-Yaakov, 1973; Soetaert et al., 2007), drives the
carbonate saturation state of the pore waters towards oversaturation, and potentially trigger carbonate
precipitation (Gaillard et al., 1989; Mucci et al., 2000; Jørgensen and Kasten, 2006; Soetaert et al.,
2007; Burdige, 2011).  In turn, the precipitation of carbonate species, such as calcite and aragonite,



consumes alkalinity within the sediments (Table 1, Eq. 1; Berner, 1970; Soetaert et al., 2007; Krumins
et al., 2013; Brenner et al., 2016). Anaerobically produced alkalinity may also be consumed close to
the sediment-water interface (SWI) by the aerobic reoxidation of reduced species such as $NH_4^+$, $Mn^{2+}$,
$Fe^{2+}$, and dissolved sulfide (Table 1, Eq. 2-4; Jourabchi et al., 2005; Krumins et al., 2013; Brenner et
al., 2016). However, the precipitation and ultimate burial of iron sulfide minerals may prevent
reoxidation of dissolved sulfide and $Fe^{2+}$ and result in the net production of alkalinity in sediments.
Thus, the net TA flux across the SWI depends on the type and intensity of anerobic respiration, on
carbonate precipitation/dissolution and whether reduced species are reoxidized by dissolved oxygen
after diffusion upwards or trapped in anaerobic sediment layers by precipitation (Hu and Cai, 2011a;
Krumins et al., 2013; Łukawska-Matuszewska et al., 2018).
To characterize the biogeochemical conditions in which sediments provide an alkalinity source
to coastal waters, it is crucial to relate this reaction network to net benthic fluxes of alkalinity and
dissolved inorganic carbon (DIC) measured *in situ*. A high ratio (> 1) of benthic TA to DIC fluxes
would increase the buffer capacity of the bottom waters. This could influence the coastal carbon cycle
by increasing the storage capacity of $CO_2$ in coastal waters over long time scales (Thomas et al., 2009;
Andersson et al., 2012; Brenner et al., 2016). The objectives of this study were to investigate if
sediments from deltaic regions exposed to large riverine inputs of carbon and minerals represent an
alkalinity source to the bottom waters and identify the biogeochemical processes responsible for the
net production of alkalinity in these sediments. To achieve these objectives, TA and DIC benthic
fluxes, dissolved oxygen uptake (DOU) fluxes, burial fluxes of reduced substances and the main
biogeochemical processes involved in organic carbon mineralization in sediments were determined
along a gradient of organic carbon and mineral inputs to the sea floor in the Rhône River delta
(France).



## 2. Study site and methods

### 2.1 The Rhône River delta

The Rhône River subaqueous delta, also called prodelta due to its prograding characteristics, is a wave-dominated delta located in the Gulf of Lions (France), a microtidal continental margin. The Rhône River is the main source of freshwater, sediments (including iron oxides), and POC to the Mediterranean Sea (Sempéré et al., 2000). The river plume is generally oriented southwestward due to the combined effects of wind forcing and the Coriolis effect (Estournel et al., 1997). The Grand Rhône River mouth is characterized by a prodeltaic lobe (Got et al., 1990) that can be divided into three main areas based on bathymetry and sedimentation rates: the proximal domain within a 2 km radius of the river outlet and with water depths between 10 and 30 m; the prodelta domain between 2 and 5 km with water depths ranging from 30 to 70 m; and the distal domain further offshore with water depths greater than 70 m. The sediments of the three domains are characterized by a strong biogeochemical gradient from the Rhône River mouth to the Gulf of Lions continental slope (Lansard et al., 2009).

Most of the riverine particles settle in the vicinity of the river mouth leading to mean apparent accumulation rates of up to 37-48 cm yr$^{-1}$ (Charmasson et al., 1988), including about 80 % of the particles deposited during flood events (Maillet et al., 2006; Cathalot et al., 2010). Thus, sediments from the proximal domain are dominated by the periodic accumulation of terrestrial organic-rich particles (Radakovich et al., 1999; Roussiez et al., 2005). Offshore, sedimentation rates decrease rapidly and reach typical values for shelf regions (< 0.1 cm yr$^{-1}$), in the distal domain (Miralles et al., 2005). The sediments in all the study area are fine grained and of cohesive nature (Roussiez et al., 2005). Their total organic carbon content is higher than 2 % close to the river mouth and decreases offshore (Lansard et al., 2008). The sedimentary inorganic carbon content ranges between 28 and 38 % (Roussiez et al., 2005) and is mostly composed of calcite and magnesian calcite (Rassmann et al., 2016). Sediment respiration rates are high in the proximal domain and decrease offshore (Lansard

et al., 2009; Pastor et al., 2011; Cathalot et al., 2013; Rassmann et al., 2016). These sediments are
characterized by strong anaerobic production of TA and DIC (Rassmann et al., 2016), but whether
this alkalinity is consumed in the oxic sediment layer or released to the bottom waters has yet to be
determined.
*2.2 Bottom water sampling and analyses*

The AMOR-B-Flux cruise took place on-board the RV Tethys II (CNRS-INSU) in September

2015. The investigated stations were located in the river plume along a nearshore-offshore transect
(Fig. 1 and Table 2). Bottom water samples were collected with 12-L Niskin® bottles as close as
possible to the sea floor. The seawater temperature was measured using a thermometer with a
precision of 0.1 °C and the salinity with a conductivity based thermosalinometer with a precision of
0.1. Triplicate pH measurements, reported on the total proton scale ($pH_T$), were carried out within 1
hour after sampling by spectrophotometry with unpurified m-cresol purple as indicator dye (Clayton
and Byrne, 1993) and a precision of ± 0.01 pH units. Dissolved oxygen concentrations were analysed
by Winkler titration (Grasshoff et al., 1983) within twelve hours after sampling with a precision of ±
0.5 µM.
*2.3 In situ benthic chamber deployments*

Benthic fluxes and sediment depth profiles of the main redox species involved in the

remineralization of organic carbon were determined with an autonomous benthic lander (Jahnke et
and Christiansen, 1989). The lander was equipped with a single benthic chamber, water syringe
sampling system, and retrofitted with a programmable, battery-powered ISEA IV In Situ
Electrochemical Analyzer and a SUBMAN-1 in situ micromanipulator from Analytical Instrument
Systems, Inc. (AIS, Inc.) to simultaneously obtain depth profiles of redox chemical species with
mercury/gold (Hg/Au) amalgam voltammetric microelectrodes (Luther et al., 2008; Tercier-Waeber
and Taillefert, 2008). The chamber encloses a 30 x 30 cm sediment surface area with a certain volume

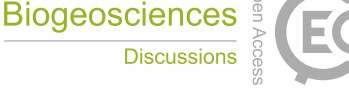



of overlying water determined by measuring the concentration of two tracers (iodide and bromide)
injected immediately after closure of the chamber. Homogenization of the overlying waters was
assured with a stirrer integrated in the chamber lid. TA and DIC concentrations were determined in
the benthic chamber water samples collected as a function of time. The slopes of the concentration-
time plots were estimated using a restricted maximum likelihood estimator (REML) that takes
uncertainties of individual measurements into account. Finally, benthic fluxes across the SWI ($F_i$)
were calculated from the slopes of these concentration-time-plots and the chamber height (Eq. 1),

$$F_i = H \cdot \frac{dC_i}{dt} \tag{1}$$

where H is the overlying water height in the benthic chamber, $C_i$ represents the concentration of the
analyte i (TA or DIC), and t is time.
*2.4 In situ microprofiling of dissolved oxygen and pH*
A separate benthic lander, carrying a benthic microprofiler (Unisense®), was deployed to measure
*in situ* microprofiles of dissolved oxygen and pH (Cai and Reimers, 1993; Rabouille et al., 2003,
Rassmann et al., 2016 and references therein). Up to five oxygen and two pH microelectrodes were
simultaneously deployed, and vertical depth profiles were measured with a 200 µm resolution. As
their response to variations in oxygen concentrations is linear, the $O_2$ microelectrodes were calibrated
with a two-point calibration technique using the bottom water $O_2$ concentration determined by
Winkler titration and the anoxic pore waters. The pH microelectrodes were calibrated using NBS
buffers (pH 4.00, 7.00 and 9.00 at 20°C) and the spectrophotometrically determined pH of the bottom
waters was used to correct for the difference in the liquid junction potential between seawater and the
NBS buffers. Signal drift of $O_2$ and pH microelectrodes during profiling was checked to be less than

5 %.





*2.5 Sediment sampling, porosity measurements, and ex situ voltammetric profiling*
At each sampling station, sediment cores were collected using an UWITEC® single corer (length
60 cm, inner diameter 9 cm) within 30 m from the site where the landers were deployed and processed
within 30 minutes after collection. Sediment porosity profiles were determined by slicing one of the
cores with a 2 mm resolution until 10 mm depth, a 5 mm resolution until 60 mm, and a 10 mm
resolution down to the bottom of the cores. Porosity was calculated from the bottom water salinity,
an average sediment density of 2.65 g cm$^{-3}$ and the weight difference between the wet and dried
sediment after one week at 60 °C.
*Ex situ* voltammetric profiles were obtained in a separate core with a AIS, Inc. DLK-70
potentiostat in a three electrode configuration, including Hg/Au working microelectrode constructed
from Pyrex glass pulled to a tip of 0.4 mm diameter to minimize particle entrainment during the
profiles (Luther et al., 2008), an Ag/AgCl reference electrode, and a platinum counter electrode. The
Hg/Au voltammetric electrode was deployed in the sediment using a DLK MAN-1 micromanipulator
(AIS, Inc). Using a combination of linear sweep and anodic and cathodic square wave voltammetry,
Hg/Au voltammetric microelectrodes are able to simultaneous quantify dissolved $O_2$, $Mn^{2+}$, $Fe^{2+}$,
total dissolved sulfide ($\Sigma H_2S = H_2S + HS^- + S^0 + S_x^{2-}$), as well as organic complexes of Fe(III) (org-
Fe(III)) and iron sulfide clusters ($FeS_{aq}$), which are not quantifiable but reported in normalized current
intensities (Tercier-Waeber and Taillefert, 2008). Hg/Au microelectrodes were calibrated for
dissolved $O_2$ using *in situ* temperature and salinity of the overlying waters to determine the dissolved
$O_2$ concentrations at saturation (Luther et al., 2008). They were also calibrated externally with $MnCl_2$
to quantify all other species according to the pilot ion method (Luther et al., 2008). All voltammetric
data was integrated using VOLTINT, a semiautomated Matlab® script with peak recognition software
(Bristow and Taillefert, 2008).



### 166    *2.6 Pore water and solid phase extractions and analyses*

Sediment pore waters were extracted using rhizon filters with a mean pore size of 0.1 μm
(Seeberg-Elverfeldt et al., 2005) in a glove bag that was extensively flushed with $N_2$ to create an
anaerobic atmosphere. Pore waters were analyzed immediately onboard for dissolved phosphate
concentrations using the paramolybdate method (Murphy and Riley, 1962) as well as for dissolved
$Fe^{2+}$ and total dissolved iron concentrations using the ferrozine method (Stookey, 1970). Pore water
and bottom water fractions were poisoned with $HgCl_2$ for TA and DIC, acidified for sulfate, and stored
at 4 °C until analysis in the laboratory. Total alkalinity was measured by open cell titration with 0.01
M HCl (Dickson et al., 2007). DIC concentrations were analyzed with a DIC analyzer
(Apollo/SciTech®) on 1 ml samples as previously described (Rassmann et al., 2016). The TA and DIC
methods were calibrated using certified reference materials for oceanic $CO_2$ measurements provided
by the Scripps Institution of Oceanography (batch n°136). The relative uncertainty for both DIC and
TA was ± 0.5 % of the final value. Sulfate concentrations were quantified after dilution by ion
chromatography on an ICS 1000 chromatograph (Dionex) with an IonPac AS 9 HC column and AG
9 HC guard by suppressed conductivity with an AERS 500 suppressor (ThermoFisher Scientific). A
9 mM solution of $Na_2CO_3$, at a flow rate of 1 ml min$^{-1}$ was used as the eluent. The relative uncertainty
of this method was ± 1.6 %. Separate pore water fractions were frozen at -18 °C for sulfate analysis
by high performance liquid chromatography using a Waters, Inc. 1525 binary pump with Waters 2487
absorbance detector at 215 nm and a Metrohm Metrosep A Supp 5 anion exchange column (150 mm
x 4.0 mm) with a 1.0 mM NaHCO$_3$ / 3.2 mM $Na_2CO_3$ eluent at a flow rate of 0.7 ml min$^{-1}$ (Beckler
et al., 2014). To measure ammonium ($NH^+_4$) concentrations, samples were diluted and analysed using
the indophenol blue method (Grasshof et al., 1983). The uncertainty of the method was about 5 %.
Pore water fractions were also acidified with 2 % HCl for $Ca^{2+}$ analysis by inductively-coupled
plasma atomic emission spectroscopy (Ultima 2, Horiba Scientific). The method was validated with
mono-elemental standards and standard solutions (IAPSO, CASS-4, and NASS-6 seawater reference
materials) and displayed an external relative uncertainty of ± 2-3 % depending on the sample series.



Close to the Rhône River mouth, at station A, Z, and AK, one core was subsampled from the side
with 1cm diameter corers made of cut 10-ml syringes every 5 cm through pre-drilled holes. The
content of these subsamples was carefully inserted in gas tight vials containing deionized water and
HgCl$_2$ solution and kept at 4°C until methane analysis. Dissolved methane was quantified after
degassing of the pore waters into the headspace and quantified by gas chromatography with a relative
uncertainty of ± 5 % (Sarradin and Caprais, 1996). The position of the sulfate-methane transition zone
(SMTZ) was determined as the zone around the depth where $[SO_4^{2-}] = [CH_4]$ (Komada et al., 2016).
Finally, acid volatile sulfur (AVS) for the determination of FeS$_s$ was extracted from the same sediment
used for the pore water extractions and conducted in triplicate by cold acid distillation of H$_2$S (g)
under anoxic conditions that was trapped by NaOH and quantified voltammetrically (Henneke et al.,

1991).

*2.7 Nanoparticulate FeS and ion activity product for FeS precipitation*
As a significant fraction of FeS nanoparticles may pass through the rhizon filters (0.1μm) used
to extract pore waters (Nakayama et al., 2016) and the ferrozine method is well known to dissolve
FeS nanoparticles (Davison et al., 1998), the difference between spectrophotometrically-determined
Fe$^{2+}$ concentrations ($[\Sigma Fe^{2+}]_{FR}$) and electrochemically-determined Fe$^{2+}$ concentrations ($[Fe^{2+}_{echem}]$) in
the pore waters was attributed to FeS nanoparticles (FeS$_0$), as demonstrated previously (Bura-Nakic
et al., 2009; Eq. 2).

$$[FeS_0] = [\textstyle\sum Fe^{2+}]_{FR} - [Fe_{echem}^{2+}] \tag{2}$$

In this interpretation, FeS$_0$ nanoparticles encompass both the molecular clusters of FeS (FeS$_{aq}$)
detected electrochemically, which must be smaller than 5 nm in diameter to diffuse to the electrode
(Buffle, 1988), and the larger FeS nanoparticles that are not detected voltammetrically. The ionic
activity product (IAP) for the precipitation of FeS was calculated using Eq. 3 (Beckler et al., 2016),

$$pIAP = \log\left(\frac{\gamma_{Fe(II)}[Fe^{2+}]\gamma_{HS}\alpha_{HS}\Sigma H_2 S}{\{H^+\}}\right) \tag{3}$$

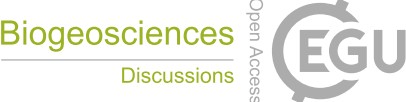



where $\gamma_{Fe(II)}$ and $\gamma_{HS}$ represent the activity coefficients of $Fe^{2+}$ and $HS^{--}$, $\alpha_{HS} = \frac{\{H^+\}K_{a1}}{\{H^+\}^2+\{H^+\}K_{a1}+K_{a1}K_{a2}}$
is calculated with the acid dissociation constant of $H_2S$ ($K_{a1} = 10^{-6.88}$) and $HS^-$ ($K_{a2} = 10^{-17}$) (Davison,
1991), and $\{H^+\}$ is the activity of the proton. Activity coefficients of $Fe^{2+}$ (Millero and Schreiber,
1982) and $HS^-$ (Millero, 1983) were calculated using Pitzer parameters.
*2.8 Calculations of oxygen uptake and AVS burial rates*
Diffusive oxygen uptake (DOU) fluxes were calculated using Fick's first law (Berner, 1980, Eq.

4),

$$DOU = -\emptyset \cdot D_s \cdot \frac{d[O_2]}{dz}\Big|_{z=0} \tag{4}$$

where $\phi$ is the sediment porosity, $D_s$ is the apparent diffusion coefficient in the sediments, and
$\frac{d[O_2]}{dz}\Big|_{z=0}$ is the oxygen gradient at the SWI. The $D_s$ coefficients were adjusted for diffusion in a porous
environment according to: $D_s = \frac{D_0}{(1+3\cdot(1-\emptyset))}$ with the diffusion coefficient in free water ($D_0$) chosen
according to Broecker and Peng (1974) and recalculated to *in situ* temperature by the Stokes-Einstein
relation (Li and Gregory, 1974).
AVS burial fluxes were estimated using available sedimentation rates ($\omega$ from Charmasson et al.
(1998) and Miralles et al. (2005)), average AVS concentrations and porosities of each sediment core,
according to Eq. 5,

$$AVS_{burial} = (1 - \emptyset) \cdot \omega \cdot AVS \cdot \rho \tag{5}$$

where $\phi$ is the sediment porosity, $\omega$ the sedimentation rate, and $\rho$ the sediment dry bulk density.
*2.9 Stoichiometric ratios*
To determine the relationship between net TA and DIC production and to establish whether
sulfate reduction represents the main source of TA and DIC in these sediments, stoichiometric ratios
of the relative production of TA compared to DIC ($r_{AD}$), as well as TA ($r_{AS}$) and DIC ($r_{DS}$) compared
to sulfate consumption, were calculated from the pore water data and compared to theoretical ratios





from the reaction stoichiometries (Table 1). Experimental stoichiometric ratios were obtained from
the slope and standard deviation of the linear regression of TA, DIC, and sulfate property-property
plots of concentration changes with respect to bottom water concentrations at each depth in the pore
waters ($\Delta$TA, $\Delta$DIC and $\Delta SO_4^{2-}$) relative to each other after correcting for differences in TA, DIC and
sulfate diffusion in the sediments (Berner, 1980, Eq. 6),
$$r_{ij} = \frac{D_i \cdot \Delta i}{D_j \cdot \Delta j} \qquad (6)$$
where i is the concentration of either TA or DIC, j the concentration of $SO_4^{2-}$ or DIC and $D_i$ and $D_j$
are the corresponding diffusion coefficients. At the pH of the pore waters (pH ~ 7.5), more than 95 %
of DIC and carbonate alkalinity are composed of bicarbonate ion ($HCO_3^-$). Given the relatively small
difference in the diffusion coefficients of $HCO_3^-$ and $CO_3^{2-}$ (11.8 and 9.55 x $10^{-6}$ $cm^2$ $s^{-1}$ at 25°C, Li
and Gregory, 1974) and the high proportion of $HCO_3^-$ relative to $CO_3^{2-}$, the diffusion coefficient of
$HCO_3^-$ was adopted for both TA and DIC diffusion.
The effect of the precipitation or dissolution of calcium carbonate on TA and DIC variations was
also accounted for by considering the $Ca^{2+}$ concentration gradients in the pore waters. For these
calculations, the absolute value of the $Ca^{2+}$ concentration relative to its bottom water concentration
($\Delta Ca^{2+}$) was added to the $\Delta$TA or $\Delta$DIC after taking the corresponding diffusion coefficients into
account ($D_{TA}\Delta$ TA + $2D_{Ca}|\Delta Ca^{2+}|$ for alkalinity and $D_{DIC}\Delta$ DIC + $D_{Ca}|\Delta Ca^{2+}|$ for DIC) and plotted
against $D_{SO42}\cdot\Delta SO_4^{2-}$. The calculated slope provided a stoichiometric ratio corrected for the
precipitation of calcium carbonate ($r_{IJc}$). Pore water saturation states, regarding Calcite ($\Omega_{Ca}$), were
calculated according to the equation proposed by Mucci (1983) and Millero (1995).

## 3.Results

### 3.1 Bottom water and surface sediment characteristics

At all stations, bottom water salinities ranged from 37.5 to 38.0 and temperatures varied from
14.7 to 20.6 °C (Table 2). Total alkalinity and DIC concentrations (average TA = $2.60 \pm 0.01$ mM and

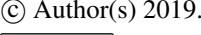



average DIC = 2.30 ± 0.02 mM, Table 2) were relatively high compared to the Mediterranean Sea
average, but common for the Gulf of Lions (Cossarini et al., 2015). The $pH_T$ of the bottom waters
varied from 8.05 to 8.09 with the highest value observed at station AK and the lowest at station E.
Although the oxygen concentration decreased with water depth, bottom waters were always well
ventilated, with dissolved $O_2$ concentrations higher than 220 µmol L$^{-1}$. Sediment porosity ranged
between 0.7 and 0.8 at the SWI, and they were similar at all stations between 20 and 400 mm depth
(Table 2).
*3.2 Benthic total and diffusive fluxes*
The *in situ* pH and $O_2$ microprofiles reflected the differences between the three study domains
under the influence of the Rhône River plume (Fig. 2). In the proximal zone (stations A and Z), the
oxygen penetration depth was only 1.5 to 2.5 mm into the sediment as also indicated by separate
voltammetric measurements (Fig. 5). The oxygen penetration depth increased from 2 to 6 mm at
station K and reached 8 to 11 mm at the most offshore station E. As a result of bad weather conditions,
no exploitable *in situ* microprofiles were recorded at stations AK and B, though *ex situ* voltammetric
profiles determined oxygen penetration depths of 4 and 2 mm, respectively (Fig. 5). All pH microfiles
indicated a pH minimum between 7.2 and 7.4 just below the OPD followed by an increase to between
7.5 and 7.6 in the manganous/ferruginous layers of the sediment around 5 mm inshore and below 12
mm offshore (Fig. 2). Below this depth, pH stabilizes.
The benthic chamber was deployed once at stations A and E and twice at station Z (Z' is the
replicate). Total alkalinity and DIC concentrations increased linearly with time in the chamber, but
concentration changes decreased along the nearshore-offshore transect (Fig. 3). The highest benthic
fluxes were recorded for the two deployments at station Z, with TA fluxes of 73.9 ± 20.6 and 56.0 ±
17.8 mmol m$^{-2}$ d$^{-1}$ and DIC fluxes of 78.3 ± 10.9 and 37.2 ± 7.2 mmol m$^{-2}$ d$^{-1}$ (Fig. 4, Table 2). At
station A, the benthic TA and DIC fluxes reached lower values of 14.3 ± 1.6 and 17.8 ± 1.6 mmol m$^{-}$
$^2$ d$^{-1}$, respectively, while benthic fluxes were lowest at station E, with a TA flux of 3.7 ± 0.9 mmol m$^{-}$



$^2$ d$^{-1}$ and a DIC flux of 9.9 ± 0.9 mmol m$^{-2}$ d$^{-1}$. In parallel, DOU fluxes reached 10.2 ± 1.3 and 10.4
± 0.9 mmol m$^{-2}$ d$^{-1}$ at stations A and Z and decreased offshore to 5.9 ± 1.0 mmol m$^{-2}$ d$^{-1}$ at station K
and 3.6 ± 0.6 mmol m$^{-2}$ d$^{-1}$ at station E (Fig. 4, Table 2). Although the relative importance of DOU
compared to TA and DIC fluxes increased offshore, the TA and DIC fluxes were always between 2
and 8 times larger than the DOU fluxes (Fig. 4).
*3.3 Electrochemistry profiles*
Dissolved Fe$^{2+}$ concentrations as a function of depth in the sediment mirrored the voltammetric
signals of soluble organic-Fe(III) complexes at stations A, Z, AK, B, and K (Fig. 5). High
concentrations of dissolved Fe$^{2+}$ were observed in the proximal domain at stations A (341 ± 22 μM)
and Z (234 ± 25 μM), where dissolved $\Sigma H_2S$ was not detected (Fig. 5). At station AK, the shallowest
station in the prodelta domain, dissolved Fe$^{2+}$ increased to a maximum concentration of 255 μM
around 2 cm depth, then decreased with sediment depth as FeS$_{aq}$ below 6.5 cm and small
concentrations of dissolved $\Sigma H_2S$ around 17 cm were produced (Fig. 5). The two deeper prodelta
stations, B and K, displayed lower Fe$^{2+}$ concentrations, including one peak not exceeding 81 μM
(station B) or 73 μM (station K) in the top 2 cm of the sediment and a second peak not exceeding 50
μM between 12-14 cm (station B) and 86 μM between 5-7.5 cm (station K) in the sediment. Although
FeS$_{aq}$ was only detected below 15 cm at station K, $\Sigma H_2S$ was produced in low concentrations (< 5
μM) around 6.5 cm at stations B and K (Fig. 5). A peak of Fe$^{2+}$ was initially formed in the top 5 cm
of the distal domain (station E) but decreased to a minimum value with depth and did not correlate
with the organic-Fe(III) voltammetric signals, which also remained low throughout the profile (Fig.
5). Finally, station E displayed generally low concentrations of $\Sigma H_2S$ in the pore waters (< 6 μM),
though the onset of $\Sigma H_2S$ production was much shallower (2.5 cm) and $\Sigma H_2S$ concentrations were
consistently higher throughout the profile than at any other stations.



*3.4 Geochemical characteristics of the pore waters and sediments*

Both TA and DIC concentrations increased rapidly within the pore waters (Fig. 6), likely reflecting the intensity of organic carbon mineralization rates in these sediments. At all stations, DIC pore water concentrations correlated well with TA (overall slope: $1.01 \pm 0.006$, $r^2=0.995$, n=134). The TA and DIC gradients were highest at stations A and Z, where maximum concentrations of both species reached around 55 mM. At station AK, TA and DIC concentrations reached a maximum of 15 mM at 25 cm depth but decreased to 6 mM at the bottom of the core. The maximum concentrations of TA and DIC of 35 mM observed at station B, were more comparable to the stations in the vicinity of the river mouth (stations A and Z) than other stations located in the prodelta domain (stations AK and K). At station K, TA and DIC concentrations reached 10 mM, whereas the lowest TA and DIC gradients were measured at station E, with concentrations reaching only 4.6 mM at the bottom of the cores (30 cm). Sulfate was completely removed from the pore waters at depths of 35, 24, and 45 cm at station A, Z, and B, respectively (Fig. 6). In turn, sulfate concentrations decreased to a minimum concentration of 20 mM at 29 and 24 cm depth at stations AK and K, whereas sulfate consumption was much smaller at station E with a minimum concentration of 28 mM (bottom water sulfate concentration was 31.4 mM). As a result, TA and DIC changes in concentration at a given depth were highly inversely correlated ($r^2 > 0.97$) with sulfate changes in concentration at stations A, Z, AK, B, and K (Table 3). At station E, sulfate variations in the observed depth were in the same order of magnitude as the measuring uncertainties. Simultaneously, TA and DIC demonstrated strong correlations ($r^2 > 0.97$) at stations A, Z, AK, B, and K (Table 3). In the proximal domain (stations A and Z), ammonium increased with sediment depth to concentrations > 3 mM (Fig. 6). At station B, ammonium reached concentrations > 2 mM with depth, whereas ammonium concentrations did not exceed 1.5 mM at station AK, 0.6 mM at station K, and 0.3 mM at station E. At all stations nitrite plus nitrate concentrations were less than 20 µM (data not shown). Significant methane concentrations (> 50 µM) were detected at the bottom of the sediment core at stations A, Z, and AK (Fig. 6), and a SMTZ was identified between 28 and 39 cm at station A and between 19 and 39 cm at



station Z. As methane was < 50 μM throughout the profile at station K and sulfate was not completely
consumed inside the sediment core at station AK, the SMTZ was not determined at these two stations.
Methane analyses were not carried out for the other stations.

The sediment pore waters were oversaturated with respect to calcite ($\Omega_{Ca} > 1$) at all stations. At

stations A, Z, and B, decreasing $Ca^{2+}$ concentrations in the pore waters indicated precipitation of
$CaCO_3$, whereas $Ca^{2+}$ concentrations remained close to the bottom water $Ca^{2+}$ concentrations (11.2
mM in Mediterranean waters) at the other stations (Fig. 6). Dissolved phosphate concentrations
($\Sigma PO_4^{3-}$) were relatively high (50-100 μM) throughout the profiles at stations A, AK, K and Z, and a
large increase in concentration (up to 160 μM at station Z) was observed at station AK, K and Z
between 15 and 22 cm. In turn, $\Sigma PO_4^{3-}$ production was minimal throughout station E pore waters (<
10 μM). Dissolved phosphate was not measured at station B. Sediment samples were analyzed for
AVS as a function of depth at stations A, AK, and E to assess one station in each domain (Fig. 6). At
station A, a peak in AVS (65 μmol $g^{-1}$) was measured around 8.0 cm followed by a second, smaller
peak (22 μmol $g^{-1}$) at 14 cm, after which AVS decreased with depth. The AVS concentrations were
low in the top portion of the sediment at station AK but increased with depth to 100 μmol $g^{-1}$ around
15 cm. At station E, only a small AVS peak of 20 μmol $g^{-1}$ was observed at 14 cm. Finally, large
concentrations of FeS nanoparticles ($FeS_0$) were found in the proximal and prodelta stations,
including two broad peaks and maximum concentrations around 1 mM at stations A and Z and a large
subsurface maximum up to 6 mM at 145 mm at station AK. These $FeS_0$ concentrations increased as
a function of depth to a relatively constant 0.5 mM below 4.5 cm at station B and below 12 cm at
station K, whereas they remained mostly negligible at station E (Fig. 6).
**4. Discussion**

In this study, we want to relate biogeochemical processes in the sediment to the observed TA and

DIC fluxes. Firstly, benthic TA and DIC fluxes in the Rhône River prodelta are compared to other
similar systems to evaluate their relative importance. In the following sections, the most likely



biogeochemical processes responsible for the high TA flux are identified based on the sediment depth
profiles collected. In particular, the role of iron sulfide mineral precipitation on the benthic TA flux is
established using a variety of analytical techniques, speciation calculations, and a mass balance
approach. Finally, the link between inputs to the sediment, carbon mineralization processes, sulfide
mineral burial, and the benthic TA flux is provided using a conceptual model.
*4.1 DIC and alkalinity fluxes from the sediment*
The sediments of the Rhône proximal and prodelta zones represent important sources of both
DIC and TA to the bottom waters (Fig. 4). The DIC fluxes observed in the proximal domain (18-78
mmol m$^{-2}$ d$^{-1}$ at station A and Z; Fig. 4) are in the range of previously measured fluxes in other deltas
where anaerobic mineralization processes are dominant, including Mississippi delta sediments from
core incubations (15-20 mmol m$^{-2}$ d$^{-1}$; Lehrter et al., 2012) or benthic chambers (36-53 mmol m$^{-2}$ d$^{-1}$
; Rowe et al., 2002), benthic chamber measurements of the Po River delta sediments and the Adriatic
shelf (15-25 mmol m$^{-2}$ d$^{-1}$ ; Hammond et al., 1999), or the Fly River delta during the most active
season (35-42 mmol m$^{-2}$ d$^{-1}$ ; Aller et al., 2008) and near the Guadalquivir River estuary (36-46 mmol
m$^{-2}$ d$^{-1}$, Ferron et al., 2009). In contrast, fewer alkalinity fluxes were measured in river deltas, though
those obtained from benthic chambers in the Danube and Dniester deltas in the Northwest Black Sea
(21-67 mmol m$^{-2}$ d$^{-1}$, Friedl et al., 1998), are within the range of values reported in this study (14-74
mmol m$^{-2}$ d$^{-1}$). Benthic TA fluxes obtained in the Guadalquivir estuary (24-30 mmol m$^{-2}$ d$^{-1}$; Ferron
et al., 2009) and the Adriatic shelf sediments off the Po River delta (0.5-10.4 mmol m$^{-2}$ d$^{-1}$; Hammond
et al., 1999) are in the lower range of TA fluxes measured in the present study. The biogeochemical
origin of these TA benthic fluxes is discussed in the next sections.
*4.2 The relative importance of nitrification/denitrification on the TA budget*
Denitrification is known as a benthic TA source to the bottom waters as 0.8 moles of TA are
produced for 1 mole of organic carbon oxidized by nitrate and the product N$_2$ does not react further



with dissolved oxygen (Table 1, Eq. 5; Thomas et al., 2009; Krummins et al., 2013; Brenner et al.,
2016). Published estimates of total denitrification rates in Rhône prodelta and shelf sediments range
between 4 mmol m$^{-2}$ d$^{-1}$ in the proximal zone to 1 mmol m$^{-2}$ d$^{-1}$ in the continental shelf (Pastor et al.,
2011). Conversion to alkalinity flux would provide a range between 0.8 and 3.2 mmol TA m$^{-2}$ d$^{-1}$. As
such, denitrification would account for < 10% of the TA flux in the proximal zone where substantial
fluxes were measured by *in situ* benthic chambers (Fig. 4). Furthermore, the only net production of
TA by denitrification must be related to external nitrate sources as nitrification (overall oxidation of
ammonium to nitrate) consumes 2 moles of TA per mole of ammonium transformed into nitrate (Table
1, Eq. 2; Hu et al., 2011a). As coastal sediments mostly display coupled nitrification-denitrification,
this process does only represent a small source of TA to the bottom waters (Brenner et al., 2016). It
can therefore be concluded that the contribution of denitrification to TA fluxes is minimal in the
proximal zone and could be proportionally more important on the shelf where TA fluxes are much
lower.

### 4.3 DIC and TA produced by sulfate reduction

Sulfate reduction typically represents a major organic carbon mineralization pathway in organic-

rich sediments that simultaneously produces two moles of total alkalinity (TA) and two moles of DIC
per mole of sulfate (Table 1, Eq. 6) (Canfield et al., 1993b; Burdige, 2011). Dissimilatory iron
reduction (Table 1, Eq. 7) in turn produces 1/4 moles of DIC and consumes 7/4 moles of H$^{+}$, resulting
in two moles of TA produced per mole of Fe. As these two processes equally produce two moles of
TA per mole of terminal electron acceptor (Table 1, Eq.6 for SO$_4^{2-}$ and Eq. 7 for Fe(OH)$_3$), they can
both contribute significantly to the bulk alkalinity production in sediment pore waters. The low
concentration of nitrate, relatively low production of reduced metals in the pore waters (Fig. 5), and
intense ammonium and DIC production in parallel with sulfate consumption at depth (Fig. 6) confirm
that sulfate reduction is one of the dominant mineralization pathways in the Rhône prodelta sediments
(Pastor et al., 2011; Rassmann et al., 2016). Experimentally-derived stoichiometric ratios of the



relative production of DIC and TA compared to sulfate consumption may indicate the dominant
reaction pathways responsible for the high alkalinity generated in these sediments (Burdige and
Komada, 2011). Factoring carbonate precipitation using the pore water $Ca^{2+}$ data, the $r_{DSc}$ were
determined to range between -2.05 and -1.86, except for one value at -1.37 (station B), whereas the
$r_{ASc}$ ratios ranged between -2.35 and -1.89  with the exception of station B at -1.58 (Table 3).
Theoretically, the $r_{DS}$ and $r_{AS}$ should equal -2.0 if sulfate reduction is the only control on DIC and TA
production (Table 1, Eq. 6), suggesting that, except at station B, the influence of other diagenetic
processes on $r_{ASc}$ and $r_{DSc}$ is limited. At station B, however the higher $r_{DSc}$ ratio (Table 3) may indicate
significant anaerobic oxidation of methane (AOM Table 1, Eq. 8) which generates a theoretical $r_{DS}$
of -1 (Borowsi et al., 1996; Komada et al., 2016). Unfortunately, methane sampling was not
performed at station B, preventing precise identification of AOM at this station.
*4.4 Formation of iron sulfide species*
Although the complete depletion of sulfate in the first 30 cm of the sediment at stations A, Z, and
B implies an equivalent production of dissolved sulfide ($\Sigma H_2S$) (Table 1, Eq. 6), pore waters displayed
little to no $\Sigma H_2S$ (Fig. 5). If all of the produced $\Sigma H_2S$ diffused upward and reacted in the oxic sediment
layer, the alkalinity produced by sulfate reduction would be consumed by the oxidation of $\Sigma H_2S$ by
dissolved $O_2$ and the pH should be lowered significantly given the large acidity generated by this
reaction (Table 1, Eq. 4). Although $\Sigma H_2S$ was nearly absent of the pore waters (Fig. 5), the pH
minimum was never lower than 7.2 and the observed alkalinity fluxes across the SWI were substantial
(Fig. 4), indicating that $\Sigma H_2S$ was removed from the pore waters below the oxic layer. Abiotic
reduction of Fe(III) oxides by $\Sigma H_2S$ (Table 1, Eq. 9), followed by precipitation of FeS in the anoxic
zone (Table 1, Eq. 10; Berner, 1970; Pyzik and Sommer, 1981; Carman and Rahm, 1997, Soetaert et
al., 2007), and eventually formation of pyrite (Table 1, Eq. 11; Rickard and Luther, 1997) may
represent a significant $\Sigma H_2S$ removal pathway. As the abiotic reduction of Fe(III) oxides by $\Sigma H_2S$
coupled with either FeS or FeS and pyrite precipitation (Table 1, Eq. 9-11) does overall not alter





alkalinity, bacterial sulfate reduction followed by abiotic precipitation of iron and sulfide from the
pore waters to either FeS or pyrite (Table 1, Eqs. 12 and 13) should result in $r_{AD} = 1$ and $r_{DS} = r_{AS} = $-
2. Formation of pyrite is accompanied by the consumption of molecular $H_2$ by sulfate-reducing
bacteria, resulting in a slight increase in the $r_{AD}$ and $r_{DS}$ to 1.1 and -1.81 for the overall reaction while
the $r_{AS}$ ratio should not change (Table 1, Eq. 14). Another possible pathway includes the concomitant
production of $Fe^{2+}$ by dissimilatory iron reduction (Table 1, Eq. 7) and $\Sigma H_2S$ by sulfate reduction
followed by precipitation of FeS. In this case, the net $r_{AD}$ and $r_{DS}$ ratios should decrease to 0.89 and -
2.25, whereas the $r_{AS}$ ratio should remain at -2 (Table 1, Eq. 15). With ensuing formation of pyrite,
theoretical mole ratios may change slightly to $r_{AD} = 0.94$ and $r_{DS} = -2.13$ without $H_2$ reoxidation (Table
1, Eq. 16), whereas $r_{AD}$ and $r_{DS}$ ratios of 1.06 and -1.89 should be reached with $H_2$ reoxidation by
sulfate-reducing bacteria (Table 1, Eq. 17). In both cases, the $r_{AS}$ ratio should remain at -2.

The observed range of $r_{ADc}$ (1.06 to 1.15) and $r_{DSc}$ (-2.05 to -1.86) ratios in the proximal and

prodelta stations, except at station B (Table 3), is fully compatible with sulfate reduction coupled to
iron reduction and FeS precipitation (possibly followed by pyritization), though $r_{ADc}$ and $r_{DSc}$ ratios
are not able to distinguish abiotic and microbial pathways of iron reduction. The occurrence of
dissimilatory iron reduction in the proximal and prodelta domains, however, is substantiated by
several other pieces of evidence. First, the production of soluble organic-Fe(III) complexes deeper
than the oxygen penetration depths (Fig. 5) indicates that these species did not result from the
oxidation of $Fe^{2+}$ by dissolved $O_2$ in the presence of organic ligands (Taillefert et al., 2000). Second,
as soluble organic-Fe(III) complexes are produced as intermediates in the reduction of Fe(III) oxides
by iron reducing bacteria (Taillefert et al., 2007; Jones et al., 2010), their concomitant detection with
$Fe^{2+}$ at all the stations in the proximal and prodelta domains (Fig. 5) suggests they were produced
during dissimilatory iron reduction. Third, the positive correlation between the current intensities of
organic-Fe(III) complexes and $Fe^{2+}$ concentrations is in line with the same correlation obtained in
iron-rich deep-sea sediments (Fig. 7) where sulfate reduction was not significant (Beckler et al.,
2016). Finally, as these organic-Fe(III) complexes are readily reduced by $\Sigma H_2S$ (Taillefert et al.,





2000), their presence in zones of sulfate reduction suggest these sediments are highly dynamic with
periods of intense sulfate reduction alternating with periods during which sulfate reduction is
repressed and replaced by microbial iron reduction. These dynamics may be controlled by the input
of organic and inorganic material from the Rhône River in the proximal domain, especially during
floods when most of the solid material is deposited on the seafloor (Cathalot et al., 2010; Pastor et
al., 2018).

### 464    *4.5 FeS precipitation*

The discrepancy between sulfate consumption and the low concentration of $\Sigma H_2S$ along with the

high TA fluxes clearly suggest that much of the sulfur was precipitated in the solid phase. Indeed,
AVS measurements show precipitation of FeS in the proximal and prodelta domains (Fig. 6). In
addition, the large phosphate concentrations observed at depth in the proximal and prodelta domains
(Fig. 6) suggest that $\Sigma PO_4^{3-}$ adsorbed to Fe(III) oxides was released in the pore waters during
secondary conversion of Fe(III) oxides to FeS (Anschutz et al., 1998; Rozan et al., 2002). More
importantly, large concentrations of nanoparticulate FeS ($FeS_0$ in the range of 1-6 mM) were
identified in the proximal and prodelta stations that decreased with distance from shore (Fig. 6). The
existence of $FeS_0$ suggests that large fractions of $Fe^{2+}$ and $\Sigma H_2S$ were actively removed from the pore
waters at the time of measurements and eventually immobilized under the form of sulfide minerals.
Although soluble $FeS_{aq}$ clusters detected electrochemically when the system is oversaturated with
respect to FeS (Theberge and Luther, 1997) are considered good indicators of the active precipitation
of iron sulfide minerals (Luther and Ferdelman, 1993; Davison et al., 1998; Taillefert et al., 2000),
they were rarely observed in the Rhône River delta (Fig. 5). Indeed, the ion activity products (pIAPs)
calculated at most stations indicate that pore waters were either undersaturated, as a result of the low
concentrations (stations AK, B, and K) or complete absence (stations A and Z) of dissolved sulfides,
or close to the solubility of amorphous FeS or mackinewite (Fig. 8). Collectively, the large
concentrations of dissolved $FeS_0$ compared to the small electrochemically active $FeS_{aq}$ complexes



and the generally low saturation state of the pore waters indicate that FeS was much more aggregated
during this time period. Overall, the presence of soluble organic-Fe(III) complexes along with
dissolved $Fe^{2+}$ throughout the profiles, the absence of $\Sigma H_2S$ and $FeS_{aq}$, and the large concentrations
of dissolved $FeS_0$ found in the pore waters despite complete removal of sulfate in the proximal and
some of the prodelta stations provide strong evidence of large FeS precipitation in a context where
sulfate-reducing conditions may alternate with iron-reducing conditions as already observed
seasonally in estuarine sediments (Taillefert et al., 2002).
*4.6. Benthic alkalinity flux as a result of iron sulfide burial*
As the extreme sedimentation rates (> 30 cm $yr^{-1}$) in the proximal domain prevent short-term
reoxidation of $\Sigma H_2S$, the burial of FeS should represent a net source of alkalinity in the pore waters
(Berner, 1982; Hu and Cai, 2011a; Brenner et al., 2016). With the precipitation of FeS, about 2 to 3
moles of alkalinity equivalent should be produced for each mole of sulfur precipitated (Table 1, Eqs.
12 and 15). Assuming concomitant dissimilatory iron and sulfate reduction dominate in the proximal
and prodelta zones, a conservative ratio of 2 moles of TA equivalent per mole of FeS precipitated can
be estimated (Table 1, Eq. 15). In this calculation, the alkalinity production flux was estimated from
the average AVS burial fluxes using Eq. 5, with the caveat that these flux comparisons are made
assuming steady-state which is questionable in such a dynamic system. Nonetheless, the average AVS
concentration of the proximal station (station A) was used, as the sedimentation rate at this station is
so high (>30 cm $y^{-1}$) that the entire sediment layer investigated is buried rapidly in a year. The
calculated AVS burial flux provides an alkalinity-equivalent flux of $25.0 \pm 7.7$ mmol $m^{-2}$ $d^{-1}$ in the
proximal domain (Table 4), which falls within the range of benthic alkalinity fluxes measured by
benthic chamber at stations A and Z (14.3 - 73.9 mmol $m^{-2}$ $d^{-1}$; Fig. 4 and Table 4). In the prodelta,
the alkalinity-equivalent flux is estimated at $9.8 \pm 2.8$ mmol $m^{-2}$ $d^{-1}$ at station AK (Table 4), which
unfortunately cannot be compared to benthic alkalinity fluxes as they were not measured. In the distal
domain, however, a low alkalinity-equivalent flux of $0.04 \pm 0.1$ mmol $m^{-2}$ $d^{-1}$ is estimated from the





average AVS burial flux at station E. This flux is much lower than the $3.7 \pm 0.9$ mmol m$^{-2}$ d$^{-1}$ flux
measured by benthic chamber (Fig. 4), a difference that could be due to denitrification and shallow
carbonate dissolution.

4.7. Benthic alkalinity flux as a result of carbonate dissolution

Calcium carbonate dissolution below the sediment-water interface as a result of the acidity

generated by aerobic respiration may represent another possible contributor to TA fluxes as
demonstrated in carbonate-rich permeable sediments (Burdige and Zimmerman, 2002; Cyronak et
al., 2013; Rao et al., 2014). Both the water column ($\Omega_{Ca} = 5.5$) and the pore waters ($\Omega_{Ca} > 1$) of the
proximal zone are largely oversaturated with respect to calcite (Rassmann et al., 2016; Fig. 6). These
findings are corroborated by a large decrease in $Ca^{2+}$ concentration in the pore waters, indicating
$CaCO_3$ precipitation at depth in proximal zone sediments. Yet, the intense consumption of dissolved
oxygen in the first millimeters below the sediment-water interface generates a large pH decrease (Fig.
2) that may induce carbonate dissolution at this scale. Calcium carbonate saturation states at a
millimeter scale near the SWI were calculated from pH profiles and an interpolation of the centimetre-
scale DIC profiles using the SeaCarb software (Fig. 9). They show that in the proximal zone, the
saturation state with respect to calcite, which is the most abundant detrital carbonate in these
sediments (Rassmann et al., 2016), is always above 1.5. Such saturation state precludes massive
carbonate dissolution at the sediment surface and discounts shallow carbonate dissolution as playing
a large role on the benthic alkalinity fluxes observed in the proximal sediments. Minor quantities of
calcium carbonate may be dissolved in microniches where the pH could be lower than 7.4 or less
abundant carbonate forms (aragonite) may dissolve in the millimetric layers where this mineral is
close to undersaturation.  These processes, however, surely represent an insignificant flux in the
proximal zone compared to the large alkalinity generated by sulfate reduction and subsequent FeS
burial. At the distal station on the shelf (Station E, Fig. 9c), the saturation state was close to 1 which
may indicate a potential contribution of calcium carbonate dissolution to the benthic alkalinity flux.



### 4.8. Linking TA and DIC fluxes to mineralization processes


Overall, the present findings indicate that FeS burial modifies the alkalinity budget in the
proximal and prodelta sediments (Brenner et al., 2016). As the order of magnitude of the measured
benthic alkalinity fluxes is compatible with the alkalinity generated during the reduction of Fe(III)
oxides, sulfate, and subsequent FeS burial in the proximal zone, these processes are likely responsible
for the large alkalinity fluxes reported in this high-sedimentation delta and, potentially, other similar
systems (Hu and Cai, 2011a).
The biogeochemical cycling of C, Fe, S, and TA close to the Rhône River mouth can be
theoretically summarized as follows (Fig. 10): (i) the high pore water DIC concentrations resulting
from the production of metabolic $CO_2$ during organic carbon mineralization lead to benthic DIC
fluxes that are only modulated by the precipitation of carbonate minerals; (ii) the high pore water TA
concentrations result from intense iron and sulfate reduction as a result of the high supply of organic
matter and Fe(III) oxides to the sediment; (iii) the precipitation of FeS and the high sedimentation
rates near the river mouth preserve the majority of reduced iron and $\Sigma H_2S$ buried in the form of FeS
minerals and potentially pyrite within the anoxic sediments (Aller et al., 1986); and (iv) ultimately,
the TA-consuming reoxidation of reduced metabolites (i.e., $NH_4^+$, $\Sigma H_2S$, $Fe^{2+}$) is not important in the
oxic sediment layers, and a significant fraction of the anaerobically-produced TA is transferred across
the SWI (Fig. 10, red dashed line). In these conditions, anaerobic and aerobic processes are
decoupled, and the consumption of oxygen no longer reflects the overall respiration rates within these
sediments (Pastor et al., 2011) as observed by the relatively lower contribution of DOU fluxes
compared to TA and DIC fluxes in the proximal domain (Fig. 4).
In contrast, sedimentation rates (Table 2), overall respiration rates (Fig. 4), and the intensity of
iron and sulfate reduction (Fig. 6) decrease in the distal domain (station E), and as a consequence the
relative proportion of aerobic processes increases (Pastor et al., 2011). Despite the relatively small
decrease in pore water sulfate concentrations with depth and low $\Sigma H_2S$ concentrations ($< 10$ µmol L$^-$
$^1$) at the most offshore station E, $\Sigma H_2S$ concentrations were the highest of all the stations. These



findings likely reflect the fact that less riverine Fe(III) oxides were available for FeS precipitation.
With low sedimentation rates (0.1 to 1 cm yr$^{-1}$) and thus low input of organic matter and Fe(III)
oxides, the overall carbon turnover is decreased and the reduced by-products of sulfate and/or iron
reduction may be transported back to the oxic sediment layers to be reoxidized by dissolved oxygen.
In this case, the alkalinity generated by anaerobic respiration processes is consumed by reoxidation
of the reduced metabolites, and the flux of alkalinity near the SWI decreases to weak values at station
E (Fig. 4 and Fig. 10, black line).
The strong TA flux to the overlying waters measured in the Rhône River delta, may contribute,
along with riverine inputs, to the overall high alkalinity of the Gulf of Lions waters compared to the
Mediterranean average (Cossarini et al., 2015). However, the influence of the benthic TA flux on the
water column pH and ultimately on the absorption of atmospheric $CO_2$ depends mainly on the TA to
DIC benthic flux ratio ($F_{TA}/F_{DIC}$), vertical mixing in the water column, and thus the residence time of
the bottom waters (Hu and Cai, 2011b, Andersson and Mackenzie, 2012). The $F_{TA}/F_{DIC}$ ratios, ranging
between 0.8 and 1 in the proximal and prodelta zones of the Rhône River delta (Fig. 11), are in the
high range of a compilation of TA to DIC flux ratios obtained in different coastal systems and
continental shelves (expanded from Hu and Cai, 2011b). As these ratios do not exceed 1, alkalinity
generated in the sediments will not decrease $p$CO$_2$ in the bottom waters and thus not draw atmospheric
$CO_2$ into the coastal ocean. Yet, the large benthic TA fluxes generated from deltaic sediments and the
elevated $F_{TA}/F_{DIC}$ (>0.8), which were unknown in the Rhône River prodelta before this study, may
modify the carbonate cycle paradigm in these coastal regions.
**5. Conclusion**
In this study, benthic respiration, as well as benthic alkalinity and DIC fluxes were quantified in
the Rhône River delta using benthic landers. These measurements demonstrated that sediments from
the proximal and prodelta domains represent a strong source of alkalinity to the water column. The
highest alkalinity and DIC fluxes were detected in the vicinity of the Rhône River mouth and were



much stronger than fluxes of dissolved oxygen, indicating the decoupling of oxic and anoxic
biogeochemical processes. As pore water oversaturation with respect to calcite prevented carbonate
dissolution to occur over the entire sediment column, the high benthic alkalinity fluxes resulted from
the high intensity of anaerobic respiration processes, mainly via sulfate reduction and precipitation
of iron sulfide minerals, but also with some contributions from dissimilatory iron reduction and AOM.
The intensity of sulfate reduction in the proximal domain also resulted in the consumption of a 10-
20% fraction of the alkalinity and DIC by the precipitation of authigenic carbonates. As the reduced
metabolites $Fe^{2+}$ and $\sum H_2S$ produced by the mineralization of organic matter were buried in the solid
phase, alkalinity was not consumed by their reoxidation in the oxic sediment layers. Consequently, a
significant fraction of the total alkalinity generated in the pore waters was transferred to the bottom
waters (benthic flux of 14-74 mmol $m^{-2}$ $d^{-1}$). Although sulfate reduction dominated the proximal and
prodelta domains, evidence for dissimilatory reduction of Fe(III) oxides was simultaneously observed
in the depth profiles, suggesting that anaerobic processes in the Rhône River prodelta are dynamic
and potentially controlled by pulsed sediment accumulations. The intensity of the alkalinity and DIC
fluxes decreased offshore as the sedimentation rate and the relative importance of anaerobic
mineralization pathways compared to aerobic processes decreased. In these conditions the more
"classical" coupling between aerobic and anaerobic reactions occurs, hence producing much lower
benthic alkalinity fluxes. Overall, these findings suggest that deltaic sediments exposed to large
riverine inputs of inorganic and organic material may provide a large source of alkalinity to the
overlying waters and thus weaken the increase in $p$CO$_2$ more significantly than previously thought in
coastal waters.
**Acknowledgments**
The authors thank the captain and crews of the RV Tethys II for their support at sea and Bruno
Bombled for his technical help on-board and in the laboratory. We thank Gael Monvoisin for the
analysis of sulfate samples at GEOPS (Paris-Sud University), Joel Craig and Olivia Studebaker for



the analysis of nutrients and AVS at Georgia Tech, and Celine Liorzou for the ICP-AES measurements
at Pôle Spéctrométrie Océan in Brest. We are grateful to Sabine Kasten, Sandra Arndt, and Andrew
Dale for interesting discussions about the interactions of AOM with carbonates and iron minerals and
sediment dynamics. This research was funded by the project Mistrals/MERMEX-Rivers and
AMORAD (ANR-11-RSNR-0002) and the National Science Foundation (OCE-1438648).

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



**Figure Captions**

**Figure 1:** Map of the Rhône River prodelta with the stations investigated during the AMOR-B-Flux cruise in September 2015.

**Figure 2:** Dissolved oxygen and pH microprofiles recorded *in situ* at the sediment-water interface at stations A, Z, K, and E. Stations A and Z are located in the proximal zone, K in the prodelta, and E in the distal zone (i.e. continental shelf).

**Figure 3:** Temporal evolution of DIC and total alkalinity concentrations in the benthic chamber at stations A, Z (measured during two deployments), and E. Error bars represent analytical uncertainties determined from triplicate measurements. The benthic fluxes and their standard deviations are provided in the text, in Figure 4 and in Table 2.

**Figure 4:** DIC and TA fluxes measured with the benthic chamber and diffusive oxygen uptake (DOU) rates calculated from *in situ* microelectrode depth profiles at stations A, Z (measured during two deployments), and E. Error bars represent either uncertainties about the linear regression of the benthic DIC and TA gradients taking into account individual error bars of each data point or error propagation and standard deviations of multiple DOU measurements. Fluxes out of the sediment are positive and fluxes into the sediment are negative.

**Figure 5:** Depth profiles of dissolved $O_2$, $Mn^{2+}$, $Fe^{2+}$, org-Fe(III), $FeS_{aq}$, and $\Sigma H_2S$ concentrations measured electrochemically in intact sediment cores at stations A, Z, AK, K, B, and E. Org-Fe(III) and $FeS_{aq}$ are reported in normalized current intensities (nA).

**Figure 6:** Depth profiles of pore water TA, DIC, $SO_4^{2-}$, $NH_4^+$, $CH_4$, $Ca^{2+}$, nanoparticulate FeS ($FeS_0$), $\Sigma PO_4^{3-}$, and AVS concentrations along with the calcium carbonate (calcite) saturation state of the pore



waters ($\Omega_{Ca}$) at stations A, Z, AK, K, B, and E. Alternating symbol shapes indicate data collected from
duplicate long and short sediment cores. The calcium carbonate (calcite) saturation state ($\Omega_{Ca}$) and
pore water $FeS_0$ concentrations were calculated whereas AVS was determined from solid phase
extractions. The two horizontal lines identify the sulfate-methane transition zone (SMTZ) found at
stations A and Z. Error bars represent standard deviations of multiple measurements for the
concentrations and error propagation for $\Omega$. Concentrations of $CH_4$ were not measured at stations B,
K, and E.

**Figure 7:** Current intensities of organic-Fe(III) complexes as a function of $Fe^{2+}$ concentrations
measured at each depth at stations A, Z, AK, B, and K compared to the same data obtained from iron-
rich deep-sea sediments (Beckler et al., 2016).

**Figure 8:** Calculated pIAP values as a function of depth into the sediment compared to the $pK_{sp}$ of
amorphous FeS and mackinawite. Due to the lack of dissolved sulfide, the pIAP values in the pore
waters of station A and Z could not be calculated.

**Figure 9:** Average pore water saturation states with respect to calcite in the oxic sediment layers at
stations: a- Proximal (St. A, Z), b- prodelta (St. K), and c- distal (St. E) calculated using the DIC
gradients at the SWI together with the average measured pH microprofiles.

**Figure 10:** Conceptual model to visualize the link between the burial of iron sulfide minerals and
benthic alkalinity fluxes. The total alkalinity (TA) produced under anaerobic conditions at depth
diffuses upwards towards the aerobic sediment layer where it is consumed during reoxidation of $Fe^{2+}$
and $\Sigma H_2S$ by dissolved oxygen (black). If the precipitation of sulfide minerals is significant, the
reduced iron and sulfide metabolites produced during anaerobic respiration are not reoxidized by



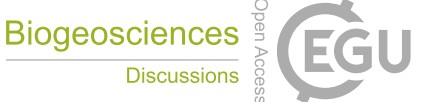

dissolved oxygen, and the TA produced is able to reach the bottom waters (red). The intensity of the
alkalinity flux into the bottom waters is indicated by the thickness of the arrow at the SWI.

**Figure 11:** TA to DIC benthic flux ratios as a function of depth at stations A, Z, and E of the Rhône
River delta compared to different coastal regions of water depth < 100 m where this ratio was
quantified from *in situ* benthic flux measurements (modified from Hu and Cai, 2011b). Other coastal
regions include Cadiz Bay and the Guadalquivir continental shelf (Spain; Ferron et al., 2009), the
Rio Tinto estuary (Spain; Ortega et al., 2008), the Po river delta and nearby Adriatic shelf (Italy;
Hammond et al. 1999), San Francisco Bay (USA; Hammond et al., 1985), and the California shelf
(USA; Berelson et al., 1996). The global coastal average TA to DIC flux ratio predicted from Krumins
et al., 2013 is also reported for reference. Note that this average is different from that reported by Hu
and Cai (2011b) which was corrected in their later publication (Hu and Cai, 2013).
**Table captions**
**Table 1:** Individual and consecutive microbial and abiotic reactions that affect the theoretical
$\Delta TA/\Delta DIC$ ($r_{AD}$), $\Delta DIC/\Delta SO_4$ ($r_{DS}$), and $\Delta TA/\Delta$sulfate ($r_{AS}$) stoichiometric ratios. Note that Eq. 14
and 17 include oxidation of $H_2$ produced by pyritization (Eq. 11) by sulfate-reducing bacteria.

**Table 2:** Sampling sites during the AMOR-B-Flux cruise in September 2015 and main characteristics
of bottom waters; dist. = distance to the Rhône River mouth; $\omega$ = sedimentation rate; Station Z was
sampled twice (Z on 09/08/15 and Z' on 09/14/15) to investigate temporal variability; n.d. = not
determined.

**Table 3:** Diffusion-corrected stoichiometric ratios corrected ($r_{ADc}$ , $r_{DSc}$, and $r_{ASc}$) or not ($r_{AD}$, $r_{DS}$, and
$r_{AS}$) for carbonate precipitation along with their associated determination coefficients ($r^2$) from linear
regression coefficients.






**Table 4:** Calculated FeS burial fluxes and their TA-equivalent production at each station compared
to TA benthic fluxes measured; n.d. = not determined.



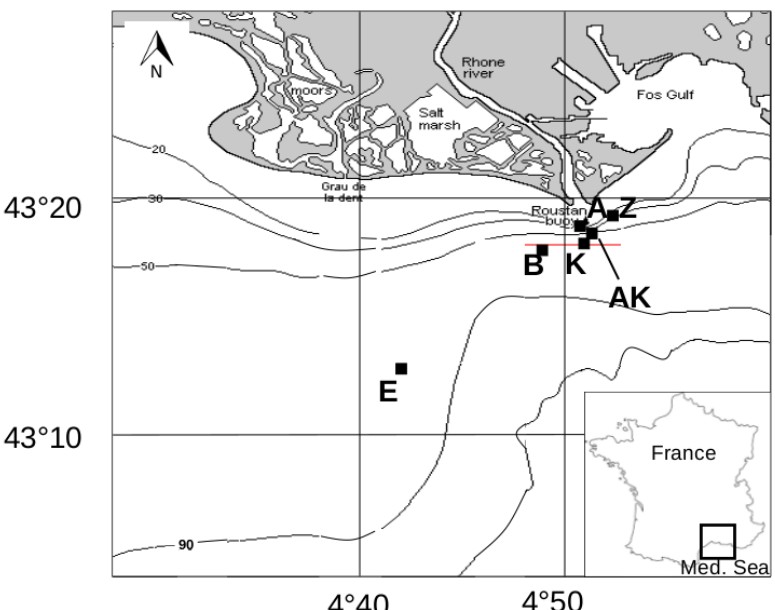

**Figure 1:** Figure 1

**Figure 2:** Figure 2



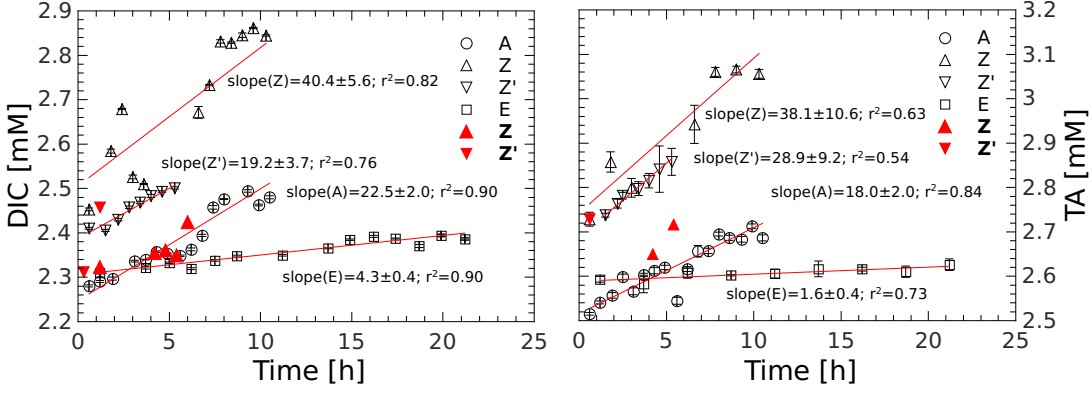

**Figure 3:** Figure 3

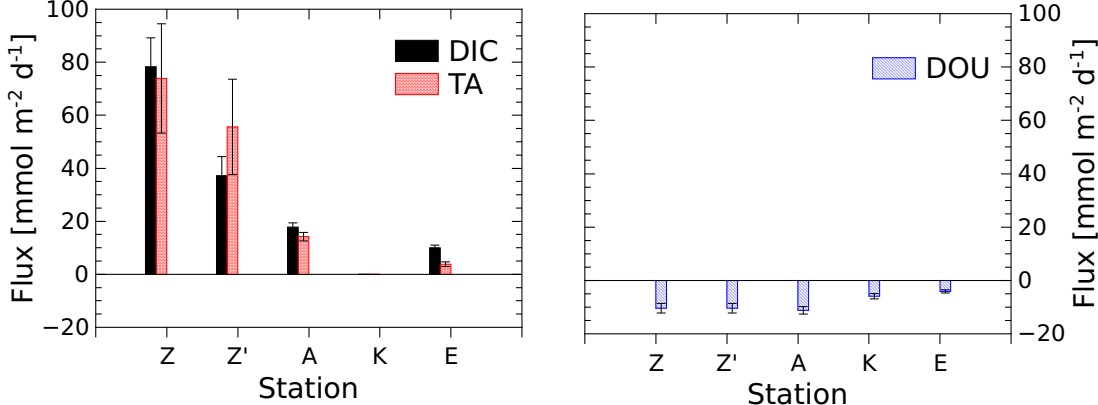

**Figure 4:** Figure 4





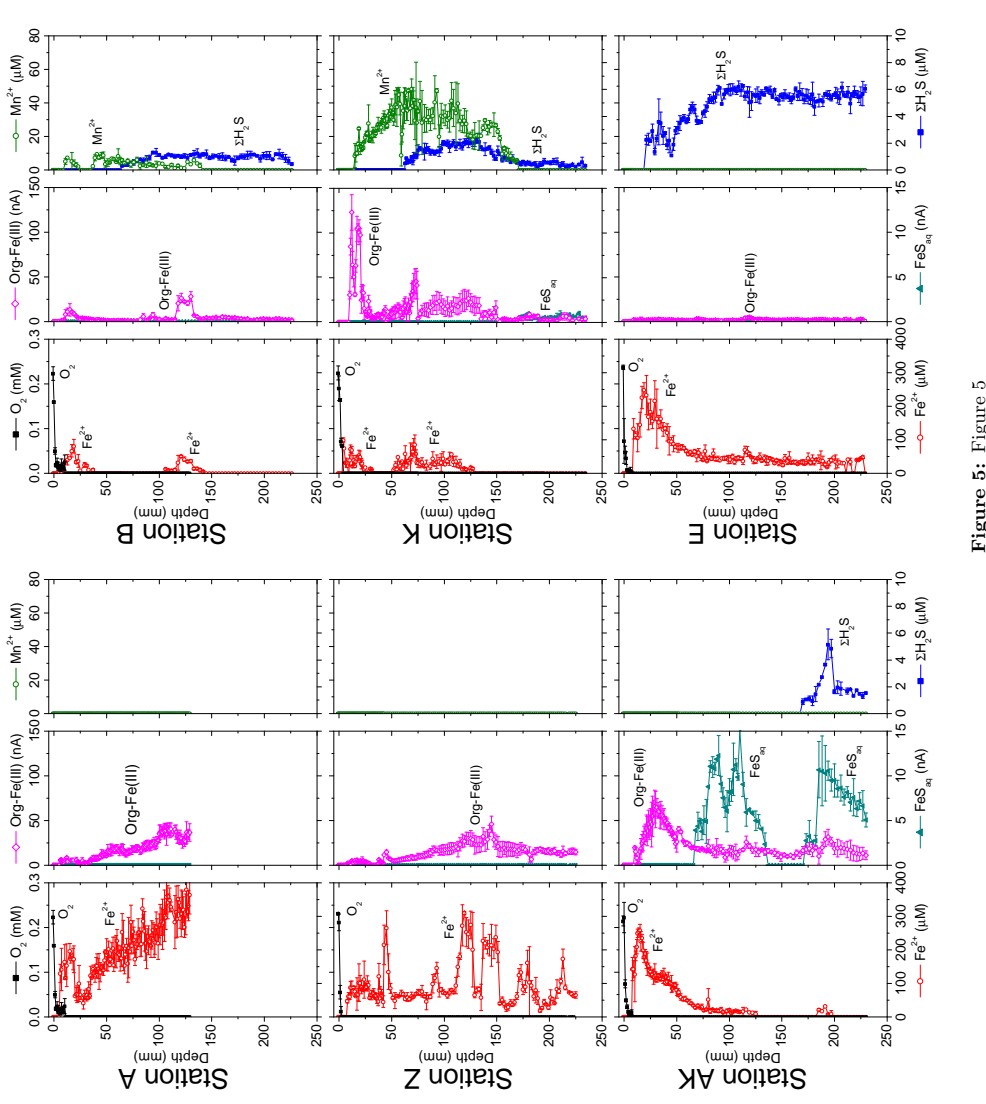

**Figure 5:** Figure 5





**Figure 6:** Figure6



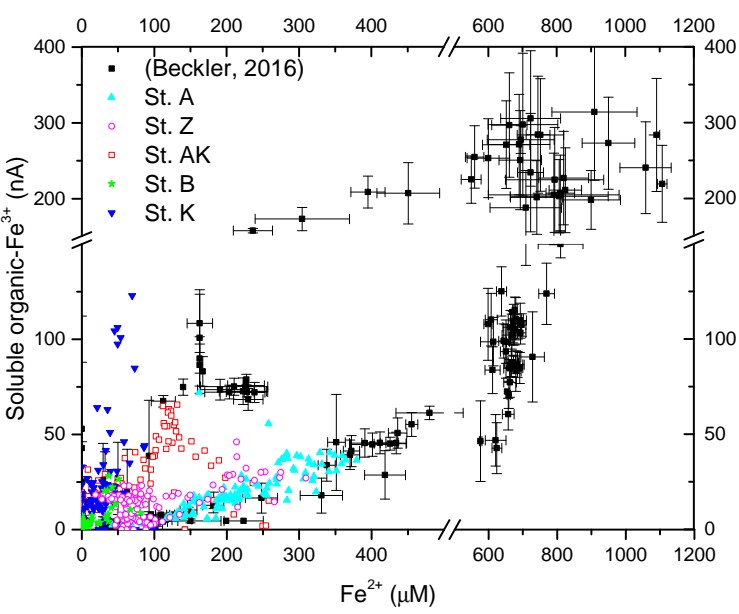

**Figure 7:** Figure 7



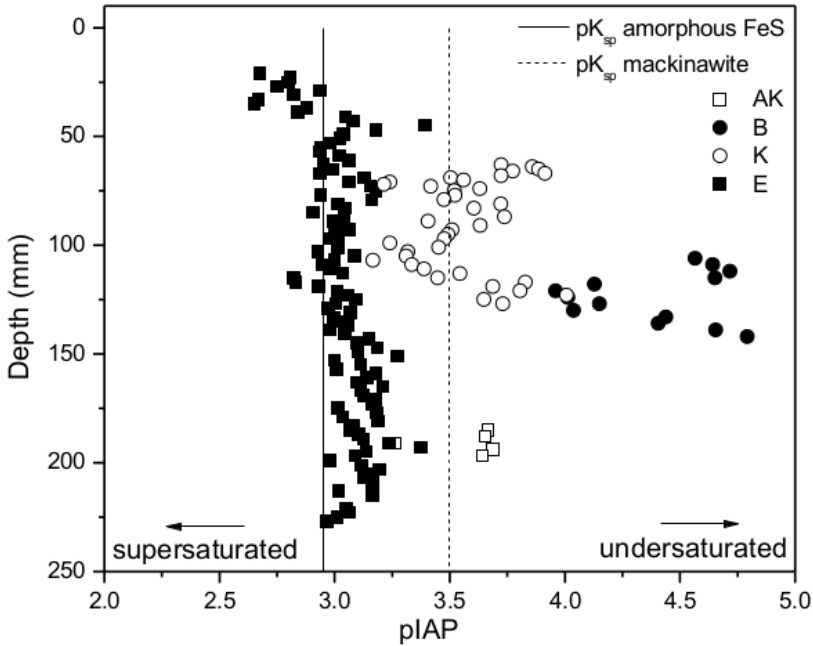

**Figure 8:** Figure 8

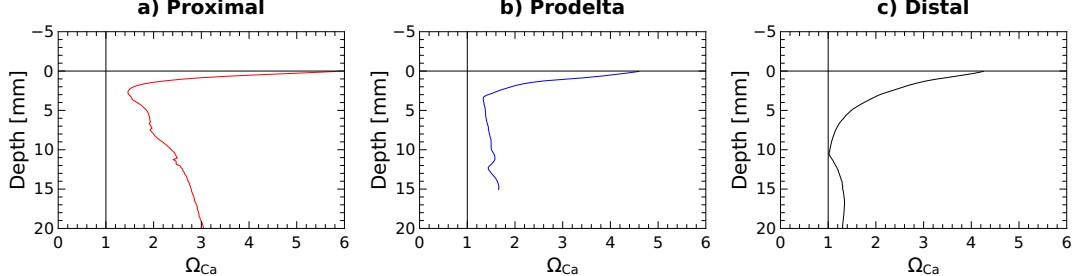

**Figure 9:** Figure 9





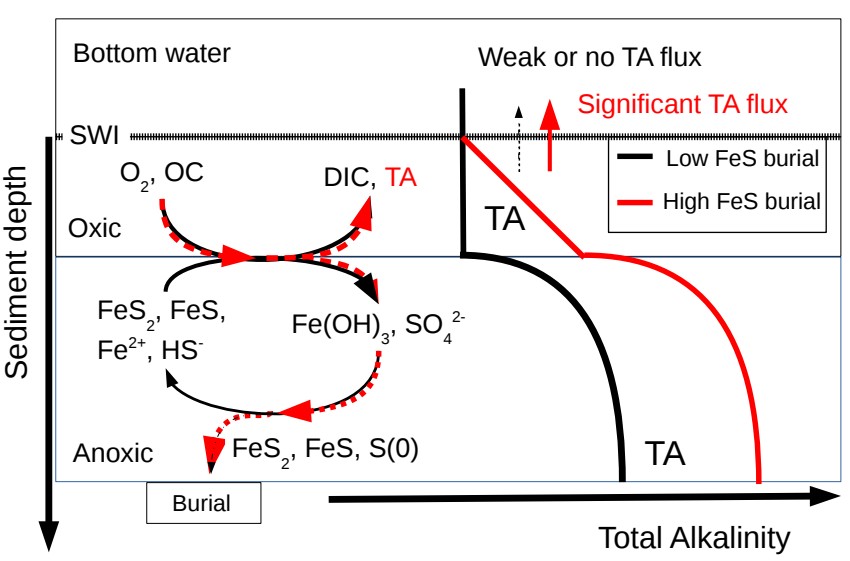

**Figure 10:** Figure 10





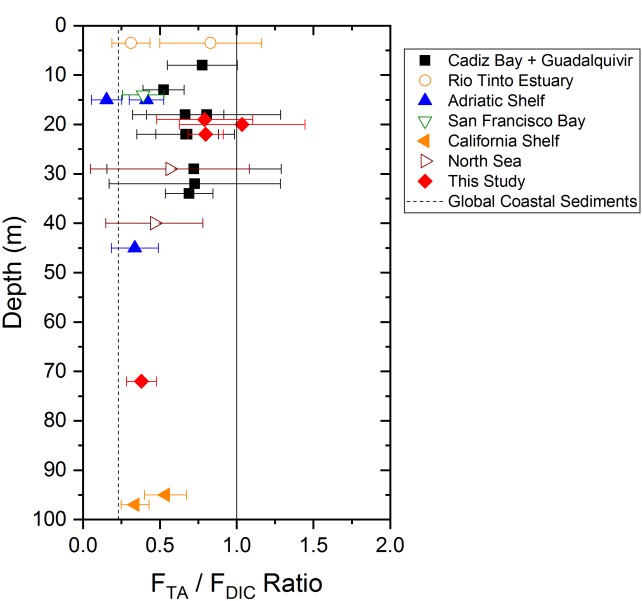

**Figure 11:** Figure 11



Table 1. Individual and consecutive microbial and abiotic reactions that affect the theoretical $\Delta TA/\Delta DIC$ ($r_{AD}$), $\Delta DIC/\Delta$sulfate ($r_{DS}$), and $\Delta TA/\Delta$sulfate ($r_{AS}$) stoichiometric ratios. Note that Eq. 13 and 16 include oxidation of $H_2$ by sulfate reducing bacteria.

| | Individual Reactions | $r_{AD}$ | $r_{DS}$ | $r_{AS}$ |
|---|---|---|---|---|
| (1) | $Ca^{2+} + HCO_3^- \rightarrow CaCO_3 + H^+$ | $\frac{-2}{-1} = 2$ | - | - |
| (2) | $NH_4^+ + 2O_2 \rightarrow NO_3^- + 2H^+ + H_2O$ | $\frac{-2}{0}$ | - | - |
| (3) | $Fe^{2+} + \frac{1}{4}O_2 + \frac{5}{2}H_2O \rightarrow Fe(OH)_3 + 2H^+$ | $\frac{-2}{0}$ | - | - |
| (4) | $H_2S + 2O_2 \rightarrow SO_4^{2-} + 2H^+$ | $\frac{-2}{0}$ | - | $\frac{-2}{+1} = -2$ |
| (5) | $CH_2O + \frac{4}{5}NO_3^- - \frac{2}{5}H^+ \rightarrow HCO_3^- + \frac{2}{5}N_2 + \frac{2}{5}H_2O$ | $\frac{4/5}{+1} = 0.8$ | - | - |
| (6) | $2CH_2O + SO_4^{2-} \rightarrow 2HCO_3^- + H_2S$ | $\frac{+2}{+2} = 1$ | $\frac{+2}{-1} = -2$ | - |
| (7) | $CH_2O + 4Fe(OH)_3 + 7H^+ \rightarrow HCO_3^- + 4Fe^{2+} + 10H_2O$ | $\frac{+8}{+1} = 8$ | - | - |
| (8) | $CH_4 + SO_4^{2-} + H^+ \rightarrow HCO_3^- + H_2S + H_2O$ | $\frac{+2}{+1} = 2$ | $\frac{+1}{-1} = -1$ | $\frac{+2}{-1} = -2$ |
| (9) | $Fe(OH)_{3(s)} + \frac{1}{2}H_2S + 2H^+ \rightarrow Fe^{2+} + \frac{1}{2}S(0) + 3H_2O$ | $\frac{+2}{0}$ | - | - |
| (10) | $Fe^{2+} + H_2S \rightarrow FeS_{(s)} + 2H^+$ | $\frac{-2}{0}$ | - | - |
| (11) | $FeS_{(s)} + H_2S \rightarrow FeS_{2(s)} + H_2$ | $\frac{0}{0}$ | - | - |
| | **Consecutive Reactions** | $r_{AD}$ | $r_{DS}$ | $r_{AS}$ |
| | Sulfate reduction, abiotic reduction of Fe(III) oxides, and precipitation of sulfide minerals | | | |
| (12) | $2CH_2O + SO_4^{2-} + Fe(OH)_{3(s)} \rightarrow 2HCO_3^- + \frac{1}{3}S(0) + \frac{2}{3}FeS_{(s)} + 2H_2O$ | $\frac{+2}{+2} = 1$ | $\frac{+2}{-1} = -2$ | $\frac{+2}{-1} = -2$ |
| (13) | $2CH_2O + SO_4^{2-} + \frac{2}{5}Fe(OH)_{3(s)} \rightarrow 2HCO_3^- + \frac{1}{5}S(0) + \frac{2}{5}FeS_{2(s)} + \frac{6}{5}H_2O + \frac{2}{5}H_2$ | $\frac{+2}{+2} = 1$ | $\frac{+2}{-1} = -2$ | $\frac{+2}{-1} = -2$ |
| (14) | $\frac{20}{11}CH_2O + SO_4^{2-} + \frac{4}{11}Fe(OH)_{3(s)} + \frac{2}{11}H^+ \rightarrow \frac{20}{11}HCO_3^- + \frac{2}{11}S(0) + \frac{4}{11}FeS_{2(s)} + \frac{1}{11}H_2S + \frac{16}{11}H_2O$ | $\frac{+2}{+\frac{20}{11}} = 1.1$ | $\frac{+\frac{20}{11}}{-1} = -1.8$ | $\frac{+2}{-1} = -2$ |
| | Concomitant dissimilatory iron and sulfate reduction with precipitation of sulfide minerals | | | |
| (15) | $\frac{9}{4}CH_2O + SO_4^{2-} + Fe(OH)_{3(s)} \rightarrow \frac{9}{4}HCO_3^- + \frac{1}{4}H^+ + FeS_{(s)} + \frac{5}{2}H_2O$ | $\frac{+2}{+\frac{9}{4}} = 0.89$ | $\frac{+\frac{9}{4}}{-1} = -2.25$ | $\frac{+2}{-1} = -2$ |
| (16) | $\frac{17}{8}CH_2O + SO_4^{2-} + \frac{1}{2}Fe(OH)_{3(s)} \rightarrow \frac{17}{8}HCO_3^- + \frac{1}{8}H^+ + \frac{1}{2}FeS_{2(s)} + \frac{5}{4}H_2O + \frac{1}{2}H_2$ | $\frac{+2}{+\frac{17}{8}} = 0.94$ | $\frac{+\frac{17}{8}}{-1} = -2.13$ | $\frac{+2}{-1} = -2$ |
| (17) | $\frac{17}{9}CH_2O + SO_4^{2-} + \frac{4}{9}Fe(OH)_{3(s)} + \frac{1}{9}H^+ \rightarrow \frac{17}{9}HCO_3^- + \frac{4}{9}FeS_{2(s)} + \frac{1}{9}H_2S + \frac{14}{9}H_2O$ | $\frac{+2}{+\frac{17}{9}} = 1.06$ | $\frac{+\frac{17}{9}}{-1} = -1.89$ | $\frac{+2}{-1} = -2$ |





Table 2. Sampling sites during the AMOR-B-Flux cruise in September 2015 and main characteristics of bottom waters; dist. = distance to the Rhône River mouth; ω = sedimentation rate; Station Z was sampled twice (Z on 09/08/15 and Z' on 09/14/15) to investigate temporal variability; n.d. = not determined.

| Domain | Proximal | | | Prodelta | | | Distal |
|---|---|---|---|---|---|---|---|
| Stations | A | Z | Z' | AK | B | K | E |
| Long. ° E | 4.850 | 4.868 | 4.868 | 4.853 | 4.833 | 4.858 | 4.684 |
| Lat. °N | 43.311 | 43.318 | 43.318 | 43.307 | 43.305 | 43.301 | 43.220 |
| Dist. [km] | 2.1 | 2.2 | 2.2 | 2.8 | 3 | 3.3 | 14.3 |
| Depth [m] | 20 | 20 | 20 | 42 | 50 | 58 | 72.5 |
| Temp. [°C] | 16.3 | 19.6 | 14.7 | 16.2 | 20.6 | 14.7 | 14.3 |
| Salinity | 37.5 | 37.6 | 37.7 | 37.7 | 38.0 | 37.7 | 37.8 |
| $O_2$ [µM] | 253.1 ± 0.3 | 249.5 ± 0.3 | 242.6 ± 0.2 | 250.2 ± 0.1 | n.d. | 241.8 ± 0.2 | 221.5 ± 0.3 |
| DIC [mM] | 2.29 ± 0.01 | 2.31 ± 0.01 | n.d. | 2.28 ± 0.01 | 2.27 ± 0.01 | 2.31 ± 0.01 | 2.33 ± 0.01 |
| TA [mM] | 2.61 ± 0.02 | 2.60 ± 0.01 | n.d. | 2.60 ± 0.02 | 2.60 ± 0.01 | 2.60 ± 0.02 | 2.61 ± 0.01 |
| $pH_T$ | 8.08 ± 0.01 | 8.06 ± 0.01 | 8.09 ± 0.01 | 8.09 ± 0.01 | 8.07 ± 0.01 | 8.08 ± 0.01 | 8.05 ± 0.01 |
| mean φ | 0.69 ± 0.04 | 0.65 ± 0.04 | 0.65 ± 0.04 | 0.68 ± 0.02 | 0.66 ± 0.03 | 0.65 ± 0.05 | 0.64 ± 0.04 |
| w [cm $yr^{-1}$] | 30 - 40[a] | | | 1 - 4[b] | | | 0.1 - 1[c] |
| Benthic fluxes (mmol $m^{-2}$ $d^{-1}$) | | | | | | | |
| TA flux | 14.3 ± 1.6 | 73.9 ± 20.6 | 56.0 ± 17.8 | n.d. | n.d. | n.d. | 3.7 ± 0.9 |
| DIC flux | 17.8 ± 1.6 | 78.3 ± 10.9 | 37.2 ± 7.2 | n.d. | n.d. | n.d. | 9.9 ± 0.9 |
| DOU | 10.2 ± 1.3 | 10.4 ± 0.9 | n.d. | n.d. | n.d. | 5.9 ± 1.0 | 3.6 ± 0.6 |

a. Data from Charmasson et al., 1998
b. Data from Lansard et al., 2009
c. Data from Miralles et al., 2005





Table 3. Diffusion-corrected stoichiometric ratios $r_{AD}$, $r_{DS}$, and $r_{AS}$ and their corresponding ratios corrected for carbonate precipitation ($r_{ADc}$, $r_{DSc}$, and $r_{ASc}$) along with their associated determination coefficients ($r^2$) from linear regression; n.d = not determined.

| Stations | A | Z | AK | B | K | E |
|---|---|---|---|---|---|---|
| $r_{AD}$ | $0.99 \pm 0.01$ | $1.08 \pm 0.02$ | $1.02 \pm 0.02$ | $1.02 \pm 0.01$ | $0.98 \pm 0.05$ | $0.90 \pm 0.04$ |
| $r^2$ | 0.998 | 0.997 | 0.998 | 0.999 | 0.986 | 0.984 |
| $r_{ADc}$ | $1.10 \pm 0.01$ | $1.16 \pm 0.03$ | $1.07 \pm 0.02$ | $1.15 \pm 0.02$ | $1.06 \pm 0.07$ | $1.15 \pm 0.11$ |
| $r^2$ | 0.999 | 0.997 | 0.996 | 0.998 | 0.974 | 0.885 |
| $r_{DS}$ | $-1.67 \pm 0.06$ | $-1.87 \pm 0.17$ | $-1.85 \pm 0.05$ | $-1.18 \pm 0.05$ | $-1.72 \pm 0.03$ | n.d. |
| $r^2$ | 0.990 | 0.969 | 0.995 | 0.988 | 0.997 | n.d. |
| $r_{DSc}$ | $-1.88 \pm 0.05$ | $-2.05 \pm 0.18$ | $-1.95 \pm 0.05$ | $-1.37 \pm 0.05$ | $-1.86 \pm 0.07$ | n.d. |
| $r^2$ | 0.994 | 0.972 | 0.996 | 0.990 | 0.994 | n.d. |
| $r_{AS}$ | $-1.66 \pm 0.07$ | $-2.03 \pm 0.17$ | $-1.89 \pm 0.06$ | $-1.21 \pm 0.04$ | $-1.69 \pm 0.07$ | n.d. |
| $r^2$ | 0.986 | 0.973 | 0.992 | 0.994 | 0.991 | n.d. |
| $r_{ASc}$ | $-2.07 \pm 0.05$ | $-2.35 \pm 0.14$ | $-2.01 \pm 0.06$ | $-1.58 \pm 0.05$ | $-1.89 \pm 0.14$ | n.d. |
| $r^2$ | 0.994 | 0.977 | 0.989 | 0.992 | 0.958 | n.d. |



Table 4. Calculated FeS burial fluxes and their TA-equivalent production at each station compared to alkalinity fluxes measured; n.d. = not determined.

| Stations | A and Z | AK | E |
|---|---|---|---|
| $\omega$ [cm yr$^{-1}$] | 30 | 3 | 0.1 |
| $\phi$ | 0.67 | 0.68 | 0.64 |
| Mean [AVS] [µmol g$^{-1}$] | 19.5 ± 4.9 | 45.0 ± 11.3 | 9.0 ± 2.3 |
| sediment density [g cm$^{-3}$] | 2.5 | 2.5 | 2.5 |
| FeS burial flux [mmol S m$^{-2}$ d$^{-1}$] | 12.5 ± 3.8 | 4.9 ± 1.4 | 0.02 ± 0.01 |
| TA-equivalent prod. (=2.0 FeS) [mmol TA m$^{-2}$ d$^{-1}$] | 25 ± 8 | 9.8 ± 2.8 | 0.04 ± 0.1 |
| Measured TA flux at SWI [mmol TA m$^{-2}$ d$^{-1}$] | A : 14.3 ± 1.6<br>Z : 73.9 ± 20.6<br>Z' : 56.0 ± 17.8 | n.d. | 3.7 ± 0.9 |