# Peer review of "Benthic alkalinity and DIC fluxes in the Rhône River prodelta generated by decoupled aerobic and anaerobic processes"

_Biogeosciences, 2019_

## Referee Comment (RC1) · Anonymous Referee #1 · 28 Mar 2019

General comments: In this study, Rassmann, Eitel et al. investigated benthic alkalinity and DIC release from various sites in the Rhône River delta area. These sites differed in their distance from the river mouth, water depth, and sedimentation rates. The authors measured fluxes to quantify the alkalinity and DIC release, and measured a variety of pore water and sedimentary constituents to investigate the responsible processes. Particular attention was given to the ratio between aerobic and anaerobic organic matter degradation and the role of FeS burial in determining the alkalinity release.

After reading the manuscript, I have somewhat mixed feelings. On the one hand, I appreciate the data set and especially the determination of organic-Fe(III) complexes and FeS nanoparticles, something that is new to me in the context of benthic alkalinity release. On the other hand, after reading I asked myself what the novelty and take home message from this work is and I am not sure if I can properly answer that question. Despite the length of the manuscript (I'd suggest to at least shorten the description of the results and move Fig. 3 to a supplement) I was still left with quite some questions.

What I generally miss in the manuscript is an appreciation of various temporal and spatial scales at which both benthic alkalinity generation and its release can be discussed. For example, if reduced constituents responsible for the alkalinity generation are released to the water column and quickly re-oxidized there, would it still contribute to the $CO_2$ storage capacity over longer time scales? Under which conditions is or is this not valid? Also, can the authors directly compare the alkalinity efflux due to FeS burial and the measured effluxes, given the high sedimentation rate, and that effluxes vary on much shorter timescales, and due to many processes other than FeS burial? And finally, how representative are the measured fluxes (and other data) on e.g. an annual timescale given the high variability in inputs over the year? Could the authors indicate that based on their earlier published work? I appreciate that the authors do not try to temporally upscale their fluxes given the variability, but it does mean that samples from different points in time may plot very differently on Fig 11.

The discussion on identification of major biogeochemical processes remains rather qualitative. I don't think it is possible with the current data to do it differently, but it is a drawback of the manuscript. Figure 10 is a nice summary of concept, but I wonder how valid it is under temporally varying conditions as observed especially at the proximal stations. I think in general more focus could be placed on the factors controlling FeS formation at the various stations and the possible impacts of flooding on these factors. Irrespective of the decision on the manuscript I hope that the following comments will help you further shape it.

Specific comments: L. 14-16: What about pore water iron data? Sulfate and nitrate

concentrations alone don't tell you something about iron reduction. And what about manganese? L. 19-21: This sentence doesn't tell me anything about the underlying mechanisms behind these concurrent observations (which you do explain in the discussion, I noticed later). If these complexes are found, does it mean that sulfide is generally limiting FeS formation? What do you mean by inorganic? Does it refer to iron oxides? L. 54-59: Yes, but this depends on the timescale. The net TA flux due to these processes may not correlate to what you measure as efflux. A diffusive efflux is primarily driven by the gradient at the sediment-water interface and may only reflect processes occurring deeper in the sediment on longer time scales. I would think this is especially relevant in systems with a (periodically) very high sedimentation rate (see further comments below). L. 62-65: Again, this is scale-dependent and only if re-oxidation of reduced constituents does not quickly occur in the water column. L. 65-68: The objectives are formulated in a very qualitative way. I understand this for the second part, but not so much for the first part, which can be formulated more strongly. First, the fluxes were quantified, so you may state that. Second, most sediments are alkalinity sources, but not all sediments release more alkalinity relative to DIC. Isn't that what you're mostly interested in, the possible excess over DIC efflux? L. 91: What about temporal variability in sedimentation rate in the prodelta? L. 106: Can you include a range of how far above the sea floor samples were approximately taken? And did you also sample overlying water from the sediment cores used for the pore water and solid phase analysis? In my experience, the composition of that water can be quite different from a Niskin bottom water sample. L. 108-110: At what temperature are the pH data presented, in situ or 25 degrees? Please add. L. 114: What are the 'main redox species'? Specify. In the results only DIC and TA lander data are presented. L. 119: Which redox chemical species? Specify, this is too vague. L. 124-125: Which method is used for measuring DIC and TA? Same as in section 2.6? If so, refer to it. L. 128 (Eq. 1): Did H remain ∼constant over time or did it decrease over time as a result of the sampling? If so, have you corrected for that? L. 144: How many cores were taken per station? I counted 3: 1 for porosity, 1 for voltammetry, 1 for pore water and

sediment sampling. Correct? L. 158: Why is S0 mentioned as part of total dissolved sulfide? L. 159-160: Does this imply that you can only make a relative comparison between stations of the same cruise? Or can you compare your results with current intensities from other cruises? L. 173-174: What was the sample volume used for alkalinity titrations? L. 177-178: Did you make any replicate measurements? L. 199-200: How was the sediment extracted? Slicing and centrifuging? More details would be appreciated. L. 210-212: I have to say I'm not familiar with this method, but I don't think I understand this. FeS0 consists of two pools, one measured by voltammetry (FeSaq), one not (larger nanoparticles). Spectrophotometry measures both pools. So how can the difference between both measurements be used to quantify both pools? I probably misunderstand, so could you explain it differently? L. 212-217: Has any particular software been used for the saturation calculations? L. 243-246: Indeed this is common practice and I'm perfectly fine with it. So more out of curiosity: do your seacarb or other calculations indeed show that HCO3- constitutes >90% of TA? Using your pH and DIC data, can you say anything about the possible presence of organic alkalinity? L. 247-254: This method applies if there are no other fates of Ca2+ aside from precipitation or dissolution of CaCO3. Can you add a sentence acknowledging this? L. 279-281: Fluxes are highest at station Z, but also most variable (highest s.d.). More interestingly, the two sampling dates show opposing trends in the TA/DIC flux ratio (below 1 for Z, above 1 for Z, although I haven't checked the statistical significance of this), something you don't specifically discuss. Can you place this observation in the larger context of spatial and temporal variability? L. 325-326: What happened at station E that it is not mentioned here? L. 330-333: What's going on at station Z? There is clearly something different between the duplicate cores at 20-25 cm depth, as reflected in the DIC, TA and SO4 data. Can you explain this deviating pattern in the duplicate core, and do you consider them reliable? (also given the extremely high value of $\Omega$ca) From which of both cores are the CH4 data, can I rely on same symbols coming from the same core? Maybe identifying the SMTZ would be easier if only data from a single core are used. L. 375-377: How are these systems different or comparable from the study area? Would

that explain their lower fluxes relative to this work? L. 390-391: This statement can be sharpened. Coupled nitrification-denitrification does not produce TA in a net sense, so any net TA production from denitrification must come from riverine nitrate inputs. Can you use e.g. monitoring data to make an estimate about the importance of this? Also, do you have any information on nitrification rates in the sediment from earlier studies? L. 400-402: True, but as you already discuss later on, if dissimilatory iron reduction is coupled to FeS burial (or re-oxidation of Fe2+), its net efflux on alkalinity is zero. So the process definitely contributes to bulk alkalinity production, but that doesn't necessarily mean it is linked to either alkalinity effluxes or long-term net alkalinity release. If I understand the Burdige and Komada method (L. 406-409) correctly, you already assume this by linking DIC and TA production solely to sulfate consumption. It'd be good to be explicit about this and state which processes are included in this method. Also, I recently came across a paper (https://doi.org/10.1016/j.marchem.2019.03.004) that uses DIC and TA pore water profiles to quantify sulfate reduction rates. I don't know how their methods are applicable to your work but it might be interesting to include it. L. 414-417: Any reason why AOM would be less important at stations A&Z compared to B? At first sight, the SO4 and other profiles do not look too different from each other. L. 424-425: Can you place this pH of 7.2 into context? Why is a minimum of 7.2 not a 'significant lowering'? (L.423) L. 426-442: I spent quite some time looking at equations 12-17 and this method. First, I'd like to see how these equations are derived (e.g. eq. 12 combines eq. 6 and 9, 10). This helps checking them and also the derivation of the ratios. Second, if you look carefully at the equations, you'd see that they are all normalized to SO4. Per mole SO4 the changes in TA and (obviously) SO4 are the same for all six reactions. So the differences in the presented ratios are solely due to the differences in DIC production. Of course this would be different if the equations were presented per mole HCO3 (ratios would be the same, but the changes in TA, DIC and SO4 would be different), but it shows that if you want to link S burial to alkalinity generation (as you do in L. 501-503), the exact pathway of iron sulfide mineral formation doesn't matter. Either way, when comparing measured to theoretical

ratios, the method assumes that there is no other removal pathway of DIC (e.g. siderite formation, to name an option). Can this indeed be excluded? L. 448-452: It'd be nice to read about the possible pathways of organic-Fe(III) complex formation earlier in the manuscript L. 458-460: This statement is less vague than in the abstract, but it still raises questions. At what time scale do these alterations take place? Should I regard 'dominated by sulfate reduction' as the default state of the sediment, only periodically (episodically? seasonally?) replaced by 'dominated by iron oxide reduction' in periods of intense flooding and sediment deposition? Is FeS mineral formation limited by sulfide and if so, does that mean that the flooding periods overprint the default state? L. 478-481: So if I understand this correctly, it means that pIAPs are poor indicators of mineral formation, as they are highest at the site with least burial (station E). Does this mean that microenvironments play an important role in the formation of FeS? I'm also not sure if I understand what you mean to say by the argument of stronger aggregation of FeS (L. 483). If FeS is currently more aggregated, does that mean that FeS formation is not active now (given undersaturation and no FeSaq) but that it had been active in the recent past in a time when the sediments were sulfate-dominated instead of iron-dominated? (this links back to my previous comment). Or does it simply mean that FeS formation just take place in microniches where local conditions are different? L. 493-495: This depends on the fate of the other products, i.e. what happens to the produced S0. But if you assume that the S0 will also be buried (or converted to FeS, which are both more likely options than reoxidation), the alkalinity release will always be 2 moles per mole S burial. L. 501-504: First, why don't you compare the AVS burial flux with the measured alkalinity flux of station A only, instead of combining A and Z? Second, a point that I am just realizing: with this very high burial rate, it'll take a long time before alkalinity produced in the sediment is diffused out. You'd expect that its transport is dominated by advection, not diffusion. So that would mean an even stronger decoupling between net TA generation in the sediment and measured effluxes. Or is there bioturbation that impacts the benthic release? L. 526-529: I agree that microniches can be important, but do your Ca2+ porewater profiles give any indication of CaCO3

dissolution at the top of the sediment? L. 534-539: Is FeS the dominant form of solid-phase S in the sediment, or is pyrite also present in substantial amounts? L. 540-541: On what timescale do these processes take place? Under steady-state conditions I understand this figure, but given the highly variable sedimentation rate at especially the proximal sites, does it still apply under these dynamic conditions? L. 545-547: but is Fe or S generally limiting FeS formation at the proximal sites? L. 556-559: so at station E Fe is limiting FeS formation, what about the prodelta sites? L. 574-576: but if the FeS burial sink is permanent, it definitely impacts water-column TA and carbonate system dynamics on the long term, as you also indicate on L.576-578 and L.601-604. I think this statement unnecessarily weakens the relevance of your manuscript. Fig. 9: What if TA data were used for this calculation? I agree though that using DIC is wiser given the possible presence of organic alkalinity.

Technical corrections: L. 3-6: This sentence is too complex. OC respiration in sediment or water column? I'd suggest to rewrite and / or split it in two sentences. L. 31-33: Ambiguous sentence. Does "of which about half is buried" refer to total oceanic POC or the 40% that is buried in shelf regions? L. 56: typo in 'anaerobic' L. 77: replace 'sediments' with 'suspended matter' or 'particles' L. 98: "These". All sediments or those in the proximal region only? L. 127-130 and various other sections in the manuscript (basically everywhere where equations are presented): add units to the variables you discuss here (i.e. $F_i$, $H$, $C_i$, etc). L. 235-239: This sentence is too complex. Please split into two or rephrase. L. 242: Add scale and temperature to pH. L. 244: For which salinity are these numbers valid? Result section: may be shortened L. 288: change to "the absolute value of the DOU fluxes" or equivalent as they have opposing signs. L. 306: 'station' instead of 'stations' L. 522: seacarb is written without capitals. Figures: Add units to the captions. Fig. 1: Add the depth to the last (lowest) line of the bathymetry. Fig. 2: Use different lines (e.g. solid and dotted) for $O_2$ and pH. Printed in black & white the figure is currently very difficult to read. Fig. 3: Could be moved to an online supplement. Add what the difference between the red and black symbols means. The error bars complicate reading of the symbols a bit, but I appreciate that

they're in. Fig. 4: I'd only plot the error bars outwards, this makes the bar plot better readable and the error bar is not visible in the black bars anyway. Fig. 5: Make it clear which measurements are from the duplicate core by using the same symbols for DIC, TA and SO4, and make them clearly different from the main core data. Fig. 6: Add a (dashed) line at $\Omega$ca=1. Also, the DIC data are poorly visible as they are mostly hidden behind the TA data. I'd leave it like this only if the point you're trying to make is that they are so similar. Fig. 7: I'd suggest not splitting the axes into two domains, given the small jump on both axes it complicates more than that it helps reading. Fig. 11: "as a function of water depth". Add the source of the North Sea data (Brenner et al.?). Hu & Cai (2013) is not in the list of references. Table 1: Be consistent with sulfate and SO4 in the caption. I think that in equation (5) it should read -1/5 H+ (instead of -2/5). Show how you derived equations (12) to (17), see earlier comment. Table 2: Add the depth interval over which mean porosity was calculated.

---

## Referee Comment (RC2) · Anonymous Referee #2 · 4 Apr 2019

I read carefully the manuscript of Rassmann, Eitel, and collaborators and I recommend it for publication after revision. This paper presents an in-depth analysis of benthic biogeochemical processes and DIC/TA release in different stations in the Rhône River Delta area. I particularly like the multitude of measurements applied to improve understanding of processes driving anaerobic formation of TA and benthic fluxes; particularly, combining in situ incubations, potentiometric and voltammetric micro-profiles with more conventional pore water and sediment analyses. This combination of methods is rarely encountered in these types of investigations which often focus primarily on submillimetric processes at the SWI. Furthermore, the amount of data collected is significant, and has to be published, definitively.

[Figure]

My main concern is related to the overall perspective of the research. The authors do not convey very clearly the scientific importance of their work. For instance, in the introduction they mention that the objective of the paper was to " investigate if sediments from deltaic regions exposed to large riverine inputs of carbon and minerals represent an alkalinity source to the bottom waters and identify the biogeochemical processes responsible for the net production of alkalinity in these sediments. ". There are several studies that have done this in coastal area as well, thus, they should portray how this work is different. And I would say that the combination of methods is unique. After reading the paper several time, I am still not sure to understand the take home message of this study. What is really new? I strongly appreciated the effort to present this very large set of data that includes Fe(III)-Lorg, sulfide species, and methane in addition to major (or classical) diagenetic species. However, I often lost myself in detail that ultimately brings little. For examples, the section on the role of nitrification / denitrification and the section on IAPs are long but their conclusions are not very relevant for the rest of discussion. Overall the discussion should be shortened.

My second concern is on the role of terrigenous organic matter in this type of sediment. The authors characterized the study site as "deltaic sediments exposed to large riverine inputs of inorganic and organic material". In these sediments, coarse particulate organic matter is deposited during flood events and supports the establishment of sulfidic conditions and the precipitation of Fe-S phases (François et al., 2014; Fagervold et al., 2014; Rassmann et al., 2016). As POM, CPOM is probably a source of DOM and organic alkalinity in pore water. What is the role of organic alkalinity on TA in these sediments? Did you calculate the theoretical TA based on DIC and pH? The production of organic alkalinity should be discussed and the contribution to TA should be estimated (it's rare to have enough data to do it). In addition, the accumulation of refractory organic carbon in sediments appears intimately associated with the sequestering of iron and sulfides in micro-environments (see the works of François). When the authors discuss about the aggregation of FeS, do they talk about microenvironments? I think the role of terrigenous organic matter on these biogeochemical processes should

be clarified.

My third concern is on the time scale of the explored biogeochemical processes. The deltaic sediments cannot be considered at steady state, specifically in the proximal stations. However, the discussion is based on a steady state view of the different reactions. So, what is the impact of floods on the oxygen demand and DIC/TA release? Are these fluxes constant over the year, with no seasonal variations? "[...] their presence in zones of sulfate reduction suggest these sediments are highly dynamic with periods of intense sulfate reduction alternating with periods during which sulfate reduction is repressed and replaced by microbial iron reduction (line 458-460). Does this sentence mean that there are two different conditions depending of the flood conditions or seasons? What are the consequences on FeS precipitation and on TA release? This sentence is too vague and raises questions on the temporal representativeness of the data (episodic event? Seasonal variations?). I have the same questions on the spatial representativeness of the data. How do the authors explain the difference between the two replicates Z and Z'? Then, I encourage the authors to discuss about the spatio-temporal representativeness of observations.

My last issue is on the role of bioturbation. I think about bioturbation when I looked at the figure 10. According to the frequency of flood events and to the accumulation rates at the proximal stations, the diffusive transport of the anaerobically-produced alkalinity in the flood deposit to the SWI, takes time no? (see the work of Anschutz and collaborators in natural turbidites (Anschutz et al., 2002; Chaillou et al., 2006) and in experimental turbidites (Chaillou et al., 2007)). Bioturbation and biodiffusion could be an efficient mechanism to transport anaerobically-produced metabolites, as TA from the anaerobic zone to the surface. Did the authors measure the bioturbation coefficients in the incubations? Did they consider the macrofauna in the studied sediment? What about the difference between total fluxes and diffusive fluxes of DO, TA and DIC? Are they similar (same magnitude)?

The authors are kindly asked to see the attached annotated PDF with my suggestions

and points. Finally, I encourage the author to shorten the result and discussion sections before to resubmit the paper. This study has definitively a strong potential and must be published.

Please also note the supplement to this comment:
https://www.biogeosciences-discuss.net/bg-2019-32/bg-2019-32-RC2-supplement.pdf

**Supplement:**

[revised manuscript text omitted]

---

## Author Comment (AC1) · 24 May 2019

(Please see pdf attached as a supplement for a formatted version with Figures)

Response to Anonymous Referee #1

C: General comments: In this study, Rassmann, Eitel et al. investigated benthic alkalinity and DIC release from various sites in the Rhône River delta area. These sites differed in their distance from the river mouth, water depth, and sedimentation rates. The authors measured fluxes to quantify the alkalinity and DIC release, and measured a variety of pore water and sedimentary constituents to investigate the responsible

processes. Particular attention was given to the ratio between aerobic and anaerobic organic matter degradation and the role of FeS burial in determining the alkalinity release. After reading the manuscript, I have somewhat mixed feelings. On the one hand, I appreciate the data set and especially the determination of organic-Fe(III) complexes and FeS nanoparticles, something that is new to me in the context of benthic alkalinity release. On the other hand, after reading I asked myself what the novelty and take home message from this work is and I am not sure if I can properly answer that question.

R: We appreciate the reviewer recognizes that the organic-Fe(III) complexes and FeS nanoparticles are useful in the interpretation of this complex data set. In turn, we are disappointed by the comment that the reviewer does not understand the novelty of this work. This work demonstrates that the burial of reduced iron and sulfur in the sediment prevents reoxidation of reduced metabolites at the sediment-water interface and therefore contributes to the alkalinity flux to the overlying waters. Although these concepts are not novel, to our knowledge, this manuscript provides for the first time in situ benthic alkalinity flux data and simultaneous biogeochemical evidence from pore water and sediment profiles that substantiate this argument. The abstract, discussion, and conclusions will be modified to emphasize the fact that this is the first study demonstrating the link between measured benthic alkalinity flux and biogeochemical processes, namely FeS precipitation and burial, responsible for this alkalinity flux.

C: Despite the length of the manuscript (I'd suggest to at least shorten the description of the results and move Fig. 3 to a supplement) I was still left with quite some questions. What I generally miss in the manuscript is an appreciation of various temporal and spatial scales at which both benthic alkalinity generation and its release can be discussed. For example, if reduced constituents responsible for the alkalinity generation are released to the water column and quickly re-oxidized there, would it still contribute to the $CO_2$ storage capacity over longer time scales? Under which conditions is or is this not valid? Also, can the authors directly compare the alkalinity efflux due to FeS

burial and the measured effluxes, given the high sedimentation rate, and that effluxes vary on much shorter timescales, and due to many processes other than FeS burial? And finally, how representative are the measured fluxes (and other data) on e.g. an annual timescale given the high variability in inputs over the year? Could the authors indicate that based on their earlier published work?

R: We agree that the manuscript could be shortened, and we will move Fig. 3 (benthic chamber TA and DIC) and Fig. 8 (pIAP) and associated paragraphs presenting the methods and describing or discussing these data to a supplementary material section to shorten the manuscript. The question of temporal and spatial scales is crucial in these dynamic environments and we tried to introduce them in the manuscript, although we probably did not include enough material dealing with this topic in the background (1. Introduction) and field site description (2.1: The Rhône River delta) sections. These two sections will be updated with additional sentences on the temporal and spatial scales as explained below.

Concerning the question on the re-oxidation of reduced components in the water column, re-oxidation of reduced metabolites has the same net effect as re-oxidation in the sediment, i.e. consumption of alkalinity by protons produced during oxidation (see equations 3 and 4 of Table 1). Thus, re-oxidation of reduced metabolites does not contribute to the $CO_2$ storage capacity over long time scales. This is the reason why the burial of reduced components (FeS and potentially $FeS_2$) in the deep sediment layers presented in this manuscript is so important: Burial of FeS and $FeS_2$ prevents re-oxidation in the sediment and in the water column and creates a net flux of alkalinity to the water column. Although we feel that this topic was well covered in the original manuscript (see lines 21-25 of the abstract, lines 51-55 of the introduction, lines 491-493 and 547-550 of the discussion), we will clarify these points in these sections and improve Figure 10 to clearly illustrate the importance of the lack of re-oxidation near the sediment-water interface.

Concerning the spatial scale, we already described in the original manuscript the different zones of the Rhône delta from the proximal zone to the continental shelf (lines 82-86) and their characteristics (sedimentation rates, depth, organic carbon content; see also Table 2) in the original manuscript. Something surely missing in this paragraph is an appreciation of the spatial heterogeneity at the local scale. As the main deposition of sediment occurs during floods, sediment layers are heterogeneous at the meter scale, and differences in pore water profiles can be detected at that scale (e.g. DIC and TA profiles on Fig. 6 for station A and Z taken from two different cores at the same stations). These points will be highlighted in the revised manuscript. However, even with this local variability (at the station scale) taken into consideration, the difference between the proximal zone stations and stations in the prodelta or the continental shelf are still obvious as highlighted in the discussion of the original manuscript (see sections 4.5, 4.6, 4.7, and 4.8) (see also Rassmann et al., 2016). Concerning the temporal variations, the Rhône prodelta system receives large inputs of particulate material in late fall and early winter during major river floods (as stated line 87-90), including terrestrial organic matter which is mineralized in the spring and summer. Although data are scarce and new programs are underway to address temporal variations, there seems to be a progressive buildup of metabolites in the sediment pore waters and progressive disappearance of fronts during winter and spring (Rassmann, unpublished data). One good example is found in Pastor et al. (2018, CSR) which describes an atypical flood in the spring and the evolution of the pore water profiles over 6 months from May to December (see DIC profiles of the prodelta station A in Fig. 4 of Pastor et al., 2018). In the general case of the fall floods, the most intense organic matter mineralization rates are reached in spring and summer in the proximal zone, where organic matter accumulation is highest, as presented in Rassmann et al (2016) for June 2014 where sulfate reduction drives the sulfate concentrations to almost 0, producing 30-40 mM of DIC and alkalinity with 500-800 $\mu$M of dissolved iron and no sulfide in the pore waters (see also Pastor et al., 2011 for a similar situation in April 2007). In addition, this pattern was found over several sampling campaigns in April 2013 (Dumoulin et al., 2018), 2014 (Rassmann et al., 2016), 2015 (this paper), and 2018 (unpublished results). Altogether, the pore water data collected over the years in the Rhône prodelta system are consistent and indicate that biogeochemical processes in the critical proximal zone reaches a reproducible state on a yearly basis due to the regularity of flood deposition in late fall and maturation of the system in spring and summer. This reproducibility of the spring-summer conditions applies probably to the observed fluxes as well.

Concerning the comparison between benthic alkalinity and FeS burial fluxes, fluxes of alkalinity and DIC certainly vary over time with the progressive buildup of alkalinity and DIC in pore waters, as sulfate reduction, iron oxide reduction, and FeS precipitation proceed during the spring and summer seasons. The alkalinity flux, however, is linked to the net precipitation of FeS, which is the difference between precipitation and re-oxidation due to transport in the oxic zone. Burial occurs when a new flood layer is deposited (in late fall) which traps the FeS produced during the year below a new sediment layer of 10-30 cm ensuring its preservation. This yearly preservation of FeS ensures the concomitant benthic alkalinity flux to represent a net flux to the bottom waters which is not affected by FeS oxidation, contrarily to sediments exposed to low sedimentation rates where FeS can be entrained by bioturbation to the oxygenated layer and be re-oxidized (therefore consuming alkalinity).

Concerning the "many processes other than FeS burial" which influence the benthic alkalinity flux, we refer to previous studies which demonstrated only two anaerobic processes (iron sulfide burial and N2 loss by denitrification) contribute to a net release of alkalinity from the sediment through the net loss of reduced species by either of these processes (see for example Hu and Cai, 2011 in GBC). All other internal cycling processes do not represent net sources of alkalinity as highlighted in the discussion already (see lines 419-442 and Table 1). We show in the manuscript that denitrification is a minor source of alkalinity, if any (discussed in section 4.2). The other potential source of alkalinity in sediment is carbonate dissolution which we showed to be absent or minimal due to large oversaturation of the pore waters with respect to calcite (Rassmann et al., 2016 and Figure 9 of this manuscript) supported in this study by the

decreasing Ca2+ concentrations in the profiles (Figure 6), certainly indicating CaCO3 precipitation.

Concerning the "representativity of the measured fluxes", it is hard to conclude as these measurements represented the first attempt to measure either in situ or ex situ benthic alkalinity fluxes in the Rhône River prodelta. During a more recent cruise in 2018, sediment cores were incubated to estimate DIC and alkalinity fluxes across the sediment-water interface. The (unpublished) results (see Figure AC1 below) display the same pattern as observed in situ in 2015 (this study) with large DIC and TA fluxes in the proximal zone (A, Z, AZ) and decreasing fluxes offshore (AK, K and C, E on the shelf). Insert Figure AC1

To clarify these points in the manuscript, we will add a few sentences in the introduction on the effect of seasonal patterns on the biogeochemistry of river-dominated sediments and a paragraph in section 2.1 containing information on the temporal variation in depositions in the Rhône River prodelta from previous studies. Finally, the discussion will highlight the features described above that argue for the decoupling between aerobic and anaerobic processes and the link between FeS burial fluxes and benthic-pelagic alkalinity fluxes.

C: I appreciate that the authors do not try to temporally upscale their fluxes given the variability, but it does mean that samples from different points in time may plot very differently on Fig 11.

R: As no data are available during the fall and winter and evidence suggests that the state of the system is different during these seasons due to flood-related increase in deposition, we prefer not to extrapolate the fluxes over this period. As the aim of the paper was to discuss the link between alkalinity fluxes and sediment diagenetic processes, however, the lack of extrapolation over time does not weaken the message of our paper.

C: The discussion on identification of major biogeochemical processes remains rather

qualitative. I don't think it is possible with the current data to do it differently, but it is a drawback of the manuscript. Figure 10 is a nice summary of concept, but I wonder how valid it is under temporally varying conditions as observed especially at the proximal stations. I think in general more focus could be placed on the factors controlling FeS formation at the various stations and the possible impacts of flooding on these factors.

R: As the reviewer indicates, the identification of the major biogeochemical processes remains qualitative, though corroborative evidence between various data sets helps draw a rather clear biogeochemical picture of the processes taking place in these sediments. We discussed the temporal variability already in the response to the previous point by the reviewer. We think that the concept of Figure 10 is valid when integrating over an entire year but certainly not over each individual season: Indeed, following late fall deposition of flood sediment layers, DIC and alkalinity build up in pore waters during the winter and FeS accumulates in the solid phase as sulfate reduction and iron oxide reduction proceed. In spring and summer, sampling conducted over 5 different years reveals that the diagenetic system is in a similar state on an pluriannual basis (see above): FeS is buried only at the end of the fall when a new flood layer is deposited closing the conceptual diagram of Figure 10. As mentioned above, the temporal variability will be included in the revised manuscript and Figure 10 will be improved to clarify these points.

C: Irrespective of the decision on the manuscript I hope that the following comments will help you further shape it.

R: We appreciate the time spent by the reviewer to carefully evaluate our work and the thoughtful comments that will certainly help us improve our manuscript.

Specific comments:

L. 14-16: What about pore water iron data? Sulfate and nitrate concentrations alone don't tell you something about iron reduction. And what about manganese?

The reviewer is right. Low nitrate concentration does not tell about iron hydroxide reduction. We will add large dissolved iron concentration between "Low nitrate" and "strong pore water sulfate gradient". However, low nitrate concentrations suggest that denitrification is not a major process in these sediments. Furthermore, a model study comparing the proportion of different mineralization pathways concluded that denitrification is only a minor process in this area (Pastor et al., 2011). In turn, sulfate data demonstrates that dissimilatory sulfate reduction is a major respiratory pathway in these sediments (See Rassmann et al., 2016). Pore water manganese data is also presented later in the paper, however, as concentrations of dissolved Mn(II) were low and manganese reduction was not expected to contribute greatly to the biogeochemistry of these sediments we did not feel it was necessary to mention manganese in the abstract.

L. 19-21: This sentence doesn't tell me anything about the underlying mechanisms behind these concurrent observations (which you do explain in the discussion, I noticed later). If these complexes are found, does it mean that sulfide is generally limiting FeS formation?

We agree that this sentence is misleading and needs to be clarified in the abstract. The detection of organic-Fe(III) compounds, which are indicative of dissimilatory iron reduction and are rapidly reduced by sulfide, demonstrates that the system is dynamic as we would not expect to observe these Fe(III) species in zones dominated by sulfate reduction.

C: What do you mean by inorganic? Does it refer to iron oxides?

Yes. This will be clarified in the revised abstract.

L. 54-59: Yes, but this depends on the timescale. The net TA flux due to these processes may not correlate to what you measure as efflux. A diffusive efflux is primarily driven by the gradient at the sediment-water interface and may only reflect processes occurring deeper in the sediment on longer time scales. I would think this is especially relevant in systems with a (periodically) very high sedimentation rate (see further comments below).

We agree with the reviewer that the efflux is driven by the concentration gradient at the sediment-water interface (SWI), but this gradient is controlled by biogeochemical reactions in the sediment. Yet, the connection between the two can be made if chemical species can migrate over the distance between the biogeochemically active zones and the SWI over a given time. It is possible to calculate the minimal distance (d) that can be traveled by chemical species over time t by calculating the diffusion distance (d = sqrt(2*DiffCoeff*t) where DiffCoeff is the diffusion coefficient of the chemical species). For a period of 6 months (between fall and spring) and with the DiffCoeff of HCO3- (D = 11.8 10-6 cm2/s at 25°C, around 6 10-6 cm2/s at 20°C in the sediment), the diffusion distance is around 15 cm. This distance is minimal as bioturbation and bioirrigation will increase transport and thus increase the connection distance. We can thus consider that most probably 20 cm are connected to the SWI and that biogeochemical processes over that depth interval can shape fluxes at the SWI over a 6 months period (the late spring, summer and early fall which certainly corresponds to the "longer time period" that the reviewer quote). This information will be added to section 4.6 to clarify this point.

L. 62-65: Again, this is scale-dependent and only if re-oxidation of reduced constituents does not quickly occur in the water column.

We disagree with this comment. If reduced constituents are buried in sediments, they are not fluxing across the sediment water interface and thus not re-oxidized in the water column (see our initial response above).

L. 65-68: The objectives are formulated in a very qualitative way. I understand this for the second part, but not so much for the first part, which can be formulated more strongly. First, the fluxes were quantified, so you may state that.

Agreed, this will be revised by replacing 'investigate' by 'determine' in the revised
manuscript.

C: Second, most sediments are alkalinity sources, but not all sediments release more alkalinity relative to DIC. Isn't that what you're mostly interested in, the possible excess over DIC efflux?

This is incorrect, sediments are generally weak sources of alkalinity because only denitrification based on external NO3- input and FeS(or FeS2) burial represent net sources of alkalinity (which is not common). Even sediments with high rates of aerobic and anaerobic respiration followed by re-oxidation of all reduced species produced during anaerobic respiration would not act as a net alkalinity sources although the large DIC quantities produced by these processes diffuse out of the sediment. Thus, measurement of multiple species was conducted to determine the extent and mechanism of alkalinity production in these sediments and its relation to diffusion across the sediment-water interface, as described in the text. This point will be emphasized in the revised manuscript.

L. 91: What about temporal variability in sedimentation rate in the prodelta?

The temporal variability in sediment rate in the Rhône River prodelta is largely unknown, except for the paper by Cathalot et al (2010, BG) which shows deposition of an abnormal flood in June 2008. It is known that the majority of the sediments in the vicinity of the river mouth is deposited during flood events which occur mostly in fall and early winter. Thus, the sedimentation rates vary due to episodic events. This information will be provided in the revised manuscript.

L. 106: Can you include a range of how far above the sea floor samples were approximately taken?

We used the depth estimation from the winch for sampling and checked the real depth with a mounted underwater depth gauge. Bottom water samples were taken within 1-2 m above the seafloor. This information will be provided in the revised manuscript.

C: And did you also sample overlying water from the sediment cores used for the pore water and solid phase analysis? In my experience, the composition of that water can be quite different from a Niskin bottom water sample.

Yes, overlying water from the sediment cores was measured and in good agreement with the bottom water DIC and TA concentrations. This information will be mentioned in the revised manuscript.

L. 108-110: At what temperature are the pH data presented, in situ or 25 degrees? Please add.

The pH samples were measured at 25°C and the pH values recalculated to in situ temperature, salinity, and pressure using CO2sys. The pH data is presented at in situ temperature and pressure. This information will be provided in the revised Method section.

L. 114: What are the 'main redox species'? Specify. In the results only DIC and TA lander data are presented.

This part of the sentence will be removed from the paper as the results from the in situ voltammetric sensors in the lander chamber (iron, manganese, sulfide) are not used in this manuscript.

L.119: Which redox chemical species? Specify, this is too vague.

This part of the sentence will be removed from the paper as the results from the in situ profiles are not used in this manuscript.

L. 124-125: Which method is used for measuring DIC and TA? Same as in section 2.6? If so, refer to it.

Yes, DIC and TA measurements were conducted using the same method as described later in the paper. As this section just describes the benthic chamber deployments we prefer to keep the analysis methods in section 2.6, and we will refer to it in the benthic

chamber section.

L.128 (Eq. 1): Did H remain constant over time or did it decrease over time as a result of the sampling? If so, have you corrected for that?

The volume (and thus the height) of the chamber remained constant during the deployment and the syringe volume was compensated by an equal amount of bottom water. This was accounted for in the calculations.

L. 144: How many cores were taken per station? I counted 3: 1 for porosity, 1 for voltammetry, 1 for pore water and sediment sampling. Correct?

Six cores were taken at each station: 1 short core (∼30 cm length) for pore water extraction under N2, one long core (∼50 cm length) for pore water extraction in lab, one core for porosity measurement, one core for methane concentrations, one core for voltammetric profiles, and one core for archives.

L. 158: Why is S0 mentioned as part of total dissolved sulfide?

S(0), including S(0) from polysulfides, may contribute to the total sulfide voltammetric peak (Taillefert et al., 2000) as the electrochemical reaction involves simultaneous oxidation of Hg(0) and reduction of S(0) at the electrode surface to form HgS at the same potential as that of the oxidation of Hg(0) in the presence of H2S. As a result, S(0), polysulfides, H2S, and HS- cannot easily be distinguished and are typically reported as total dissolved sulfide (2S) when measured electrochemically. This information will be provided in the revised manuscript.

L. 159-160: Does this imply that you can only make a relative comparison between stations of the same cruise? Or can you compare your results with current intensities from other cruises?

Normalized intensities can be compared to any data sets collected at any time because they are only normalized to the sensitivity of a particular electrode at the time it was used.

L. 173-174: What was the sample volume used for alkalinity titrations?

The sample volume used for alkalinity titrations was 3-6 ml of pore water, according to available quantity. This information will be provided in the revised manuscript.

L. 177-178: Did you make any replicate measurements?

Yes, triplicate measurements were made for DIC and either duplicate or triplicate measurements were made for TA, depending on the total pore water volume available. Replication adopted for the chemical analyses will be provided in section 2.6 of the revised manuscript.

L. 199-200: How was the sediment extracted? Slicing and centrifuging? More details would be appreciated.

The sediment was collected using Rhizons as mentioned in the original manuscript (lines 167-169).

L. 210-212: I have to say I'm not familiar with this method, but I don't think I understand this.

FeS0 consists of two pools, one measured by voltammetry (FeSaq), one not (larger nanoparticles). Spectrophotometry measures both pools. So how can the difference between both measurements be used to quantify both pools? I probably misunderstand, so could you explain it differently? FeS0 represents the difference between two measurements of Fe(II) by spectrophotometry and by electrochemistry. The Fe(II) measured spectrophotometrically in extracted pore waters includes both Fe(II), FeS(aq), and larger FeS nanoparticles which may pass through the rhizon filters used to extract pore waters but are not measured by electrochemistry due to their limited diffusion to the electrode. In turn, the Fe(II) measured electrochemically includes only Fe(II) and small FeS(aq) molecular clusters that are smaller than 5 nm and thus voltammetrically measurable. As a result, the difference between these two measurements represents the FeS nanoparticules (named FeS0) that are small enough to pass through the rhizon

filters but too large to diffuse to the electrode. We feel this information is well presented in lines 204-209 and Equation 2. Unfortunately, we cannot directly compare FeS(aq) and FeS0 concentrations as FeS(aq) determined by voltammetry is not quantifiable but is reported in normalized current intensities (as described in lines 161-162). This last point will be highlighted in the revised manuscript.

L. 212-217: Has any particular software been used for the saturation calculations?

IAP can be calculated based on Eq. 3 with the parameters described in lines 217-220. This calculation was done in Microsoft excel as ionic strength and activity coefficients from conventional seawater were used in these calculations. The equilibrium constant was recalculated at the ionic strength of seawater, the measured $Fe^{2+}$ concentrations were used as 'free' available $Fe^{2+}$, as $Fe^{2+}$ does not form strong complexes, and 2S concentrations were used to calculate the speciation of sulfide species (assuming no elemental sulfur or polysulfide were present in the pore waters). These details will be provided in the supplemental material of the revised manuscript.

L. 243-246: Indeed this is common practice and I'm perfectly fine with it. So more out of curiosity: do your seacarb or other calculations indeed show that $HCO_3^-$ constitutes >90% of TA? Using your pH and DIC data, can you say anything about the possible presence of organic alkalinity?

Yes, the calculations using Seacarb show that $HCO_3^-$ concentrations in the pore water are always > 90%, except for one point at 30 cm depth at station K were $HCO_3^-$ represents "only" 89 % (see new figure below-AC2).

Insert Fig. AC2

To estimate the amount of organic alkalinity, we could theoretically calculate carbonate alkalinity using DIC concentrations and pH and substract the result from the measured value of total alkalinity. The difference should equal the organic part of alkalinity. In the bottom water samples, we measured pH, TA and DIC. The calculations give a concentration of organic alkalinity between 8 and 28 $\mu$mol/L (< 1% of TA) in the bottom water samples (sampled with the Niskin bottle) without any particular spatial trend. In sediments, the issue is more complicated. pH microprofiles were measured in situ down to 20 mm only, with a vertical resolution of 200 $\mu$m. DIC and TA were measured in extracted pore water samples, with a vertical resolution of 2 cm. Furthermore, these extracted pore waters represent an average value of several mm around the rhizons. This means, that we only have an overdetermined carbonate system down to two cm sediment depth with one single value for DIC and TA and a full pH profile with around 100 values ranging from 8.1 to 7.2. We could still use an averaged pH value together with the DIC value in order to calculate the fraction of carbonate alkalinity and non-carbonate alkalinity in the power waters, but as these calculations are sensitive to pH, the approach is questionable. A first guess using an averaged pH value around 0, 1, and 2 cm depth lead to concentrations of organic alkalinity not exceeding 60 $\mu$mol/L. These low organic alkalinity concentrations in surface sediments may also prevail in deep sediments as the DIC/TA ratios are close to 1 (Figure 6), which is not the case when large quantities of organic alkalinity accumulate in sediments (Lukawaska-Matuszewska et al. 2018 Marine Chemistry, doi: 10.1016/j.marchem.2018.01.012). This is definitely an issue that could be resolved in further investigations by measuring pH profiles over longer depths in the sediment. As this question by the reviewer was merely out of curiosity and as the lack of data on organic alkalinity deep in the sediment does not change the main conclusions of the manuscript, we will not address this comment further in the revised manuscript.

L. 247-254: This method applies if there are no other fates of Ca2+ aside from precipitation or dissolution of CaCO3. Can you add a sentence acknowledging this?

In sediments, calcium is essentially related to calcite (pure or magnesian). Calcium could as well be consumed by the formation of Ca(Mg)CO3. The Mg fraction can get as high as 50 % (dolomite) but is generally around 20% in magnesian calcites. If this is the case, the Ca/DIC and Ca/TA ratios would be lower than for CaCO3. For

simplicity and having no information about the Mg content of the reprecipitating mineral phase, only CaCO3 is considered. We have also to keep in mind that the correction itself affects the ratios by less than 10 %. A sentence will be added in the manuscript acknowledging that most Ca2+ is trapped in Ca-rich carbonate.

L. 279-281: Fluxes are highest at station Z, but also most variable (highest s.d.). More interestingly, the two sampling dates show opposing trends in the TA/DIC flux ratio (below 1 for Z, above 1 for Z, although I haven't checked the statistical significance of this), something you don't specifically discuss. Can you place this observation in the larger context of spatial and temporal variability?

It is difficult to account for statistical significance from just two flux calculations. Given the error bars for the two flux measurements, one cannot tell if the TA/DIC ratio is different for the two flux measurements at station Z and Z'. Spatial heterogeneity (at the scale of several meters) between the deployment Z and Z' is certainly present as shown by the significant variation of fluxes. This is due to the deposition conditions during floods which may vary locally (both in quantity and quality, see response to an earlier comment by the reviewer). Yet the large difference and the significant alkalinity fluxes observed at this station compared to the shelf site are indicative of specific processes in the proximal zone. Sediment core incubations conducted in 2018 (Figure AC1 shown above) seem to point in the direction of TA/DIC flux below 1. TA/DIC flux superior to 1 could mean, that a high fraction of TA is organic or that calcium carbonates dissolve at the surface. But as reported in this article and in Rassmann et al. (2016), bottom waters and pore waters are oversaturated with respect to calcium carbonates and calcium concentration decreases with depth. We can therefore only stress that the fluxes have been measured in situ with a benthic chamber enclosing 30x30 cm of sediment surface and should therefore be considered a robust measure of the benthic fluxes during the sampling period. These points will be highlighted in the revised discussion.

L. 325-326: What happened at station E that it is not mentioned here?

Due to small variations in sulfate concentrations with depth at station E the variations with depth are of the same order of magnitude as the measuring uncertainties. As a result, sulfate to DIC or alkalinity ratios were not included in Table 3. The text will be modified to clarify this point in the revised manuscript.

L. 330-333: What's going on at station Z? There is clearly something different between the duplicate cores at 20-25 cm depth, as reflected in the DIC, TA and SO4 data. Can you explain this deviating pattern in the duplicate core, and do you consider them reliable? (also given the extremely high value of $\Omega$ca) From which of both cores are the CH4 data, can I rely on same symbols coming from the same core?

The values measured are reliable. The system is heterogenous and intra-station differences can be quite important. Yet the two profiles (short and long cores) for station A and Z match at some point at depth. Furthermore, the profiles measured at station A and Z are comparable with DIC, TA, and SO4$^{2-}$ profiles measured in other campaigns (e.g.: Rassmann et al., 2016; Pozzato et al., 2017). Despite the different shapes of the individual profiles, the DIC/TA and DIC/SO4$^{2-}$ ratios (and if we would look at them also the DIC/Ca ratios) are always the same. The CH4 core was a different core and the symbols do not indicate that CH4 was measured on the same core. This information will be provided in the revised manuscript.

C: Maybe identifying the SMTZ would be easier if only data from a single core are used.

The concentrations of dissolved methane were measured in another core than the concentrations of DIC, TA, and SO4$^{2-}$ for practical reasons, as it more efficient to use horizontal mini-cores at each depth in the sediment core to sample this insoluble gas (see methods). This information will be provided in the revised manuscript.

L. 375-377: How are these systems different or comparable from the study area? Would hat explain their lower fluxes relative to this work?

[Figure]

A first point to stress here, is that measurements of in situ TA fluxes across the sediment water interface in river dominated margins are still rare in the literature. In terms of freshwater discharge, the Rhône river (1700 m3/s) transports much more water than the Guadalquivir (160 m3/s), but is comparable with the Po (1500 m3/s). The Danube (8200 m3/s), the Fly river (6000-7000 m3/s ) and the Mississippi (15,000-18,000 m3/s) transport even more water and particles. (Friedl et al., 1998; Hammond et al., 1999; Aller et al., 2008; Ferron et al., 2009; Lehrter et al., 2012). The Guadalquivir, the Po and the Rhône dominated margins are microtidal, mediterranean systems and bottom waters display comparable oxygen concentrations, temperatures and salinities. In contrast, the Fly river bottom waters are warmer and less saline and the Mississippi bottom waters experience seasonal hypoxia. The fluxes we compare have all been measured in water depth between 10-150 m. The sediments are either of cohesive nature or contain sandy layers. All systems are characterized by organic rich sediments and high respiration rates and are within the same range of Alkalinity fluxes. The only exception is the Po River delta for which the stations were located in the shelf zone (rather than the prodelta) and are closer in nature and Alkalinity flux to station E from this study (1-5 mmol m-2 d-1). A sentence will be added in the text to explain this point. The position of the stations in different studies cited here are partly further on the shelf and reflect the low TA fluxes as measured on the continental shelf near the Rhone River delta. This is mostly the case in A sentence will be added in the text to explain that.

L. 390-391: This statement can be sharpened. Coupled nitrification-denitrification does not produce TA in a net sense, so any net TA production from denitrification must come from riverine nitrate inputs. Can you use e.g. monitoring data to make an estimate about the importance of this?

In microtidal systems such that Rhône River delta, haline stratification is strong, such that riverine nitrate is confined in the surface river plume (e.g., 0.5 to 1 m, Many et al., 2018 PiO) where it is either diluted or consumed by phytoplankton growth. The marine origin of the bottom water is assessed by their salinity (37.5-38.0; Table 2) and their

low nitrate concentrations (Bonin et al., 2002, Wat. Res. 36, 722-732). Furthermore, nitrate profiles presented in Pastor et al. (2011) all show an increase of concentration in the porewater indicating that the nitrate flux is directed from the sediment to the water column. Therefore, riverine nitrate does probably not influence denitrification observed in the proximal zone sediment as of the bottom waters. This explanation will be provided in the revised manuscript.

C: Also,do you have any information on nitrification rates in the sediment from earlier studies?

Unfortunately, no nitrification rates are available in the sediments for this area. Few data in the surface and bottom waters of the Rhône River mouth are available (see Bonin et al., 2002, Wat. Res. 36, 722-732). This paper will be cited in the revised manuscript.

L. 400-402: True, but as you already discuss later on, if dissimilatory iron reduction is coupled to FeS burial (or re-oxidation of Fe2+), its net efflux on alkalinity is zero. So the process definitely contributes to bulk alkalinity production, but that doesn't necessarily mean it is linked to either alkalinity effluxes or long-term net alkalinity release.

We agree with the reviewer that dissimilatory iron reduction contributes to alkalinity production (buildup in pore waters), but not necessarily to its efflux out of the sediment. Therefore, we called it "bulk TA production" in the original manuscript, to differentiate with the "net TA production" (the difference between bulk production and consumption). This information will be clarified in the revised manuscript. In turn, we disagree on the first part of the comment: FeS burial and Fe2+ reoxidation have opposite effects on TA efflux. Indeed, FeS burial generates an alkalinity efflux (this is the main point of this paper as displayed on Fig. 10, red arrows). In contrast, re-oxidation of Fe2+ consumes alkalinity at the oxic-anoxic boundary and may cancel the diffusion of alkalinity generated during iron oxide reduction out of the sediment. We feel that these differences are well presented in the manuscript and do not require further clarification.

[Figure]

C: If I understand the Burdige and Komada method (L. 406-409) correctly, you already assume this by linking DIC and TA production solely to sulfate consumption. It'd be good to be explicit about this and state which processes are included in this method.

Burdige and Komada discuss how the rcs (DIC/SO42- ratio) can be modified by other processes (methanogenesis, carbonate precipitation/dissolution) or by the fact, that organic matter is already partly oxidized when undergoing sulfate reduction. In this manuscript, we add another possibility for the modulation of rcs: the interaction with the iron cycle. Indeed, as shown in Table 1, this ratio (DIC/SO4) may vary with the iron reaction pathways (from -2.25 to -1.8). This point will be clarified in the revised manuscript.

C: Also, I recently came across a paper (https://doi.org/10.1016/j.marchem.2019.03.004) that uses DIC and TA pore water profiles to quantify sulfate reduction rates. I don't know how their methods are applicable to your work but it might be interesting to include it.

In the cited paper, the shape of TA pore water profiles is used as a proxy for sulfate reduction rates in order to disentangle OSR (Organoclastic Sulfate Reduction) and AOM (Anaerobic Oxidation of Methane) which both consume sulfate. The authors use a reactive transport model and include the precipitation/dissolution of CaMgCO3 to estimate sulfate reduction rates and contributions of the different pathways. Their approach is applied to long sediment cores where such processes develop over several meters (as opposed to our short sediment cores). In contrast to our paper, they do not have iron data, and they discuss the coupling with iron reduction and the precipitation of iron sulfide minerals on a purely theoretical level, which is different than our study. We agree it would be interesting to expand our research with a model study, but we think this should be the object of a different manuscript. This comment will not be addressed further in the revised manuscript.

L. 414-417: Any reason why AOM would be less important at stations A&Z compared

to B? At first sight, the SO4 and other profiles do not look too different from each other.

The reason for proposing that AOM is more significant at station B than at stations A and Z is based on the the low DIC/SO42- ratio observed at station B (the lowest in the whole data set). Unfortunately, we do not have any pore water CH4 data at this station. Station B is still located in the main deposition area of the river plume and receives more organic matter than station AK and K. Compared to station A and Z, station B is characterized by a deeper water depth and lower accumulation rates which may favor long-term stability and therefore development of AOM. As these statements were already speculative, we will likely not expand this discussion in the revised manuscript.

L. 424-425: Can you place this pH of 7.2 into context? Why is a minimum of 7.2 not a 'significant lowering'? (L.423)

pH minima between 7.2 and 7.4 were found just below the oxygen penetration depth in the sediments of all stations. The pH minimum at station A, Z and B was the same as the minimum at stations AK, K and E, where less sulfate was consumed in the sediments. We will modify the revised manuscript to put this information in the context of all the pH profiles collected.

L. 426-442: I spent quite some time looking at equations 12-17 and this method. First, I'd like to see how these equations are derived (e.g. eq. 12 combines eq. 6 and 9, 10). This helps checking them and also the derivation of the ratios. Second, if you look carefully at the equations, you'd see that they are all normalized to SO4. Per mole SO4 the changes in TA and (obviously) SO4 are the same for all six reactions. So the differences in the presented ratios are solely due to the differences in DIC production. Of course this would be different if the equations were presented per mole HCO3 (ratios would be the same, but the changes in TA, DIC and SO4 would be different), but it shows that if you want to link S burial to alkalinity generation (as you do in L. 501-503), the exact pathway of iron sulfide mineral formation doesn't matter.

It is incorrect to state that "the exact pathway of iron sulfide mineral formation does

not matter' as, in this manuscript, we relate alkalinity generation to FeS burial and not $SO_4^{2-}$ consumption in the sediment. As can be established from equations 12 and 15, the TA/FeS production ratio can be 2/(2/3)=3 in equation 12 and 2/1=2 in equation 15. Hence, the exact pathway matters and we favor (see paper) the dissimilatory pathway for iron hydroxides with sulfate reduction and precipitation of FeS, hence a ratio of 2. Either way, when comparing measured to theoretical ratios, the method assumes that there is no other removal pathway of DIC (e.g. siderite formation, to name an option). Can this indeed be excluded?

The precipitation of iron and sulfide is extremely fast, likely much faster than precipitation of siderite ($FeCO_3$). Thus, we did not consider this or other pathways in our calculations. This information and the appropriate references supporting that statement will be added in the revised manuscript.

L. 448-452: It'd be nice to read about the possible pathways of organic-Fe(III) complex formation earlier in the manuscript

We recognize that this information could have been provided in the introduction. However, to focus the introduction on the role of carbon mineralization processes on alkalinity generation and avoid increasing the length of the manuscript we chose to not present any detailed biogeochemical pathways in the introduction. This comment will not be considered further in the revised manuscript.

L. 458-460: This statement is less vague than in the abstract, but it still raises questions. At what time scale do these alterations take place? Should I regard 'dominated by sulfate reduction' as the default state of the sediment, only periodically (episodically? seasonally?) replaced by 'dominated by iron oxide reduction' in periods of intense flooding and sediment deposition? Is FeS mineral formation limited by sulfide and if so, does that mean that the flooding periods overprint the default state?

The number of questions raised by the reviewer indicates that this sentence is misleading. Furthermore, the limited amount of information on temporal dynamics on this

system regarding the redox state of the pore waters prevents a sound answer to these questions. We thus decided to change the sentence, keeping the dynamic nature of the system "these sediments are highly dynamic" but to remove the potential temporal succession "periods of intense sulfate reduction alternating with . . ." by "suggest these sediments are highly dynamic with concomitant intense sulfate reduction, microbial iron reduction and rapid FeS precipitation" which is a plausible explanation. Further work will be needed to elucidate these processes and their temporal succession. The abstract will also be modified in that direction.

L.478-481: So if I understand this correctly, it means that pIAPs are poor indicators of mineral formation, as they are highest at the site with least burial (station E). Does this mean that microenvironments play an important role in the formation of FeS? I'm also not sure if I understand what you mean to say by the argument of stronger aggregation of FeS (L. 483). If FeS is currently more aggregated, does that mean that FeS formation is not active now (given undersaturation and no FeSaq) but that it had been active in the recent past in a time when the sediments were sulfate-dominated instead of iron-dominated? (this links back to my previous comment). Or does it simply mean that FeS formation just take place in microniches where local conditions are different?

Mineral formation can be determined from the pIAP assuming the system is at equilibrium. At all stations besides station E, the pore waters were either undersaturated or near saturation with respect to FeS, even though the presence of significant FeS0 concentrations and removal of sulfide from pore waters indicated mineral formation. This disagreement with the calculated pIAP here indicates that the system was not at equilibrium and provides another piece of evidence for a highly dynamic system. As indicated earlier, to shorten the manuscript and not diluting the take-home message of the manuscript, the pIAP calculations will be moved to the supplementary material. Per line 483: the lack of FeS(aq) signals, but presence of high FeS0 concentrations (which pass through the rizon filters) suggests that that iron sulfide particles were already aggregated at the time of sampling. Again this points to a dynamic zone with iron

sulfide precipitation dependent on organic and inorganic (i.e., Fe(III) oxides) inputs. As suspected by the reviewer, microniches, such as leave fragments in the proximal zone, have been shown to play a role in FeS/FeS2 formation (see Charles et al., 2014 and response to Reviewer 2 below). These may have changed local conditions or exarcerbated FeS formation kinetics such that the system may appear undersaturated. This part of the comment will be addressed together with pIAP calculation in the supplementary material.

L. 493-495: This depends on the fate of the other products, i.e. what happens to the produced S0. But if you assume that the S0 will also be buried (or converted to FeS, which are both more likely options than reoxidation), the alkalinity release will always be 2 moles per mole S burial.

It is correct to state that 2 moles of TA will be released per mole S buried, but our interest is on the FeS form which was measured as AVS. In that respect, the ratio of net alkalinity flux to buried FeS will not be 2 (but 3 see above) if S0 is buried which would be the most probable option given the high sedimentation rate and the short residence time of the sediment layer near the sediment-water interface. In our case, we favor Eq. 15 (see explanation in text) and adopted a conservative ratio of 2.

L. 501-504: First, why don't you compare the AVS burial flux with the measured alkalinity flux of station A only, instead of combining A and Z?

The idea of this paper was to check if the conceptual link between FeS burial and alkalinity flux was substantiated by the flux values. Therefore, we tried to come up with order of magnitudes rather than attribute numbers to a single station. Furthermore, recent sedimentation rates were only available for the overall area but not for individual stations. We therefore chose to compare average alkalinity fluxes in the proximal zone and average FeS burial in the same zone. This information will be provided in the revised manuscript.

C: Second, a point that I am just realizing: with this very high burial rate, it'll take

a long time before alkalinity produced in the sediment is diffused out. You'd expect that its transport is dominated by advection, not diffusion. So that would mean an even stronger decoupling between net TA generation in the sediment and measured effluxes. Or is there bioturbation that impacts the benthic release?

As pointed out above in response to a previous comment by the reviewer, the spring-summer diagenetic processes described in this paper are the result of the late fall deposition of flood layers and their maturation. The progressive buildup of alkalinity and DIC in pore waters begins during winter and is certainly accompanied by FeS precipitation as sulfate reduction and iron oxide reduction proceed. The net alkalinity flux produced is linked to the net precipitation of FeS, which represents the difference between precipitation and re-oxidation due to bioturbation transport in the oxic zone. Burial occurs when a new flood layer is deposited (in late fall) which traps the FeS produced during the year before below a new sediment layer of 10-30 cm, ensuring its preservation. As a result, the net alkalinity flux can arise by diffusion over a depth of 15 cm as the diffusion time over this distance is 6 months (with $D(HCO_3)=6\times10^{-6}$ $cm^2/s$ see calculations provided earlier). In the first 5-10 centimeters, transport may be increased by bio-irrigation as can be observed in A and Z DIC/TA profiles which are concave. Except for the flood period, advection is limited as sedimentation remains low. Hence, the processes that produce alkalinity (i.e., FeS precipitation) and occur in the first 20 cm of sediment can most probably be linked to the bottom water fluxes by diffusion during the late spring, summer, and early fall. These concepts will be added to the revised manuscript to clarify this point.

L. 526-529: I agree that microniches can be important, but do your Ca2+ porewater profiles give any indication of CaCO3 dissolution at the top of the sediment?

In the sediment surface layer at station A and Z, Ca2+ concentrations either decrease (St.A) or remain constant with a further decrease deeper (St.Z), thus providing no sign of carbonate dissolution in the upper layers. Furthermore, with the uncertainty of 2-3 % as reported in the method section (line 191), it is difficult to detect minor changes of

Ca2+ linked to small dissolution of carbonate. This evidence for low CaCO3 dissolution from the data collected will be higlighted in the revised manuscript.

L. 534-539: Is FeS the dominant form of solid-phase S in the sediment, or is pyrite also present in substantial amounts?

Only FeS was measured in these sediments in a limited number of cores as these analyses take considerable time. Pyrite precipitation may probably explain the unaccounted TA flux from the FeS burial calculations, but given the residence time of the sediment layers in the Rhône River prodelta, pyrite precipitation most probably occurs after a few years, when the sediment is buried deep in the sediment. This discussion will be provided in the revised manuscript. L. 540-541: On what timescale do these processes take place? Under steady-state conditions I understand this figure, but given the highly variable sedimentation rate at especially the proximal sites, does it still apply under these dynamic conditions?

See comments above about the temporal variations. A couple of sentences explaining the time frame in which biogeochemical processes occur in these proximal sediments will be provided in the revised manuscript.

L. 545-547: but is Fe or S generally limiting FeS formation at the proximal sites?

As no or little sulfide was found in pore waters, it seems reasonable to assume that sulfide is the limiting element. This information will be provided in the revised manuscript.

L. 556-559: so at station E Fe is limiting FeS formation, what about the prodelta sites?

At Station E, the largest fraction of OM oxidation occurs via oxic and suboxic respiratory processes, and FeS concentrations are much lower than at the prodelta sites. Based on the high deposition rates observed in the prodelta, these sites receive increased iron inputs from the Rhône and are thus less likely to be iron limited. In turn, sulfide seems to be the limiting element at these sites. These facts will be emphasized in the revised manuscript.
L. 574-576: but if the FeS burial sink is permanent, it definitely impacts water-column TA and carbonate system dynamics on the long term, as you also indicate on L.576-578 and L.601-604. I think this statement unnecessarily weakens the relevance of your manuscript.

We agree with the reviewer that " [The TA source] definitely impacts water-column TA and carbonate system dynamics on the long term", but the sentence line 574 still holds as the sediments release more DIC than TA and thus contribute to increasing pCO2 of bottom waters rather than decreasing. In turn, this increase of pCO2 is weaker due to the concomitant TA release compared to what it would be if only DIC is released. Hence, it is crucial to determine the TA sources from anaerobic sediments. This sentence will be modified slightly to emphasize these points.

Fig. 9: What if TA data were used for this calculation? I agree though that using DIC is wiser given the possible presence of organic alkalinity.

For these calculations, it is not possible to use the alkalinity profile as its shape near the SWI (at the centimeter scale) is unknown and probably very different from the DIC profile, as it is likely affected by the eventual reoxidation of reduced species (Fe2+, Mn2+, NH4+, HS-) which consume TA. In this paper, we conclude that a major fraction of these reduced species is buried within the anaerobic sediment layers, but some may still be oxidized and thus consume TA in the first mm of the sediment. Although uncertain, the assumption of a linear DIC profile is reasonable but already questionable and represents the best option given the potential variations in the TA profiles. Given this argument, this comment will not be considered further in the revised manuscript.

Technical corrections: Unless addressed specifically below, we agree with all the technical corrections provided below, and these comments will be incorporated in the revised manuscript.

L. 3-6: This sentence is too complex. OC respiration in sediment or water column? I'd suggest to rewrite and / or split it in two sentences. L. 31-33: Ambiguous sentence.

Does "of which about half is buried" refer to total oceanic POC or the 40% that is buried in shelf regions? L. 56: typo in 'anaerobic' L. 77: replace 'sediments' with 'suspended matter' or 'particles' L. 98: "These". All sediments or those in the proximal region only? L. 127-130 and various other sections in the manuscript (basically everywhere where equations are presented): add units to the variables you discuss here (i.e. Fi, H, Ci, etc). L. 235-239: This sentence is too complex. Please split into two or rephrase. L. 242: Add scale and temperature to pH. L. 244: For which salinity are these numbers valid? Result section: may be shortened L. 288:change to "the absolute value of the DOU fluxes" or equivalent as they have opposing signs. L. 306: 'station' instead of 'stations' L. 522: seacarb is written without capitals. Figures: Add units to the captions. Fig. 1: Add the depth to the last (lowest) line of the bathymetry. Fig. 2: Use different lines (e.g. solid and dotted) for O2 and pH. Printed in black & white the figure is currently very difficult to read. Fig. 3: Could be moved to an online supplement. Add what the difference between the red and black symbols means. The error bars complicate reading of the symbols a bit, but I appreciate that they're in.

The red symbols were used involuntarily in this figure and will be converted to black symbols in the revised version. The figure will be moved into the supplemental material section to decrease the length of the manuscript.

Fig. 4: I'd only plot the error bars outwards, this makes the bar plot better readable and the error bar is not visible in the black bars anyway. Fig. 5: Make it clear which measurements are from the duplicate core by using the same symbols for DIC,TA and SO4, and make them clearly different from the main core data. Fig. 6: Add a (dashed) line at $\Omega$ca=1. Also, the DIC data are poorly visible as they are mostly hidden behind the TA data. I'd leave it like this only if the point you're trying to make is that they are so similar. Fig. 7: I'd suggest not splitting the axes into two domains, given the small jump on both axes it complicates more than that it helps reading. Fig. 11: "as a function of water depth". Add the source of the North Sea data (Brenner et al.?). Hu & Cai (2013) is not in the list of references. Table 1: Be

consistent with sulfate and SO4 in the caption. I think that in equation (5) it should read -1/5 H+ (instead of -2/5). Show how you derived equations (12) to (17), see earlier comment. Table 2: Add the depth interval over which mean porosity was calculated.

Please also note the supplement to this comment:
https://www.biogeosciences-discuss.net/bg-2019-32/bg-2019-32-AC1-supplement.pdf

[Figure]

Figure AC1 : DIC and TA fluxes measured on cores in the Rhone Prodelta and shelf in May 2018

[Figure]

Fig. AC2 : proportion of HCO3- in DIC during the Amor-BFlux cruise in 2015

---

## Author Comment (AC2) · 24 May 2019

(Please seee formatted response as pdf in supplement AC2-RC2)

Response to Anonymous Referee #2

C: I read carefully the manuscript of Rassmann, Eitel, and collaborators and I recommend it for publication after revision. This paper presents an in-depth analysis of benthic biogeochemical processes and DIC/TA release in different stations in the Rhône River Delta area. I particularly like the multitude of measurements applied to improve understanding of processes driving anaerobic formation of TA and benthic fluxes; particularly, combining in situ incubations, potentiometric and voltammetric micro-profiles with more conventional pore water and sediment analyses. This combination of methods is rarely encountered in these types of investigations which often focus primarily on submilli metric processes at the SWI. Furthermore, the amount of data collected is significant, and has to be published, definitively.

C: My main concern is related to the overall perspective of the research. The authors do not convey very clearly the scientific importance of their work. For instance, in the introduction they mention that the objective of the paper was to " investigate if sediments from deltaic regions exposed to large riverine inputs of carbon and minerals represent an alkalinity source to the bottom waters and identify the biogeochemical processes responsible for the net production of alkalinity in these sediments. ". There are several studies that have done this in coastal area as well, thus, they should portray how this work is different. And I would say that the combination of methods is unique.

C: After reading the paper several time, I am still not sure to understand the take home message of this study. What is really new?

R: We appreciate the reviewer's comment that the data collected is unique and improves our understanding of the processes driving anaerobic TA formation. In turn, we are disappointed that the reviewer does not understand the novelty of this work and will modify the revised manuscript to emphasize the scientific importance of this work: that the burial of reduced iron and sulfur in the sediment prevents reoxidation of reduced metabolites at the sediment-water interface and therefore contributes to an alkalinity flux to the overlying waters. Although these concepts are not novel, to our knowledge, this manuscript provides for the first time in situ benthic alkalinity flux data and simultaneous biogeochemical evidence from pore water and sediment profiles that substantiate this argument (strong indications that FeS precipitate in the sediment column, calculation of FeS burial compared to Alkalinity generation). The abstract, discussion, and conclusions will be modified to emphasize the fact that this is the first study demonstrating benthic alkalinity flux and simultaneously providing biogeochemical evidence

for the processes responsible for this alkalinity flux.

C: I strongly appreciated the effort to present this very large set of data that includes Fe(III)-Lorg, sulfide species, and methane in addition to major (or classical) diagenetic species. However, I often lost myself in detail that ultimately brings little. For examples, the section on the role of nitrification/ denitrification and the section on IAPs are long but their conclusions are not very relevant for the rest of discussion. Overall the discussion should be shortened.

R: We agree with the reviewer that the IAP section is long and we will move it to the supplemental material to shorten the manuscript as indicated in response to the same comment by Reviewer 1. In turn, we disagree that the role of nitrification/denitrification is not relevant to the discussion. Denitrification is well known to generate TA, though nitrification consumes TA as well such that only exogenous nitrate fed by the River in this case could contribute to TA production. This information has to be provided in the discussion but we will keep it concise in the revised manuscript to satisfy this reviewer's comment.

C: My second concern is on the role of terrigenous organic matter in this type of sediment. The authors characterized the study site as "deltaic sediments exposed to large riverine inputs of inorganic and organic material". In these sediments, coarse particulate organic matter is deposited during flood events and supports the establishment of sulfidic conditions and the precipitation of Fe-S phases (François Charles et al., 2014; Fagervold et al., 2014; Rassmann et al., 2016). As POM, CPOM is probably a source of DOM and organic alkalinity in pore water. What is the role of organic alkalinity on TA in these sediments? Did you calculate the theoretical TA based on DIC and pH?

R: This issue has been discussed as well by reviewer 1. We calculated theoretical TA in the bottom waters based on the DIC and pH values. The departure between theoretical and measured TA provided bottom water organic alkalinity ranging between 8-28 $\mu$mol/l. In the sediment, we only have pH in the first 2 cm on a vertical resolution of

200 $\mu$m whereas DIC and TA have been measured with a vertical resolution of 2 cm. As explained earlier in response to the same comment by reviewer 1, this difference in resolution is an issue that is difficult to overcome, and organic alkalinity increases generally in the anoxic zone deeper in the sediment, where no pH data are available. A first guess using an averaged pH value around 0, 1, and 2 cm depth leads to concentrations of organic alkalinity not exceeding 60 $\mu$mol/L. These low organic alkalinity concentrations in surface sediments may also prevail in deeper layers as the DIC/TA ratio is close to 1 (Figure 6) which is not the case when large quantities of organic alkalinity accumulate in sediments. The occurrence of large organic alkalinity is definitely an issue that could be resolved in further investigations by measuring pH profiles over longer depths in the sediment. As our dataset is not appropriate for the calculation of organic alkalinity at depth (where it occurs mostly) and the lack of organic alkalinity data does not change the main conclusions of the manuscript, we will not address this comment further in the revised manuscript.

C: The production of organic alkalinity should be discussed and the contribution to TA should be estimated (it's rare to have enough data to do it).

R: Yes, but as already explained, we lack data to estimate the amount of organic alkalinity in the sediments.

C: In addition, the accumulation of refractory organic carbon in sediments appears intimately associated with the sequestering of iron and sulfides in micro-environments (see the works of François). When the authors discuss about the aggregation of FeS, do they talk about microenvironments?

R: No, the main point of mentioning FeS aggregation (size of superior or inferior to 5nm) was to explain the differences between FeS0 and voltammetrically determined FeSaq, which points to a dynamic system. Microenvironments will be mentioned in the manuscript as they have been shown to play a role in the proximal zone with FeS2 formation on leaves.

C: I think the role of terrigenous organic matter on these biogeochemical processes should be clarified.

R: Previous investigations used Delta C14 and Delta C13 isotope analyses to highlight the importance of organic material in the Rhône River prodelta (Tesi et al., 2007; Lansard et al., 2009; Cathalot et al., 2013; Pozzato et al., 2018). These studies demonstrated that terrestrial and riverine organic matter (i.e., produced by riverine primary production) constitute the major fraction of organic matter in the sediments close to the river mouth (station A and Z) and that the terrestrial fraction decreases with distance from the river mouth and is lowest at station E. As the characteristics of natural organic matter might be of interest to the reader, we will briefly present the average oxidation state, POM concentrations, and C/N ratios in section 2.1 of the revised manuscript.

C: My third concern is on the time scale of the explored biogeochemical processes. The deltaic sediments cannot be considered at steady state, specifically in the proximal stations. However, the discussion is based on a steady state view of the different reactions. So, what is the impact of floods on the oxygen demand and DIC/TA release? Are these fluxes constant over the year, with no seasonal variations?

R: The question of temporal and spatial scales is crucial in these environments and we tried to introduce them in the paper although we probably did not include enough material dealing with this topic in the background (1. Introduction) and field site description (2.1: The Rhone River delta) sections. These two sections will be updated with additional sentences on the temporal and spatial scales as explained in response to the same point made by Reviewer 1 (see 2nd comment by Reviewer 1). "Concerning the temporal variations, the Rhône prodelta system receives large inputs of particulate material in late fall and early winter during major river floods (as stated line 87-90), including terrestrial organic matter which is mineralized in the spring and summer. Although data are scarce and new programs are underway to address temporal variations, there seems to be a progressive buildup of metabolites in the sediment pore waters and progressive disappearance of fronts during winter and spring (Rassmann,

unpublished data). One good example is found in Pastor et al. (2018, CSR) which describes an atypical flood in the spring and the evolution of the pore water profiles over 6 months from May to December (see DIC profiles of the prodelta station A in Fig. 4 of Pastor et al., 2018). In the general case of the fall floods, the most intense organic matter mineralization rates are reached in spring and summer in the proximal zone, where organic matter accumulation is highest, as presented in Rassmann et al (2016) for June 2014 where sulfate reduction drives the sulfate concentrations to almost 0, producing 30-40 mM of DIC and alkalinity with 500-800 $\mu$M of dissolved iron and no sulfide in the pore waters (see also Pastor et al., 2011 for a similar situation in April 2007). In addition, this pattern was found over several sampling campaigns in April 2013 (Dumoulin et al., 2018), 2014 (Rassmann et al., 2016), 2015 (this paper), and 2018 (unpublished results). Altogether, the pore water data collected over the years in the Rhône prodelta system are consistent and indicate that biogeochemical processes in the critical proximal zone reaches a reproducible state on a yearly basis due to the regularity of flood deposition in late fall and maturation of the system in spring and summer. This reproducibility of the spring-summer conditions applies probably to the observed fluxes as well."

C: "[. . .] their presence in zones of sulfate reduction suggest these sediments are highly dynamic with periods of intense sulfate reduction alternating with periods during which sulfate reduction is repressed and replaced by microbial iron reduction (line 458-460). Does this sentence mean that there are two different conditions depending of the flood conditions or seasons? What are the consequences on FeS precipitation and on TA release? This sentence is too vague and raises questions on the temporal representativeness of the data (episodic event? Seasonal variations?).

R: Both reviewers were puzzled by this sentence (see specific comment by Reviewer 1 about lines 458-460), and we think the sentence is misleading. It will be modified to "suggest these sediments are highly dynamic with concomitant intense sulfate reduction, microbial iron reduction and rapid FeS precipitation" as already discussed in

response to Reviewer 1. The abstract will also be modified to reflect this comment.

C: I have the same questions on the spatial representativeness of the data. How do the authors explain the difference between the two replicates Z and Z'? Then, I encourage the authors to discuss about the spatio-temporal representativeness of observations.

R: The difference between both samplings can be due to spatial heterogeneity at the local scale due to differential deposition at these sites. As explained in response to the same comment by Reviewer 1 (see spatial variations in response to the 2nd comment by Reviewer 1), we have already described the different zones of the Rhône delta from the proximal zone to the continental shelf (lines 82-86) and their characteristics (sedimentation rates, depth, organic carbon content; see also Table 2) in the original manuscript. Something surely missing in this paragraph is an appreciation of the spatial heterogeneity at the local scale. As the main deposition of sediment occurs during floods, sediment layers are heterogeneous at the meter scale, and differences in pore water profiles can be detected at that scale (e.g. DIC and TA profiles on Fig. 6 for station A and Z taken from two different cores at the same station). These points will be highlighted in the revised manuscript. However, even with this local variability taken into consideration, the difference between the proximal zone stations and stations in the prodelta or the continental shelf are still obvious as highlighted in the discussion of the original manuscript (see sections 4.5, 4.6, 4.7, and 4.8) (see also Rassmann et al., 2016).

C: My last issue is on the role of bioturbation. I think about bioturbation when I looked at the figure 10. According to the frequency of flood events and to the accumulation rates at the proximal stations, the diffusive transport of the anaerobically-produced alkalinity in the flood deposit to the SWI, takes time no? (see the work of Anschutz and collaborators in natural turbidites (Anschutz et al., 2002; Chaillou et al., 2006) and in experimental turbidites (Chaillou et al., 2007)).

R: As pointed out in response to similar comments by Reviewer 1 (see 2nd main comment and specific comment of L501-504), the spring-summer diagenetic processes described in this paper are the result of the late fall deposition of flood layers and their maturation. The fluxes of alkalinity and DIC certainly vary over time with the progressive buildup of alkalinity and DIC in pore waters, as sulfate reduction, iron oxide reduction, and FeS precipitation proceed during the spring and summer seasons. The alkalinity flux, however, is linked to the net precipitation of FeS, which is the difference between precipitation and re-oxidation due to transport in the oxic zone. Burial occurs when a new flood layer is deposited (in late fall) which traps the FeS produced during the year below a new sediment layer of 10-30 cm ensuring its preservation. This yearly preservation of FeS ensures the concomitant benthic alkalinity flux to represent a net flux to the bottom waters which is not affected by FeS oxidation, contrarily to sediments exposed to low sedimentation rates where FeS can be entrained by bioturbation to the oxygenated layer and be re-oxidized (therefore consuming alkalinity). As a result, the net alkalinity flux can arise by diffusion over a depth of 15 cm as the diffusion time over this distance is 6 months (with $D(HCO_3^-)=6\times10^{-6}$ cm2/s). In the first 5 centimeters, transport may be increased by bio-irrigation as can be observed in the A and Z concave DIC/TA profiles. Except for the flood period, advection is limited as sedimentation remains low. Hence, the processes that produce alkalinity (i.e., FeS precipitation) and occur in the first 20 cm of sediment can most probably be linked to the bottom water fluxes by diffusion during the late spring, summer, and early fall. These concepts will be added to the revised manuscript to clarify this point. Anschutz et al. (2002) on a 4 month-old natural turbidite and Chaillou et al. (2007) on artificial turbidite showed that after a few month steady-state was not reached (10 months, Chaillou et al., 2007) but that a relative buildup of nutrients (i.e., NH4+ as an integrative diagenetic indicator) occurred. In Anschutz et al (2002), concentration of NH4+ $>400\mu$M were recorded at the bottom of the turbidite layer implying rapid diagenesis after turbidite deposition. Unfortunately, DIC was not measured in these studies, although it is likely that DIC profiles would have been similar to NH4+ profiles. In Pastor et al. (2011) in the Rhone prodelta, a peculiar flood (occurring in spring) was followed over 6 months: DIC connone

centrations reached 35mM at the bottom of the flood layer (30 cm) after 6 months, with a shape similar to normal-year spring and summer profiles. We will acknowledge the non steady-state issues in the new manuscript by referencing these papers.

C: Bioturbation and biodiffusion could be an efficient mechanism to transport anaerobically-produced metabolites, as TA from the anaerobic zone to the surface. Did the authors measure the bioturbation coefficients in the incubations? Did they consider the macrofauna in the studied sediment? What about the difference between total fluxes and diffusive fluxes of DO, TA and DIC? Are they similar (same magnitude)?

R: Bioturbation or bioirrigation was not measured during this cruise, although macrofauna has been measured in previous cruises (see Charles et al., 2014 or Bonifacio et al., 2014). In addition, bioirrigation is poorly quantified in the study area except the paper by Lansard et al. (2009) in which total oxygen Uptake (TOU) was compared to Diffusive oxygen uptake (DOU) in sediment incubations to estimate the effect of bioirrigation. TOU/DOU ratios of 1.2 +/- 0.4 were found over the offshore transect indicating that bio-irrigation is probably not particularly efficient in this zone. The role of bioturbation on the transport of TA and other pore water constituents can unfortunately not be determined by comparing diffusive fluxes of TA and DIC to benthic TA and DIC fluxes, as the vertical resolution of the rhizons is too low ($\sim$ 2 cm) to be able to estimate accurate gradients near the sediment-water interface. TOU fluxes were unfortunately not measured.

C: The authors are kindly asked to see the attached annotated PDF with my suggestions

R: The comments of the attached annotated pdf document will be taken into account. Most of them are minor comments. Here we will only respond to the comments on the content of the manuscript. Finally, we agree with the majority of the specific comments provided below, and these comments will be incorporated in the revised manuscript unless addressed specifically below.

[Figure]

L19: "Not sure to understand the link between organic-Fe(III) and the variability of the organic and inorganic particulate input..."

This point was already brought up by Reviewer 1 that some statements need context in the abstract (see response to specific comments of lines 19-21 by Reviewer 1).

L45: "produce?" implied to be better than 'create TA'

The change will be done

L55: reference needed

Several references were already provided in the sentences immediately above this one, such that we feel they do not need to be repeated here.

L66: "what is the role of the organic alkalinity produced by the terrestrial OM? Please discuss about the role of OM source on the production of TA."

See the response to the 2nd concern of Reviewer 2 above.

L77: "? Fe-rich particles?"

We mean mostly iron oxyhydroxides ($FeOH3$) . This comment will be ignored.

L95: "It 's very high. Detrital IC input from where? geology in the watershed?"

The carbonate content quoted here comes from numerous studies of C in sediments. There are many calcareous formations in the drainage basin of the Rhône River especially in the Southern Alps.

L99: "does it mean that you will performed diffusif and total flux calculations? what about the role of bioturbation to the exfluxes of TA and DIC?"

As mentioned above in response to the last main concern of the reviewer, we cannot compute the diffusive DIC and TA fluxes as the vertical resolution of the rhizons is too low ($\sim$2 cm) to determine accurate gradients near the sediment-water interface. Concerning the role of bio-irrigation on the fluxes, see remarks above and the low

TOU/DOU ratio measured in the area (1.2+-0.4).

L105: "ok, see my comment on the table 2"

L114: "Which ones? please detailled"

This information will be provided in the revised manuscript.

L122: "Do you measure the bioturbation / bioirrigation coefficient? "

No, we did not measure bioturbation, as indicated earlier.

L125: "with the syringe system?"

Yes, with the syringe system described by Jahnke and Christiansen, 1989.

L130: "and oxygen? "

Unfortunately not, due to technical issues.

L141: "no TRIS?"

We did not use TRIS buffers, but corrected the salinity shift by using pH values of the bottom water measured via spectrophotometry (line 139). This information will be provided in the revised manuscript.

L145: "what is the porewater volume you collected? ∼20mL? enough to mesaure all the parameters?"

Pore water volumes were between 12 and 15 ml. 3 to 5 ml were used for DIC, 3 ml for TA, 1 ml for $NH_4+$ and 1 ml for $SO_4^2-$, 1-2 ml for $Ca^{2+}$, 1 ml for phosphate and dissolved iron. $CH_4$ was measured on a separate core.

L169: "I know this technique but I am surprise by the volume you collected with the pore size of 0.1um. "

According to the manufacturers, this is the pore size of the rhizons.

L171: "Viollier et al. for Fe2+? The ferrozine method revisited: Fe(II)/Fe(III) determination in natural waters."

We used Stookey (1970) as a basis for the method. The method of Viollier et al. is based on Stookey's method and only use a 10 minute reaction period with hydroxylamine which we extended to 24 hours, as initially reported in Stookey. We feel this information is not needed in the revised manuscript, as the methods section only needs to report what was conducted.

L181: "these methods seem complicated. Why not use the Rodier method ? I am just curious. Why two different methods ? what is the difference"

These two methods are HPLC methods which are more precise than the nephelometric method (quoted as Rodier). HPLC ensures a precision of around 1% for sulfate determination and requires 100 $\mu$l whereas the nephelometric method requires larger samples and has an uncertainty of 3-5%. As some of our stations had limited SO42- decrease and the amount of sample was small, we preferred using HPLC methods. We used two methods to establish a comparison between the methods for our own purposes. Motivation for the two HPLC methods will be provided in the revised manuscript.

L188: "I suppose it is only for the porewater Ca concentrations. not for the determination of total Ca fluxes in the chamber."

Yes, it is only for pore waters. Actually, we tried both, but the changes in Ca2+ concentrations over time were too small compared to the uncertainty of the used method. This comment will not be addressed in the revised manuscript.

L191: "please add the limits of detection"

L212: "Did you use the PHREEQC software to do the IAP calculation? please add information Same question for the omega calcite calculations "

As mentioned in response to a similar comment by Reviewer 1, IAP were estimated in a spreadsheet. The equilibrium constant was recalculated at the ionic strength of seawater, the measured Fe2+ concentrations were used as 'free' available Fe2+, as Fe2+ does not form strong complexes, and 2S concentrations were used to calculate the speciation of sulfide species (assuming no elemental sulfur or polysulfide were present in the pore waters). Finally, the activity coefficients provided in the methods section were used for these calculations. These details will be provided in the supplemental material of the revised manuscript. For the OMEGA calcite calculations, the Seacarb package for R was used.

L227: "Whatever the depth? DO you calculate de AVS burial for each depth/layer?"

As mentioned in these lines, we estimated an average concentration for the core and multiplied by the sedimentation rate and porosity. We feel the description was clear enough to not warrant any modifications in the revised manuscript.

L235: "this is the data repported in the table 3 (corrected from Diffusion and Ca). Ok I understand now! It is difficult to understand what do you want to do. I think you need to better introduce the goal of these calculations. What is the avantage to do these calculations compared to use a DIC - TA diagram where you add the theoretical slopes of the different reactions and the corrected data? Please see the last paper of Pain et al. 2018 and the figure 9 therein"

These calculations will be better introduced in the revised manuscript though similar calculations were already reported in Rassmann et al. (2016). We are not sure if adding a new figure would be useful, but we will reference the paper of Pain et al. 2018 to clarify the methodology used.

L242: "Do Di and Dj corrected from temperature?"

Yes, the diffusion coefficients were calculated as a function of temperature, pressure, and salinity. This information will be provided in the revised manuscript.

L242: "Did you use a softawar (Seacarb or PHREEQC?"

No, these calculations were conducted in Excel using the functions reported in the

literature (Li and Gregory, 1974). This information will be provided in the revised manuscript.

L246: "ok but TA is not only carbonate alkalinity TA = [HCO3-] + 2[CO32-] + [B(OH)4-] + [OH-] + [HPO42-] + 2[PO43-] + [H3SiO4-] + 2[H2SiO42-] + [HS-] + 2[S-] + [NH3+] + [Org-] - [H+] - [H3PO4] See my previous comment on the role of OM and organic alkalinity on TA"

Yes, but carbonate alkalinity is the major fraction of TA. As already explained above in response to a similar comment by this reviewer, we are lacking data to estimate the non-carbonate alkalinity fraction accurately.

L247: "You measure TA but you talk about carbonate alkalinity in the text. Please clarify " This comment is not clear, as we did not discuss carbonate alkalinity in the lines identified by the reviewer. We only describe the effect of the precipitation of calcium carbonate on TA and DIC variations in this sections.

L270: "please change the number it is figure 3."

The figure number is correct. Oxygen was measured with Clark type electrodes (Figure 2) and by voltammetry (Figure 5). Both methods concur. Figure 3 shows the evolution of DIC and TA concentrations in the benthic chamber over time and will be moved into the supplementary material.

L287: ""relative importance" : not clear, what does "importance" mean ?"

L289: "what about Mn2+? not used? please delete the data from the figure" A statement will be added to explain that Mn2+ was undetected in most sediments from the Rhône River prodelta

L292: "mean value over the lenght of the core?"

Here we are actually describing maximum concentrations. These details will be added to the revised manuscript.

L343: "below the limit of detection or not measured?"

As mentioned in this sentence, dissolved phosphate was not measured at station B. As this sentence is clear enough, this comment will not be considered further in the revised manuscript.

L357: "so it's mainly based on a qualitative approach. Despite the lenght of discussion, there is still some questions on: - the "steady state" approach of the redox processes under sedimentary transient conditions, (especially at the proximal stations A Z and to a lesser extent AK

L357 does not address the 'steady-state' approach, and this comment by the reviewer is not clear. However, the problem of comparing profiles obtained at one time period to TA fluxes integrated over several months has already been addressed in response to the 3rd main concern of the reviewer (see above). This comment will not be addressed further at this location in the revised manuscript.

- the role of terrestrial organic matter on benthic TA, mainly in the proximal zone

Similarly, we already addressed the role of terrestrial organic matter on the benthic TA flux in response to the 2nd main concern of Reviewer 2. We refer to our above response to address this comment.

- the role of bioturbation to explain the exfluxes of TA and DIC

As mentioned above in response to the last main concern of Reviewer 2, bioturbation is poorly constrained in this area, though high resolution electrochemical profiles show no apparent effect on the gradients near the sediment-water interface. In addition, the measured benthic TA and DIC fluxes are in situ fluxes over 30 x 30 cm of enclosed sediment such that bioturbation should be accounted for in these measurements. Besides the changes highlighted in response to the last main concern of the reviewer, this comment will not be addressed further here.

The sections on the formation of iron sulfide species and FeS precipitations are too

**BGD**

long and could be shortened. I don't understand what is really new here

The section will be shortened and the novelties better highlighted to shorten the discussion.

L426: "just the distance between the two fronts ((>5cm)"

This comment is not clear and cannot be addressed easily. Sulfide concentrations are low in each sediment core below the oxygen penetration depth, suggesting not oxidation but FeS precipitation. This comment will not be considered further in the revised manuscript.

L456: "I am not sure this figure is necessary" for Fig. 7

We disagree as the correlation between Fe2+ and organic-Fe(III) provides strong evidence for the microbial reduction of Fe(III) oxides as discussed in the manuscript. This information appears important to highlight the possible concomitant nature of the microbial processes in these sediments. This comment will not be considered in the revised manuscript.

L488: "so is it necessary to present the PIAP (in supplementary material?). The discussion should be shortened "

We agree, as acknowledged in response to comments of both reviewers. This will shorten the manuscript significantly.

L491: "yes, but the accumulation is not constant over the year... not at steady state after the deposition"

We recognize that these sedimentation events are extreme and the temporal variations of sedimentation rates will be emphasized in the revised manuscript as explained above in response to the comments of both reviewers.

L511: "For me, it is the first step of the discussion (it is also the first reaction presented in the table 1). I suggest to the authors to move this section to start the discussion."

This is an interesting idea by the reviewer, as we could move this section earlier in the discussion (after section 4.1), though the conclusion of this section requires to present the effect of FeS burial on the alkalinity flux for comparison. Although minor, we will definitely consider this option in the revised manuscript.

L546: "Yes but I suppose this rate is not constant : massive input of "new" and terrigeneous-rich OM material"

We addressed the seasonal variations associated with flood events in the fall and slow build up of alkalinity during the spring and summer seasons in response to comments made by both reviewers. The discussion will be strengthened by such discussion.

L868: "Why do you show the results in red? I would remove these incubations (Z and Z' in red)"

As mentioned in response to a similar comment by Reviewer 1, this was the result of an error during data plotting. The correct figure will be provided in the revised manuscript.

L891: "below the dl?"

No, not measured. We feel this statement is unambiguous here.

Fig 1: "compared to the others figures, this one is "ugly" !"

The figure will be modified to improve its clarity.

Fig 2: "change the color between O2 and pH. At this scale, dotted lines are not visible"

The figure will be modified for better visibility.

Fig 3: red triangles??

The figure will be reformatted and moved into the supplement section.

Table 2: "Concentrations in the bottom water? please add the information"

This information will be provided in the revised version.

Please also note the supplement to this comment:
https://www.biogeosciences-discuss.net/bg-2019-32/bg-2019-32-AC2-supplement.pdf

---

## Author Response (AR1)

Dear Prof. Dr. Jack Middelburg,

Please find below our revision letter which goes along with our revised manuscript. We have made our modifications according to the answers given to the reviewers comments for BGD.

Our answers to the individual comments are given in blue and when the manuscript has been modified, the modifications are given in *italic* for small changes. When entire paragraphs have been rewritten, only the line numbers are indicated to avoid unnecessary length of this response letter.

We hope that our responses are satisfying and that our manuscript has evolved towards a publishable form for Biogeosciences.

Kind regards,

Dr. Christophe Rabouille

**Anonymous Referee #1**

- General comments: In this study, Rassmann, Eitel et al. investigated benthic alkalinity and DIC release from various sites in the Rhône River delta area. These sites differed in their distance from the river mouth, water depth, and sedimentation rates. The authors measured fluxes to quantify the alkalinity and DIC release, and measured a variety of pore water and sedimentary constituents to investigate the responsible processes. Particular attention was given to the ratio between aerobic and anaerobic organic matter degradation and the role of FeS burial in determining the alkalinity release.
After reading the manuscript, I have somewhat mixed feelings. On the one hand, I appreciate the data set and especially the determination of organic-Fe(III) complexes and FeS nanoparticles, something that is new to me in the context of benthic alkalinity release. On the other hand, after reading I asked myself what the novelty and take home message from this work is and I am not sure if I can properly answer that question.

R: We appreciate the reviewer recognizes that the organic-Fe(III) complexes and FeS nanoparticles are useful in the interpretation of this complex data set. In turn, we are disappointed by the comment that the reviewer does not understand the novelty of this work. This work demonstrates that the burial of reduced iron and sulfur in the sediment prevents reoxidation of reduced metabolites at the sediment-water interface and therefore contributes to the alkalinity flux to the overlying waters. Although these concepts are not novel, to our knowledge, this manuscript provides for the first time in situ benthic alkalinity flux data and simultaneous biogeochemical evidence from pore water and sediment profiles that substantiate this argument. The abstract, discussion, and conclusions has been modified to emphasize the fact that this is the first study demonstrating the link between measured benthic alkalinity flux and biogeochemical processes, namely FeS precipitation and burial, responsible for this alkalinity flux.

We have added and rephrased several sentences to focus the reader better towards our aims:

added Lines 3-8; rephrased Lines 20-22 (abstract): By preventing reduced iron and sulfide from reoxidation, the precipitation and burial of iron sulfide increases the alkalinity release from the sediments; rephrased Lines 76-84 (Introduction); added Lines 419-421 (Discussion)

- Despite the length of the manuscript (I'd suggest to at least shorten the description of the results and move Fig. 3 to a supplement) I was still left with quite some questions. What I generally miss in the manuscript is

an appreciation of various temporal and spatial scales at which both benthic alkalinity generation and its release can be discussed. For example, if reduced constituents responsible for the alkalinity generation are released to the water column and quickly re-oxidized there, would it still contribute to the CO2 storage capacity over longer time scales? Under which conditions is or is this not valid? Also, can the authors directly compare the alkalinity efflux due to FeS burial and the measured effluxes, given the high sedimentation rate, and that effluxes vary on much shorter timescales, and due to many processes other than FeS burial? And finally, how representative are the measured fluxes (and other data) on e.g. an annual timescale given the high variability in inputs over the year? Could the authors indicate that based on their earlier published work?

R: We agree that the manuscript could be shortened, and have moved Fig. 3 (benthic chamber TA and DIC) and Fig. 8 (pIAP) and associated paragraphs presenting the methods and describing or discussing these data to a supplementary material section to shorten the manuscript.

The question of temporal and spatial scales is crucial in these dynamic environments and we tried to introduce them in the manuscript, although we probably did not include enough material dealing with this topic in the background (1. Introduction) and field site description (2.1: The Rhône River delta) sections. These two sections have been updated with additional sentences on the temporal and spatial scales as explained below.

Added Lines 63-70 (Introduction); added Line 95: *"mean apparent accumulation rates of up to 37-48 cm yr-1"*; added Lines 110-127 (section 2.1: Rhône River delta)

Concerning the question on the re-oxidation of reduced components in the water column, re-oxidation of reduced metabolites has the same net effect as re-oxidation in the sediment, i.e. consumption of alkalinity by protons produced during oxidation (see equations 3 and 4 of Table 1). Thus, re-oxidation of reduced metabolites does not contribute to the CO2 storage capacity over long time scales. This is the reason why the burial of reduced components (FeS and potentially FeS2) in the deep sediment layers presented in this manuscript is so important: Burial of FeS and FeS2 prevents re-oxidation in the sediment and in the water column and creates a net flux of alkalinity to the water column. Although we feel that this topic was well covered in the original manuscript (see lines 21-25 of the abstract, lines 51-55 of the introduction, lines 491-493 and 547-550 of the discussion), we have clarified these points in these sections and improved Figure 10 to clearly illustrate the importance of the lack of re-oxidation near the sediment-water interface.

Lines 3-5: *Yet, the intensity of anaerobic respiration processes in the sediments tempered by the reoxidation of reduced metabolites near the sediment-water interface controls the production of benthic alkalinity.*

Lines 26-28: *By preventing reduced iron and sulfide reoxidation, the precipitation and burial of iron sulfide increases the alkalinity release from the sediments*

Concerning the spatial scale, we already described in the original manuscript the different zones of the Rhône delta from the proximal zone to the continental shelf (lines 82-86) and their characteristics (sedimentation rates, depth, organic carbon content; see also Table 2) in the original manuscript. Something surely missing in this paragraph was an appreciation of the spatial heterogeneity at the local scale. As the main deposition of sediment occurs during floods, sediment layers are heterogeneous at the meter scale, and differences in pore water profiles can be detected at that scale (e.g. DIC and TA profiles on Fig. 6 for station A and Z taken from two different cores at the same stations). These points were highlighted in the revised manuscript. However, even with this local variability (at the station scale) taken into consideration, the difference between the proximal zone stations and stations in the prodelta or the continental shelf are still obvious as highlighted in the discussion of the original manuscript (see sections 4.5, 4.6, 4.7, and 4.8) (see also Rassmann et al., 2016).

Lines: 332-334 (Results): *"The relatively high variability between these two measurements is probably due*

*to high spatial heterogeneity of the sediments due to the deposition conditions during floods.*

*Added Lines 433-439 (Discussion): The observed fluxes show some variability between stations in the proximal zone, most probably due to the high inter- (i.e., km scale between stations A and Z) and intra-station (i.e., < 100 m between Z and Z') biogeochemical heterogeneities associated with massive and rapid deposition events during floods. This heterogeneity is also visible in pore water profiles from two different cores at station Z or A (Fig. 5). Despite this subkilometer variability near the river mouth, the biogeochemical gradient from the proximal zone to the continental shelf is large enough to contrast the different zones."*

Concerning the temporal variations, the Rhône prodelta system receives large inputs of particulate material in late fall and early winter during major river floods (as stated line 87-90), including terrestrial organic matter which is mineralized in the spring and summer. Although data are scarce and new programs are underway to address temporal variations, there seems to be a progressive buildup of metabolites in the sediment pore waters and progressive disappearance of fronts during winter and spring (Rassmann, unpublished data). One good example is found in Pastor et al. (2018, CSR) which describes an atypical flood in the Spring and the evolution of the pore water profiles over 6 months from May to December (see DIC profiles of the prodelta station A in Fig. 4 of Pastor et al., 2018). In the general case of the fall floods, the most intense organic matter mineralization rates are reached in spring and summer in the proximal zone, where organic matter accumulation is highest, as presented in Rassmann et al (2016) for June 2014 where sulfate reduction drives the sulfate concentrations to almost 0, producing 30-40 mM of DIC and alkalinity with 500-800 µM of dissolved iron and no sulfide in the pore waters (see also Pastor et al., 2011 for a similar situation in April 2007). In addition, this pattern was found over several sampling campaigns in April 2013 (Dumoulin et al., 2018), 2014 (Rassmann et al., 2016), 2015 (this paper), and 2018 (unpublished results). Altogether, the pore water data collected over the years in the Rhône prodelta system are consistent and indicate that biogeochemical processes in the critical proximal zone reaches a reproducible state on a yearly basis due to the regularity of flood deposition in late fall and maturation of the system in spring and summer. This reproducibility of the spring-summer conditions applies probably to the observed fluxes as well.

*Added Lines 115-127: "Although data are scarce, metabolites from carbon remineralization processes probably build-up progressively during winter and spring (Rassmann, unpublished data). This temporal evolution yields similar diagenetic signatures from mid-spring to end of summer, including almost complete sulfate reduction, large concentration of DIC and alkalinity (30-40 mM), 500-800 µM of dissolved iron, and no dissolved sulfide in the pore waters (Rassmann et al., 2016; Pastor et al., 2011). This pattern was observed consistently over several sampling campaigns, including April 2007 (Pastor et al., 2011), April 2013 (Dumoulin et al., 2018), May 2014 (Rassmann et al., 2016), September 2015 (this paper), and May 2018 (unpublished results). Altogether, the pore water data collected over the years in the Rhône prodelta system are consistent and indicate that biogeochemical processes in the critical proximal zone reach a reproducible state on a yearly basis due to the regularity of flood deposition in late fall and maturation of the system in spring and summer. This reproducibility of the spring-summer conditions probably also applies to benthic fluxes."*

Concerning the comparison between benthic alkalinity and FeS burial fluxes, fluxes of alkalinity and DIC certainly vary over time with the progressive buildup of alkalinity and DIC in pore waters, as sulfate reduction, iron oxide reduction, and FeS precipitation proceed during the spring and summer seasons. The alkalinity flux, however, is linked to the net precipitation of FeS, which is the difference between precipitation and re-oxidation due to transport in the oxic zone. Burial occurs when a new flood layer is deposited (in late fall) which traps the FeS produced during the year below a new sediment layer of 10-30 cm ensuring its preservation. This yearly preservation of FeS ensures the concomitant benthic alkalinity flux to

represent a net flux to the bottom waters which is not affected by FeS oxidation, contrarily to sediments exposed to low sedimentation rates where FeS can be entrained by bioturbation to the oxygenated layer and be re-oxidized (therefore consuming alkalinity).

*Added Lines 601-615: "The connection between alkalinity fluxes at the sediment-water interface and FeS burial at depth is questionable given the low residence time of the sediment near the interface (< 1yr in the first 30 cm) and the temporal variability in deposition processes (see section 2.1). Chemical gradients and thus benthic fluxes are shaped by biogeochemical reactions occurring within the diffusion length, i.e. the distance (d) that can be travelled by diffusion of chemical species over a given time:*

$$d = \sqrt{2 * D * t} \tag{6}$$

*where d is the diffusive length (cm), $D_s$ the diffusion coefficient in the sediment ($cm^2\ s^{-1}$) and t the time (s). For a period of 6 months (between fall and spring), and using the diffusion coefficient of $HCO_3^-$ ($D_s = 7.10^{-6}$ $cm^2\ s^{-1}$ at 20°C), the diffusion distance reaches around 15 cm. This distance represents a minimal estimate as transport is likely enhanced by bioturbation and bioirrigation such that 20 cm of sediment and pore water may be considered connected to the SWI on a semi-annual basis. These findings indicate that biogeochemical processes over that depth interval are able to shape net benthic alkalinity fluxes at the SWI over a 6-month period after the fall floods. The FeS burial effect is strengthened by the episodic but large deposition of new sediment during the following fall floods."*

Concerning the "many processes other than FeS burial" which influence the benthic alkalinity flux, we refer to previous studies which demonstrated only two anaerobic processes (iron sulfide burial and N2 loss by denitrification) contribute to a net release of alkalinity from the sediment through the net loss of reduced species by either of these processes (see for example Hu and Cai, 2011 in GBC). All other internal cycling processes do not represent net sources of alkalinity as highlighted in the discussion already (see lines 419-442 and Table 1). We show in the manuscript that denitrification is a minor source of alkalinity, if any (discussed in section 4.2). The other potential source of alkalinity in sediment is carbonate dissolution which we showed to be absent or minimal due to large oversaturation of the pore waters with respect to calcite (Rassmann et al., 2016 and Figure 9 of this manuscript) supported in this study by the decreasing $Ca^{2+}$ concentrations in the profiles (Figure 6), certainly indicating CaCO3 precipitation.

We merged the sections 4.2 and 4.7 into one new section 4.2 to shorten the manuscript (Lines 454-490)

Concerning the "representativity of the measured fluxes", it is hard to conclude as these measurements represented the first attempt to measure either in situ or ex situ benthic alkalinity DIC and TA fluxes in the Rhône River prodelta. During a more recent cruise in 2018, sediment cores were incubated to estimate DIC and alkalinity fluxes across the sediment-water interface. The (unpublished) results (see graph below) display the same pattern as observed in situ in 2015 (this study) with large DIC and TA fluxes in the proximal zone (A, Z, AZ) and decreasing fluxes offshore (AK, K and C, E on the shelf).

[Figure]

To clarify these points in the manuscript, a few sentences have been added in the introduction on the effect of seasonal patterns on the biogeochemistry of river-dominated sediments and a paragraph in section 2.1 containing information on the temporal variation in depositions in the Rhône River prodelta from previous studies. Finally, the discussion has been modified to highlight the features described above that argue for the decoupling between aerobic and anaerobic processes and the link between FeS burial fluxes and benthic-pelagic alkalinity fluxes.

Added Lines 63-70; added Lines 117-127

added Lines 612-615: *These findings indicate that biogeochemical processes over that depth interval are able to shape net benthic alkalinity fluxes at the SWI over a 6-month period after the fall floods. The FeS burial effect is strengthened by the episodic but large deposition of new sediment during the following fall floods.*

- I appreciate that the authors do not try to temporally upscale their fluxes given the variability, but it does mean that samples from different points in time may plot very differently on Fig 11.

R: As no data are available during the fall and winter and evidence suggests that the state of the system is different during these seasons due to flood-related increase in deposition, we prefer not to extrapolate the fluxes over this period. As the aim of the paper was to discuss the link between alkalinity fluxes and sediment diagenetic processes, however, the lack of extrapolation over time does not weaken the message of our paper.

- The discussion on identification of major biogeochemical processes remains rather qualitative. I don't think it is possible with the current data to do it differently, but it is a drawback of the manuscript. Figure 10 is a nice summary of concept, but I wonder how valid it is under temporally varying conditions as observed especially at the proximal stations. I think in general more focus could be placed on the factors controlling FeS formation at the various stations and the possible impacts of flooding on these factors.

As the reviewer indicates, the identification of the major biogeochemical processes remains qualitative, though corroborative evidence between various data sets helps draw a rather clear biogeochemical picture of the processes taking place in these sediments. We discussed the temporal variability already in the response to the previous point by the reviewer. We think that the concept of Figure 8 is valid when integrating over an entire year but certainly not over each individual season: Indeed, following late fall deposition of flood sediment layers, DIC and alkalinity build up in pore waters during the winter and FeS accumulates in the solid phase as sulfate reduction and iron oxide reduction proceed. In spring and summer, sampling conducted

over 5 different years reveals that the diagenetic system is in a similar state on an pluriannual basis (see above): FeS is buried only at the end of the fall when a new flood layer is deposited closing the conceptual diagram of Figure 8. As mentioned above, the temporal variability has been included in the revised manuscript and Figure 8 has been improved to clarify these points.

Line 22:  during the spring and summer months.

Lines 64-68:  *In river dominated margins, episodic floods can deposit several cm of new sediment during a short period (days to weeks) (Cathalot et al., 2010). In these conditions, the net flux of alkalinity from the sediment depends on the net balance of alkalinity production and consumption rates in the sediment and the intensity of upward alkalinity transport*

added Line 84*: before the usual flood season in late summer.*

Added Lines 116-127

- Irrespective of the decision on the manuscript I hope that the following comments will help you further shape it.

R: We appreciate the time spent by the reviewer to carefully evaluate our work and the thoughtful comments that certainly helped us to improve our manuscript.

**Specific comments:**

L. 14-16: What about pore water iron data? Sulfate and nitrate concentrations alone don't tell you something about iron reduction. And what about manganese?

R: The reviewer is right. Low nitrate concentration does not tell about iron hydroxide reduction. We shifted to a more general statement *"In the zone close to the river mouth, pore water redox species indicated that TA and DIC were mainly produced by microbial sulfate and iron reduction"* instead of naming the oxidants. However, low nitrate concentrations suggest that denitrification is not a major process in these sediments. Furthermore, a model study comparing the proportion of different mineralization pathways concluded that denitrification is only a minor process in this area (Pastor et al., 2011). In turn, sulfate data demonstrates that dissimilatory sulfate reduction is a major respiratory pathway in these sediments (See Rassmann et al., 2016). Pore water manganese data is also presented later in the paper, however, as concentrations of dissolved Mn(II) were low and manganese reduction was not expected to contribute greatly to the biogeochemistry of these sediments we did not feel it was necessary to mention manganese in the abstract.

L. 19-21: This sentence doesn't tell me anything about the underlying mechanisms behind these concurrent observations (which you do explain in the discussion, I noticed later). If these complexes are found, does it mean that sulfide is generally limiting FeS formation?

R: We agree that this sentence is misleading and has been clarified in the abstract. The detection of organic-Fe(III) compounds, which are indicative of dissimilatory iron reduction and are rapidly reduced by sulfide, demonstrates that the system is dynamic as we would not expect to observe these Fe(III) species in zones dominated by sulfate reduction.

Rephrased Lines16-17: *"In the zone close to the river mouth, pore water redox species indicated that TA and DIC were mainly produced by microbial sulfate and iron reduction."*

- What do you mean by inorganic? Does it refer to iron oxides?

R: Yes. This has been clarified in the revised abstract.

L. 54-59: Yes, but this depends on the timescale. The net TA flux due to these processes may not correlate to what you measure as efflux. A diffusive efflux is primarily driven by the gradient at the sediment-water interface and may only reflect processes occurring deeper in the sediment on longer time scales. I would think this is especially relevant in systems with a (periodically) very high sedimentation rate (see further comments below).

R: We agree with the reviewer that the efflux is driven by the concentration gradient at the sediment-water interface (SWI), but this gradient is controlled by biogeochemical reactions in the sediment. Yet, the connection between the two can be made if chemical species can migrate over the distance between the biogeochemically active zones and the SWI over a given time. It is possible to calculate the minimal distance (d) that can be traveled by chemical species over time t by calculating the diffusion distance ($d = sqrt(2*DiffCoeff*t)$ where DiffCoeff is the diffusion coefficient of the chemical species). For a period of 6 months (between fall and spring) and with the DiffCoeff of $HCO_3^-$ ($D = 11.8 \ 10^{-6} \ cm^2/s$ at 25°C, around $6 \ 10^{-6} \ cm^2/s$ at 20°C in the sediment), the diffusion distance is around 15 cm. This distance is minimal as bioturbation and bioirrigation will increase transport and thus increase the connection distance. We can thus consider that most probably 20 cm are connected to the SWI and that biogeochemical processes over that depth interval can shape fluxes at the SWI over a 6 months period (the late spring, summer and early fall which certainly corresponds to the "longer time period" that the reviewer quote). This information will has been added to section 4.6 to clarify this point.

Added Lines 601-615

L. 62-65: Again, this is scale-dependent and only if re-oxidation of reduced constituents does not quickly occur in the water column.

R: We disagree with this comment. If reduced constituents are buried in sediments, they are not fluxing across the sediment water interface and thus not re-oxidized in the water column (see our initial response above).

L. 65-68: The objectives are formulated in a very qualitative way. I understand this for the second part, but not so much for the first part, which can be formulated more strongly.
First, the fluxes were quantified, so you may state that.

R: Agreed, this has been revised by replacing '*investigate*' by '*determine*' in the revised manuscript (line 76)

- Second, most sediments are alkalinity sources, but not all sediments release more alkalinity relative to DIC. Isn't that what you're mostly interested in, the possible excess over DIC efflux?

R: This is incorrect, sediments are generally weak sources of alkalinity because only denitrification based on external $NO_3^-$ input and FeS(or $FeS_2$) burial represent net sources of alkalinity (which is not common). Even sediments with high rates of aerobic and anaerobic respiration followed by re-oxidation of all reduced species produced during anaerobic respiration would not act as a net alkalinity sources although the large DIC quantities produced by these processes diffuse out of the sediment. Thus, measurement of multiple species was conducted to determine the extent and mechanism of alkalinity production in these sediments and its relation to diffusion across the sediment-water interface, as described in the text. This point has been

emphasized in the revised manuscript.

See new section 4.2 (Lines 454-490)

L. 91: What about temporal variability in sedimentation rate in the prodelta?

R: The temporal variability in sediment rate in the Rhône River prodelta is largely unknown, except for the paper by Cathalot et al (2010, BG) which shows deposition of an abnormal flood in June 2008. It is known that the majority of the sediments in the vicinity of the river mouth is deposited during flood events which occur mostly in fall and early winter. Thus, the sedimentation rates vary due to episodic events. This information has been provided in the revised manuscript.

Rephrased Lines 110-113): *Most of the Rhône River particles are deposited in the proximal and prodelta areas during flood events (80 % of the particles; Maillet et al., 2006; Cathalot et al., 2010; Zebracki et al., 2015), mainly in late fall and early winter, leading to the periodic accumulation of terrestrial organic-rich particles in these sediments (Radakovich et al., 1999; Roussiez et al., 2005).*

L. 106: Can you include a range of how far above the sea floor samples were approximately taken?

R: We used the depth estimation from the winch for sampling and checked the real depth with a mounted underwater depth gauge. Bottom water samples were taken within 1-2 m above the seafloor. This information has been provided in the revised manuscript.

Added Lines 133-134: *Bottom water samples were collected with 12-L Niskin$^®$ bottles from 1 – 2 m above the sea floor. The sampling depth was checked with a mounted underwater depth gauge.*

- And did you also sample overlying water from the sediment cores used for the pore water and solid phase analysis? In my experience, the composition of that water can be quite different from a Niskin bottom water sample.

R: Yes, overlying water from the sediment cores was measured and in good agreement with the bottom water DIC and TA concentrations. This information has been mentioned in the revised manuscript.

Added Lines 143-145: *To obtain concentrations as close as possible to the seafloor, overlying water from the sediment cores was sampled as well and analysed for TA and DIC concentrations.*

L. 108-110: At what temperature are the pH data presented, in situ or 25 degrees? Please add.

R: The pH samples were measured at 25°C and the pH values recalculated to in situ temperature, salinity, and pressure using CO2sys. The pH data is presented at in situ temperature and pressure. This information has been provided in the revised Method section.

L. 114: What are the 'main redox species'? Specify. In the results only DIC and TA lander data are presented.

R: This part of the sentence has been removed from the paper as the results from the in situ voltammetric sensors in the lander chamber (iron, manganese, sulfide) are not used in this manuscript.

Rephrased Lines 138-139 *Triplicate pH measurements were carried out within 1 hour after sampling at 25°C*

rephrased Line 140:*The CO2SYS software (Pierrot et al., 2006) was used to report pH on the total proton scale ($pH_T$) and at in situ temperature and salinity.*

L.119: Which redox chemical species? Specify, this is too vague.

R: This part of the sentence has been removed from the paper as the results from the in situ profiles are not used in this manuscript.

L. 124-125: Which method is used for measuring DIC and TA? Same as in section 2.6? If so, refer to it.

R: Yes, DIC and TA measurements were conducted using the same method as described later in the paper. As this section just describes the benthic chamber deployments we prefer to keep the analysis methods in section 2.6, and we refer to it in the benthic chamber section.

L.128 (Eq. 1): Did H remain constant over time or did it decrease over time as a result of the sampling? If so, have you corrected for that?

R: The volume (and thus the height) of the chamber remained constant during the deployment and the syringe volume was compensated by an equal amount of bottom water. This was accounted for in the calculations.

Added Lines 153-155: *TA and DIC samples were collected as a function of time and their concentrations corrected for the dilution that occurred by replacing the sample volume collected by ambient water.*

L. 144: How many cores were taken per station? I counted 3: 1 for porosity, 1 for voltammetry, 1 for pore water and sediment sampling. Correct?

R: Six cores were taken at each station: 1 short core (~30 cm length) for pore water extraction under $N_2$, one long core (~50 cm length) for pore water extraction in lab, one core for porosity measurement, one core for methane concentrations, one core for voltammetric profiles, and one core for archives.

Rephrased Lines 147-175:*At each sampling station, six sediment cores (2 for pore waters, 1 for porosity, 1 for voltammetry, 1 for methane, and 1 for archives)*

L. 158: Why is S0 mentioned as part of total dissolved sulfide?

R: S(0), including S(0) from polysulfides, may contribute to the total sulfide voltammetric peak (Taillefert et al., 2000) as the electrochemical reaction involves simultaneous oxidation of Hg(0) and reduction of S(0) at the electrode surface to form HgS at the same potential as that of the oxidation of Hg(0) in the presence of $H_2S$. As a result, S(0), polysulfides, $H_2S$, and HS$^-$ cannot easily be distinguished and are typically reported as total dissolved sulfide ($\sum H_2S$) when measured electrochemically. This information has been provided in the revised manuscript (line 189-190)

L. 159-160: Does this imply that you can only make a relative comparison between stations of the same cruise? Or can you compare your results with current intensities from other cruises?

R: Normalized intensities can be compared to any data sets collected at any time because they are only normalized to the sensitivity of a particular electrode at the time it was used.

L. 173-174: What was the sample volume used for alkalinity titrations?

R: The sample volume used for alkalinity titrations was 3-6 ml of pore water, according to available quantity. This information has been provided in the revised manuscript. We added as well the volumes for the other

dissolved pore water species.

Line 206: *Total alkalinity was measured on 3 - 6 ml sample volume*

L. 177-178: Did you make any replicate measurements?

R: Yes, triplicate measurements were made for DIC and either duplicate or triplicate measurements were made for TA, depending on the total pore water volume available. Replication adopted for the chemical analyses has been provided in section 2.6 of the revised manuscript.

Lines 207-208: *Depending on the available sample volume, duplicate or triplicate* [alkalinity] *titration were performed*

Line 210: [DIC concentrations were] *reported as the average and standard deviations of of triplicate measurements*

L. 199-200: How was the sediment extracted? Slicing and centrifuging? More details would be appreciated.

R: The sediment was collected using Rhizons as mentioned in the original manuscript (lines 199-200).

L. 210-212: I have to say I'm not familiar with this method, but I don't think I understand this. FeS0 consists of two pools, one measured by voltammetry (FeSaq), one not (larger nanoparticles). Spectrophotometry measures both pools. So how can the difference between both measurements be used to quantify both pools? I probably misunderstand, so could you explain it differently?

R: $FeS_0$ represents the difference between two measurements of Fe(II) by spectrophotometry and by electrochemistry. The Fe(II) measured spectrophotometrically in extracted pore waters includes both Fe(II), $FeS_{(aq)}$, and larger FeS nanoparticles which may pass through the rhizon filters used to extract pore waters but are not measured by electrochemistry due to their limited diffusion to the electrode. In turn, the Fe(II) measured electrochemically includes only Fe(II) and small $FeS_{(aq)}$ molecular clusters that are smaller than 5 nm and thus voltammetrically measurable. As a result, the difference between these two measurements represents the FeS nanoparticules (named $FeS_0$) that are small enough to pass through the rhizon filters but too large to diffuse to the electrode. We feel this information is well presented in lines 247-257 and Equation 2. Unfortunately, we cannot directly compare $FeS_{(aq)}$ and $FeS_0$ concentrations as $FeS_{(aq)}$ determined by voltammetry is not quantifiable but is reported in normalized current intensities (as described in lines 192-193).

L. 212-217: Has any particular software been used for the saturation calculations?

R: IAP can be calculated based on Eq. S1 with the parameters described in Supplementary material section S1. This calculation was done in Microsoft excel as ionic strength and activity coefficients from conventional seawater were used in these calculations. The equilibrium constant was recalculated at the ionic strength of seawater, the measured $Fe^{2+}$ concentrations were used as 'free' available $Fe^{2+}$, as $Fe^{2+}$ does not form strong complexes, and $\sum H_2S$ concentrations were used to calculate the speciation of sulfide species (assuming no elemental sulfur or polysulfide were present in the pore waters). These details have been provided in the supplemental material of the revised manuscript.

L. 243-246: Indeed this is common practice and I'm perfectly fine with it. So more out of curiosity: do your seacarb or other calculations indeed show that HCO3- constitutes >90% of TA? Using your pH and DIC data,

can you say anything about the possible presence of organic alkalinity?

R: Yes, the calculations using Seacarb show that $HCO_3^-$ concentrations in the pore water are always $> 90\%$, except for one point at 30 cm depth at station K were $HCO_3^-$ represents "only" 89 % (see new figure below).

[Figure]

To estimate the amount of organic alkalinity, we could theoretically calculate carbonate alkalinity using DIC concentrations and pH and substract the result from the measured value of total alkalinity. The difference should equal the organic part of alkalinity. In the bottom water samples, we measured pH, TA and DIC. The calculations give a concentration of organic alkalinity between 8 and 28 µmol/L ($< 1\%$ of TA) in the bottom water samples (sampled with the Niskin bottle) without any particular spatial trend.

In sediments, the issue is more complicated. pH microprofiles were measured in situ down to 20 mm only, with a vertical resolution of 200 µm. DIC and TA were measured in extracted pore water samples, with a vertical resolution of 2 cm. Furthermore, these extracted pore waters represent an average value of several mm around the rhizons. This means, that we only have an overdetermined carbonate system down to two cm sediment depth with one single value for DIC and TA and a full pH profile with around 100 values ranging from 8.1 to 7.2. We could still use an averaged pH value together with the DIC value in order to calculate the fraction of carbonate alkalinity and non-carbonate alkalinity in the power waters, but as these calculations are sensitive to pH, the approach is questionable. A first guess using an averaged pH value around 0, 1, and 2 cm depth lead to concentrations of organic alkalinity not exceeding 60 µmol/L. These low organic alkalinity concentrations in surface sediments may also prevail in deep sediments as the DIC/TA ratios are close to 1 (Figure 6), which is not the case when large quantities of organic alkalinity accumulate in sediments (Lukawaska-Matuszewska et al. 2018 Marine Chemistry, doi: 10.1016/j.marchem.2018.01.012). This is definitely an issue that could be resolved in further investigations by measuring pH profiles over longer depths in the sediment.  Because of the lack of data to fully address this question we added a short sentence in the discussion (line 491-496): "*Organic alkalinity was estimated in the bottom waters using bottom water TA, pH and DIC concentrations to be less than 1% of TA. In the pore waters, the data set did not allow estimating organic alkalinity directly, but the $r_{AD}$ close to 1 indicates that the organic alkalinity fraction is limited contrarily to previous findings where organic alkalinity plays an important role and $r_{AD}$ ratios $> 1.3$ have been recorded at similar pH (Lukawska-Matuszewska, 2016).*"

L. 247-254: This method applies if there are no other fates of $Ca^{2+}$ aside from precipitation or dissolution of $CaCO_3$. Can you add a sentence acknowledging this?

R: In sediments, calcium is essentially related to calcite (pure or magnesian). Calcium could as well be consumed by the formation of $Ca(Mg)CO_3$. The Mg fraction can get as high as 50 % (dolomite) but is generally around 20% in magnesian calcites. If this is the case, the Ca/DIC and Ca/TA ratios would be lower than for $CaCO_3$. For simplicity and having no information about the Mg content of the reprecipitating mineral phase, only $CaCO_3$ is considered. We have also to keep in mind that the correction itself affects the ratios by less than 10 %. A sentence has been added in the manuscript acknowledging that most $Ca^{2+}$ is trapped in Ca-rich carbonate.

Added Lines 295-298: "We assumed that the stoichiometric ratio of $CaCO_3$ represents a good approximation as more than 90 % of the calcium carbonates in this area are composed of calcite and less than 5 % are made of magnesian calcite (Rassmann et al., 2016)."

L. 279-281: Fluxes are highest at station Z, but also most variable (highest s.d.). More interestingly, the two sampling dates show opposing trends in the TA/DIC flux ratio (below 1 for Z, above 1 for Z, although I haven't checked the statistical significance of this), something you don't specifically discuss. Can you place this observation in the larger context of spatial and temporal variability?

R: It is difficult to account for statistical significance from just two flux calculations. Given the error bars for the two flux measurements, one cannot tell if the TA/DIC ratio is different for the two flux measurements at station Z and Z'. Spatial heterogeneity (at the scale of several meters) between the deployment Z and Z' is certainly present as shown by the significant variation of fluxes. This is due to the deposition conditions during floods which may vary locally (both in quantity and quality, see response to an earlier comment by the reviewer). Yet the large difference and the significant alkalinity fluxes observed at this station compared to the shelf site are indicative of specific processes in the proximal zone.
Sediment core incubations conducted in 2018 (figure shown above) seem to point in the direction of TA/DIC flux below 1. TA/DIC flux superior to 1 could mean, that a high fraction of TA is organic or that calcium carbonates dissolve at the surface. But as reported in this article and in Rassmann et al. (2016), bottom waters and pore waters are oversaturated with respect to calcium carbonates and calcium concentration decreases with depth. We can therefore only stress that the fluxes have been measured in situ with a benthic chamber enclosing 30x30 cm of sediment surface and should therefore be considered a robust measure of the benthic fluxes during the sampling period. These points have been highlighted in the revised discussion.

Line 149: *The chamber encloses a 30 x 30 cm sediment surface area*

added Lines 332-334: *The relatively high variability between these two measurements is probably due to high spatial heterogeneity of the sediments due to the deposition conditions during floods*

added Lines 433-439: "*The observed fluxes show some variability between stations in the proximal zone, most probably due to the high inter- (i.e., km scale between stations A and Z) and intra-station (i.e., < 100 m between Z and Z') biogeochemical heterogeneities associated with massive and rapid deposition events during floods. This heterogeneity is also visible in pore water profiles from two different cores at station Z or A (Fig. 5). Despite this subkilometer variability near the river mouth, the biogeochemical gradient from the proximal zone to the continental shelf is large enough to contrast the different zones.*"

L. 325-326: What happened at station E that it is not mentioned here?

R: Due to small variations in sulfate concentrations with depth at station E the variations with depth are of the same order of magnitude as the measuring uncertainties. As a result, sulfate to DIC or alkalinity ratios were not included in Table 3. The text has been modified to clarify this point in the revised manuscript.

Rephrased Lines 384-385: *sulfate variations with depth were limited and the uncertainty on $\Delta SO_4^{2-}$ was too*

L. 330-333: What's going on at station Z? There is clearly something different between the duplicate cores at 20-25 cm depth, as reflected in the DIC, TA and SO4 data. Can you explain this deviating pattern in the duplicate core, and do you consider them reliable? (also given the extremely high value of Ωca) From which of both cores are the CH4 data, can I rely on same symbols coming from the same core?

R: The values measured are reliable. The system is heterogenous and intra-station differences can be quite important. Yet the two profiles (short and long cores) for station A and Z match at some point at depth. Furthermore, the profiles measured at station A and Z are comparable with DIC, TA, and $SO_4^{2-}$ profiles measured in other campaigns (e.g.: Rassmann et al., 2016; Pozzato et al., 2017). Despite the different shapes of the individual profiles, the DIC/TA and DIC/$SO_4^{2-}$ ratios (and if we would look at them also the DIC/Ca ratios) are always the same. The $CH_4$ core was a different core and the symbols do not indicate that CH4 was measured on the same core. This information has been provided in the revised manuscript.

Rephrased Lines 230-234

- Maybe identifying the SMTZ would be easier if only data from a single core are used.

The concentrations of dissolved methane were measured in another core than the concentrations of DIC, TA, and $SO_4^{2-}$ for practical reasons, as it more efficient to use horizontal mini-cores at each depth in the sediment core to sample this insoluble gas (see methods). This information has been provided in the revised manuscript.

Rephrased Lines 232-236: *"Close to the Rhône River mouth, at station A, Z, and AK, one additional core was subsampled for methane analysis with 1 cm diameter corers made of cut 10-ml syringes inserted every 5 cm through pre-drilled holes on the side of the core. Due to the technical challenge of sampling non soluble methane in pore waters, this sampling could not be carried out on the same cores as the other pore water analyses."*

L. 375-377: How are these systems different or comparable from the study area? Would hat explain their lower fluxes relative to this work?

R: A first point to stress here, is that measurements of in situ TA fluxes across the sediment water interface in river dominated margins are still rare in the literature. In terms of freshwater discharge, the Rhône river (1700 $m^3$/s) transports much more water than the Guadalquivir (160 $m^3$/s), but is comparable with the Po (1500 $m^3$/s). The Danube (8200 $m^3$/s), the Fly river (6000-7000 $m^3$/s ) and the Mississippi (15,000-18,000 $m^3$/s) transport even more water and particles. (Friedl et al., 1998; Hammond et al., 1999; Aller et al., 2008; Ferron et al., 2009; Lehrter et al., 2012). The Guadalquivir, the Po and the Rhône dominated margins are microtidal, mediterranean systems and bottom waters display comparable oxygen concentrations, temperatures and salinities. In contrast, the Fly river bottom waters are warmer and less saline and the Mississippi bottom waters experience seasonal hypoxia. The fluxes we compare have all been measured in water depth between 10-150 m. The sediments are either of cohesive nature or contain sandy layers. All systems are characterized by organic rich sediments and high respiration rates and are within the same range of Alkalinity fluxes. The only exception is the Po River delta for which the stations were located in the shelf zone (rather than the prodelta) and are closer in nature and Alkalinity flux to station E from this study (1-5 mmol m$^{-2}$ d$^{-1}$).

The position of the stations in different studies cited here are partly further on the shelf and reflect the low TA fluxes as measured on the continental shelf near the Rhone River delta. A sentence was added to emphasize that point (line 450-453): *"Benthic TA fluxes obtained in the Guadalquivir estuary (24-30 mmol*

*m$^{-2}$ d$^{-1}$; Ferron et al., 2009) and the Adriatic shelf sediments off the Po River delta (0.5-10.4 mmol m$^{-2}$ d$^{-1}$; Hammond et al., 1999) are in the lower range of TA fluxes measured in the present study,* *likely because the sampling stations were located further on the shelf.*"

L. 390-391: This statement can be sharpened. Coupled nitrification-denitrification does not produce TA in a net sense, so any net TA production from denitrification must come from riverine nitrate inputs. Can you use e.g. monitoring data to make an estimate about the importance of this?

R: In microtidal systems such that Rhône River delta, haline stratification is strong, such that riverine nitrate is confined in the surface river plume (e.g., 0.5 to 1 m, Many et al., 2018 PiO) where it is either diluted or consumed by phytoplankton growth. The marine origin of the bottom water is assessed by their salinity (37.5-38.0; Table 2) and their low nitrate concentrations (Bonin et al., 2002, Wat. Res. 36, 722-732). Furthermore, nitrate profiles presented in Pastor et al. (2011) all show an increase of concentration in the porewater indicating that the nitrate flux is directed from the sediment to the water column. Therefore, riverine nitrate does probably not influence denitrification observed in the proximal zone sediment as of the bottom waters. This explanation has been provided in the revised manuscript.

Rephrased section 4.2, lines 452-488

- Also,do you have any information on nitrification rates in the sediment from earlier studies?

Unfortunately, no nitrification rates are available in the sediments for this area. Few data in the surface and bottom waters of the Rhône River mouth are available (see Bonin et al., 2002, Wat. Res. 36, 722-732). This paper has been cited in the revised manuscript (line 486).

L. 400-402: True, but as you already discuss later on, if dissimilatory iron reduction is coupled to FeS burial (or re-oxidation of Fe2+), its net efflux on alkalinity is zero. So the process definitely contributes to bulk alkalinity production, but that doesn't necessarily mean it is linked to either alkalinity effluxes or long-term net alkalinity release.

R: We agree with the reviewer that dissimilatory iron reduction contributes to alkalinity production (buildup in pore waters), but not necessarily to its efflux out of the sediment. Therefore, we called it "bulk TA production" in the original manuscript, to differentiate with the "net TA production" (the difference between bulk production and consumption).

In turn, we disagree on the first part of the comment: FeS burial and Fe$^{2+}$ reoxidation have opposite effects on TA efflux. Indeed, FeS burial generates an alkalinity efflux (this is the main point of this paper as displayed on Fig. 10, red arrows). In contrast, re-oxidation of Fe$^{2+}$ consumes alkalinity at the oxic-anoxic boundary and may cancel the diffusion of alkalinity generated during iron oxide reduction out of the sediment. We feel that these differences are well presented in the manuscript and do not require further clarification.

- If I understand the Burdige and Komada method (L. 406-409) correctly, you already assume this by linking DIC and TA production solely to sulfate consumption. It'd be good to be explicit about this and state which processes are included in this method.

R: Burdige and Komada discuss how the $r_{cs}$ (DIC/SO$_4^{2-}$ ratio) can be modified by other processes (methanogenesis, carbonate precipitation/dissolution) or by the fact, that organic matter is already partly oxidized when undergoing sulfate reduction. In this manuscript, we add another possibility for the modulation of $r_{cs}$: the interaction with the iron cycle. Indeed, as shown in Table 1, this ratio (DIC/SO4) may vary with the iron reaction pathways (from -2.25 to -1.8). This point has been added in the revised

manuscript.

Added lines 508-512: "*Assuming that sulfate reduction is responsible for the majority of the bulk alkalinity production, experimentally-derived stoichiometric ratios of the relative production of DIC and TA compared to sulfate consumption may identify the effect of other reaction pathways responsible for bulk alkalinity production or consumption in these sediments (Burdige and Komada, 2011).*"

- Also, I recently came across a paper (https://doi.org/10.1016/j.marchem.2019.03.004) that uses DIC and TA pore water profiles to quantify sulfate reduction rates. I don't know how their methods are applicable to your work but it might be interesting to include it.

R: In the cited paper, the shape of TA pore water profiles is used as a proxy for sulfate reduction rates in order to disentangle OSR (Organoclastic Sulfate Reduction) and AOM (Anaerobic Oxidation of Methane) which both consume sulfate. The authors use a reactive transport model and include the precipitation/dissolution of $CaMgCO_3$ to estimate sulfate reduction rates and contributions of the different pathways. Their approach is applied to long sediment cores where such processes develop over several meters (as opposed to our short sediment cores). In contrast to our paper, they do not have iron data, and they discuss the coupling with iron reduction and the precipitation of iron sulfide minerals on a purely theoretical level, which is different than our study. We agree it would be interesting to expand our research with a model study, but we think this should be the object of a different manuscript. This comment was not addressed further in the revised manuscript.

L. 414-417: Any reason why AOM would be less important at stations A&Z compared to B? At first sight, the SO4 and other profiles do not look too different from each other.

R: The reason for proposing that AOM is more significant at station B than at stations A and Z is based on the the low $DIC/SO_4^{2-}$ ratio observed at station B (the lowest in the whole data set). Unfortunately, we do not have any pore water $CH_4$ data at this station. Station B is still located in the main deposition area of the river plume and receives more organic matter than station AK and K. Compared to station A and Z, station B is characterized by a deeper water depth and lower accumulation rates which may favor long-term stability and therefore development of AOM. As these statements were already speculative, we did not expand this discussion in the revised manuscript.

L. 424-425: Can you place this pH of 7.2 into context? Why is a minimum of 7.2 not a 'significant lowering'? (L.423)

R: pH minima between 7.2 and 7.4 were found just below the oxygen penetration depth in the sediments of all stations. The pH minimum at station A, Z and B was the same as the minimum at stations AK, K and E, where less sulfate was consumed in the sediments. We rephrased lines 324-327: "*All pH microprofiles indicated a pH minimum between 7.2 and 7.4 just below the OPD followed by an increase to between 7.5 and 7.6 in the manganese/ferruginous layers of the sediment around 5 mm inshore and below 12 mm offshore (Fig. 2). Below this depth, the pH stabilized in the pore waters.*"

L. 426-442: I spent quite some time looking at equations 12-17 and this method. First, I'd like to see how these equations are derived (e.g. eq. 12 combines eq. 6 and 9, 10). This helps checking them and also the derivation of the ratios. Second, if you look carefully at the equations, you'd see that they are all normalized to SO4. Per mole SO4 the changes in TA and (obviously) SO4 are the same for all six reactions. So the differences in the presented ratios are solely due to the differences in DIC production. Of course this would be different if the equations were presented per mole HCO3 (ratios would be the same, but the changes in TA, DIC and SO4 would be different), but it shows that if you want to link S burial to alkalinity generation (as you do in L. 501-503), the exact pathway of iron sulfide mineral formation doesn't

matter.

R: It is incorrect to state that "the exact pathway of iron sulfide mineral formation does not matter' as, in this manuscript, we relate alkalinity generation to FeS burial and not $SO_4^{2-}$ consumption in the sediment. As can be established from equations 12 and 15, the TA/FeS production ratio can be 2/(2/3)=3 in equation 12 and 2/1=2 in equation 15. Hence, the exact pathway matters and we favor (see paper) the dissimilatory pathway for iron hydroxides with sulfate reduction and precipitation of FeS, hence a ratio of 2. No further changes were made in the text.

- Either way, when comparing measured to theoretical ratios, the method assumes that there is no other removal pathway of DIC (e.g. siderite formation, to name an option). Can this indeed be excluded?

R: The precipitation of iron and sulfide is extremely fast, likely much faster than precipitation of siderite ($FeCO_3$). Thus, we did not consider this or other pathways in our calculations. This information and the appropriate references supporting that statement has been added in the revised manuscript.

Added lines 534-536: *As the precipitation of siderite is too slow to compete with FeS precipitation (Jiang and Tosca, 2019; Pyzik and Sommer, 1981)*

L. 448-452: It'd be nice to read about the possible pathways of organic-Fe(III) complex formation earlier in the manuscript

R: We recognize that this information could have been provided in the introduction. However, to focus the introduction on the role of carbon mineralization processes on alkalinity generation and avoid increasing the length of the manuscript we chose to not present any detailed biogeochemical pathways in the introduction. This comment was not considered further in the revised manuscript.

L. 458-460: This statement is less vague than in the abstract, but it still raises questions. At what time scale do these alterations take place? Should I regard 'dominated by sulfate reduction' as the default state of the sediment, only periodically (episodically? seasonally?) replaced by 'dominated by iron oxide reduction' in periods of intense flooding and sediment deposition? Is FeS mineral formation limited by sulfide and if so, does that mean that the flooding periods overprint the default state?
R: The number of questions raised by the reviewer indicates that this sentence is misleading. Furthermore, the limited amount of information on temporal dynamics on this system regarding the redox state of the pore waters prevents a sound answer to these questions. We thus decided to remove the sentence from the abstract and to keep these thoughts for the discussion section as (line 561-569): "*Finally, as these organic-Fe(III) complexes are readily reduced by ΣH₂S (Taillefert et al., 2000), their presence in zones of sulfate reduction suggests these sediments are biogeochemically dynamic with periods of microbial iron reduction followed by sulfate reduction and rapid FeS precipitation which culminates in spring and summer. This dynamics may be temporally controlled by the input of organic and inorganic material from the Rhône River in the proximal domain during major floods in late fall and winter, which generates large DIC and TA concentrations, large iron(II) concentrations, and completely exhaust sulfate at depth in the pore waters (Pastor et al., 2018) in a reproducible manner over the years during the spring and summer.*"
Further work will be needed to elucidate these processes and their temporal succession.

L.478-481: So if I understand this correctly, it means that pIAPs are poor indicators of mineral formation, as they are highest at the site with least burial (station E). Does this mean that microenvironments play an important role in the formation of FeS? I'm also not sure if I understand what you mean to say by the argument of stronger aggregation of FeS (L. 483). If FeS is currently more aggregated, does that mean that FeS formation is not active now (given undersaturation and no FeSaq) but that it had been active in the recent past in a time when the sediments were sulfate-dominated instead of iron-dominated? (this links back to my

previous comment). Or does it simply mean that FeS formation just take place in microniches where local conditions are different?

R: Mineral formation can be determined from the pIAP assuming the system is at equilibrium. At all stations besides station E, the pore waters were either undersaturated or near saturation with respect to FeS, even though the presence of significant $FeS_0$ concentrations and removal of sulfide from pore waters indicated mineral formation. This disagreement with the calculated pIAP here indicates that the system was not at equilibrium and provides another piece of evidence for a highly dynamic system. As indicated earlier, to shorten the manuscript and not diluting the take-home message of the manuscript, the pIAP calculations has been moved to the supplementary material. Per line 483: the lack of $FeS_{(aq)}$ signals, but presence of high $FeS_0$ concentrations (which pass through the rizon filters) suggests that that iron sulfide particles were already aggregated at the time of sampling. Again this points to a dynamic zone with iron sulfide precipitation dependent on organic and inorganic (i.e., Fe(III) oxides) inputs. As suspected by the reviewer, microniches, such as leave fragments in the proximal zone, have been shown to play a role in $FeS/FeS_2$ formation (see Charles et al., 2014 and response to Reviewer 2 below). These may have changed local conditions or exarcerbated FeS formation kinetics such that the system may appear undersaturated. This part of the comment has been addressed together with pIAP calculation in the supplementary material.

L. 493-495: This depends on the fate of the other products, i.e. what happens to the produced S0. But if you assume that the S0 will also be buried (or converted to FeS, which are both more likely options than reoxidation), the alkalinity release will always be 2 moles per mole S burial.

R: It is correct to state that 2 moles of TA has been released per mole S buried, but our interest is on the FeS form which was measured as AVS. In that respect, the ratio of net alkalinity flux to buried FeS will not be 2 (but 3 see above) if S0 is buried which would be the most probable option given the high sedimentation rate and the short residence time of the sediment layer near the sediment-water interface. In our case, we favor Eq. 15 (see explanation in text) and adopted a conservative ratio of 2. We did not make changes as the text is explicit enough on our choice.

L. 501-504: First, why don't you compare the AVS burial flux with the measured alkalinity flux of station A only, instead of combining A and Z?

R: The idea of this paper was to check if the conceptual link between FeS burial and alkalinity flux was substantiated by the flux values. Therefore, we tried to come up with order of magnitudes rather than attribute numbers to a single station. Furthermore, recent sedimentation rates were only available for the overall area but not for individual stations. We therefore chose to compare average alkalinity fluxes in the proximal zone and average FeS burial in the same zone.

- Second, a point that I am just realizing: with this very high burial rate, it'll take a long time before alkalinity produced in the sediment is diffused out. You'd expect that its transport is dominated by advection, not diffusion. So that would mean an even stronger decoupling between net TA generation in the sediment and measured effluxes. Or is there bioturbation that impacts the benthic release?

As pointed out above in response to a previous comment by the reviewer, the spring-summer diagenetic processes described in this paper are the result of the late fall deposition of flood layers and their maturation. The progressive buildup of alkalinity and DIC in pore waters begins during winter and is certainly accompanied by FeS precipitation as sulfate reduction and iron oxide reduction proceed. The net alkalinity flux produced is linked to the net precipitation of FeS, which represents the difference between precipitation and re-oxidation due to bioturbation transport in the oxic zone. Burial occurs when a new flood layer is deposited (in late fall) which traps the FeS produced during the year before below a new sediment layer of 10-30 cm, ensuring its preservation. As a result, the net alkalinity flux can arise by diffusion over a depth of

15 cm as the diffusion time over this distance is 6 months (with D(HCO3)=6x10$^{-6}$ cm$^2$/s see calculations provided earlier). In the first 5-10 centimeters, transport may be increased by bio-irrigation as can be observed in A and Z DIC/TA profiles which are concave. Except for the flood period, advection is limited as sedimentation remains low. Hence, the processes that produce alkalinity (i.e., FeS precipitation) and occur in the first 20 cm of sediment can most probably be linked to the bottom water fluxes by diffusion during the late spring, summer, and early fall. These concepts have been added to the revised manuscript to clarify this point.

Added lines 598-613

L. 526-529: I agree that microniches can be important, but do your Ca2+ porewater profiles give any indication of CaCO3 dissolution at the top of the sediment?

R: In the sediment surface layer at station A and Z, Ca$^{2+}$ concentrations either decrease (St.A) or remain constant with a further decrease deeper (St.Z), thus providing no sign of carbonate dissolution in the upper layers. Furthermore, with the uncertainty of 2-3 % as reported in the method section (line 191), it is difficult to detect minor changes of Ca$^{2+}$ linked to small dissolution of carbonate. This evidence for low CaCO$_3$ dissolution from the data collected has been higlighted in the revised manuscript.

Added lines 475-477: *Minor quantities of calcium carbonate may be dissolved in microniches where the pH could be lower than 7.4 or less abundant carbonate forms (aragonite) may dissolve in the millimetric layers where this mineral is close to undersaturation.*

L. 534-539: Is FeS the dominant form of solid-phase S in the sediment, or is pyrite also present in substantial amounts?

R: Only FeS was measured in these sediments in a limited number of cores as these analyses take considerable time. Pyrite precipitation may probably explain the unaccounted TA flux from the FeS burial calculations, but given the residence time of the sediment layers in the Rhône River prodelta, pyrite precipitation most probably occurs after a few years, when the sediment is buried deep in the sediment. This discussion has been provided in the revised manuscript.

Added Lines 628-630: *Although precipitation of pyrite may also preserve the bulk alkalinity generated in the pore waters, pyrite precipitation is slow enough compared to FeS precipitation that it may occur only deeper in the sediment.*

L. 540-541: On what timescale do these processes take place? Under steady-state conditions I understand this figure, but given the highly variable sedimentation rate at especially the proximal sites, does it still apply under these dynamic conditions?

R: See comments above about the temporal variations. A couple of sentences explaining the time frame in which biogeochemical processes occur in these proximal sediments have been provided in the revised manuscript.

L. 545-547: but is Fe or S generally limiting FeS formation at the proximal sites?

R: As no or little sulfide was found in pore waters, it seems reasonable to assume that sulfide is the limiting element. This information has been provided in the revised manuscript.

Line 634: *limited by the diffusion of sulfate in the sediment;*

L. 556-559: so at station E Fe is limiting FeS formation, what about the prodelta sites?

R: At Station E, the largest fraction of OM oxidation occurs via oxic and suboxic respiratory processes, and FeS concentrations are much lower than at the prodelta sites. Based on the high deposition rates observed in the prodelta, these sites receive increased iron inputs from the Rhône and are thus less likely to be iron limited. In turn, sulfide seems to be the limiting element at these sites. These facts has been emphasized in the revised manuscript.

Line 636: (talking about the proximal domain) : *limited by the diffusion of sulfate in the sediment;*

line 654: (talking about the more offshore distal domain):  *and that this system is probably iron limited.*

L. 574-576: but if the FeS burial sink is permanent, it definitely impacts water-column TA and carbonate system dynamics on the long term, as you also indicate on L.576-578 and L.601-604.
I think this statement unnecessarily weakens the relevance of your manuscript.

R: We agree with the reviewer that " [The TA source] definitely impacts water-column TA and carbonate system dynamics on the long term", but the sentence line 574 still holds as the sediments release more DIC than TA and thus contribute to increasing $pCO_2$ of bottom waters rather than decreasing. In turn, this increase of $pCO_2$ is weaker due to the concomitant TA release compared to what it would be if only DIC is released. Hence, it is crucial to determine the TA sources from anaerobic sediments. This sentence has been modified slightly to emphasize these points.

Rephrased lines 668-674: "*As these ratios do not exceed 1, alkalinity generated in the sediments will not decrease $pCO_2$ from the bottom waters and thus not draw significant atmospheric $CO_2$ into the coastal ocean. Yet, the large benthic TA fluxes generated from deltaic sediments as a result of the periodic FeS sink in these sediments after large floods will definitely impact water-column TA and carbonate system dynamics on the long term. The elevated $F_{TA}/F_{DIC}$ ratio (>0.8), which were unknown in the Rhône River prodelta before this study, will therefore modify the carbonate cycle paradigm in these coastal regions*".

Fig. 9: What if TA data were used for this calculation? I agree though that using DIC is wiser given the possible presence of organic alkalinity.

R: For these calculations, it is not possible to use the alkalinity profile as its shape near the SWI (at the centimeter scale) is unknown and probably very different from the DIC profile, as it is likely affected by the eventual reoxidation of reduced species ($Fe^{2+}$, $Mn^{2+}$, $NH_4^+$, $HS^-$) which consume TA.  In this paper, we conclude that a major fraction of these reduced species is buried within the anaerobic sediment layers, but some may still be oxidized and thus consume TA in the first mm of the sediment. Although uncertain, the assumption of a linear DIC profile is reasonable but already questionable and represents the best option given the potential variations in the TA profiles. Given this argument, this comment will not be considered further in the revised manuscript.

Technical corrections:

Unless addressed specifically below, we agree with all the technical corrections provided below, and these comments have been incorporated in the revised manuscript.

L. 3-6: This sentence is too complex. OC respiration in sediment or water column? I'd suggest to rewrite and / or split it in two sentences.

L. 31-33: Ambiguous sentence. Does "of which about half is buried" refer to total oceanic POC or the 40% that is buried in shelf regions?

L. 56: typo in 'anaerobic'

L. 77: replace 'sediments' with 'suspended matter' or 'particles'

L. 98: "These". All sediments or those in the proximal region only?

L. 127-130 and various other sections in the manuscript (basically everywhere where equations are presented): add units to the variables you discuss here (i.e. $F_i$, H, $C_i$, etc).

L. 235-239: This sentence is too complex. Please split into two or rephrase.

L. 242: Add scale and temperature to pH.

L. 244: For which salinity are these numbers valid?

Result section: may be shortened

L. 288:change to "the absolute value of the DOU fluxes" or equivalent as they have opposing signs.

L. 306: 'station' instead of 'stations'

L. 522: seacarb is written without capitals.

Figures: Add units to the captions.

Fig. 1: Add the depth to the last (lowest) line of the bathymetry.

Fig. 2: Use different lines (e.g. solid and dotted) for O2 and pH. Printed in black & white the figure is currently very difficult to read.

Fig. 3: Could be moved to an online supplement. Add what the difference between the red and black symbols means. The error bars complicate reading of the symbols a bit, but I appreciate that they're in.

R: The red symbols were used involuntarily in this figure and has been converted to black symbols in the revised version. The figure has been moved into the supplemental material section to decrease the length of the manuscript.

Fig. 4: I'd only plot the error bars outwards, this makes the bar plot better readable and the error bar is not visible in the black bars anyway.

Fig. 5: Make it clear which measurements are from the duplicate core by using the same symbols for DIC,TA and SO4, and make them clearly different from the main core data.

Fig. 6: Add a (dashed) line at $\Omega_{ca}=1$. Also, the DIC data are poorly visible as they are mostly hidden behind the TA data. I'd leave it like this only if the point you're trying to make is that they are so similar.

Fig. 7: I'd suggest not splitting the axes into two domains, given the small jump on both axes it complicates more than that it helps reading.

Fig. 11: "as a function of water depth". Add the source of the North Sea data (Brenner et al.?). Hu & Cai (2013) is not in the list of references.

Table 1: Be consistent with sulfate and SO4 in the caption. I think that in equation (5) it should read -1/5 H+ (instead of -2/5). Show how you derived equations (12) to (17), see earlier comment.

Table 2: Add the depth interval over which mean porosity was calculated.

**Anonymous Referee #2**

- I read carefully the manuscript of Rassmann, Eitel, and collaborators and I recommend it for publication after revision. This paper presents an in-depth analysis of benthic biogeochemical processes and DIC/TA release in different stations in the Rhône River Delta area. I particularly like the multitude of measurements applied to improve understanding of processes driving anaerobic formation of TA and benthic fluxes; particularly, combining in situ incubations, potentiometric and voltammetric micro-profiles with more conventional pore water and sediment analyses. This combination of methods is rarely encountered in these types of investigations which often focus primarily on submilli metric processes at the SWI. Furthermore, the amount of data collected is significant, and has to be published, definitively.

- My main concern is related to the overall perspective of the research. The authors do not convey very clearly the scientific importance of their work. For instance, in the introduction they mention that the objective of the paper was to " investigate if sediments from deltaic regions exposed to large riverine inputs of carbon and minerals represent an alkalinity source to the bottom waters and identify the biogeochemical processes responsible for the net production of alkalinity in these sediments. ". There are several studies that have done this in coastal area as well, thus, they should portray how this work is different. And I would say that the combination of methods is unique.

- After reading the paper several time, I am still not sure to understand the take home message of this study. What is really new?

R: We appreciate the reviewer's comment that the data collected is unique and improves our understanding of the processes driving anaerobic TA formation. In turn, we are disappointed that the reviewer does not understand the novelty of this work and will modify the revised manuscript to emphasize the scientific importance of this work: that the burial of reduced iron and sulfur in the sediment prevents reoxidation of reduced metabolites at the sediment-water interface and therefore contributes to an alkalinity flux to the overlying waters. Although these concepts are not novel, to our knowledge, this manuscript provides for the first time in situ benthic alkalinity flux data and simultaneous biogeochemical evidence from pore water and sediment profiles that substantiate this argument (strong indications that FeS precipitate in the sediment column, calculation of FeS burial compared to Alkalinity generation). The abstract, discussion, and conclusions have been modified to emphasize the fact that this is the first study demonstrating benthic alkalinity flux and simultaneously providing biogeochemical evidence for the processes responsible for this alkalinity flux. Specifically we rewrote the end of the introduction (Line 76-84): *"The objectives of this study were to determine the magnitude of the alkalinity flux to the bottom waters from deltaic regions sediments exposed to large riverine inputs of carbon and minerals and identify the underlying biogeochemical processes responsible for the net production of alkalinity in these sediments. This study is one of the first to simultaneously quantify the spatial distribution of benthic TA and DIC fluxes, dissolved oxygen uptake (DOU) fluxes, burial fluxes of reduced substances, and the main biogeochemical processes involved in organic carbon mineralization in sediments. These processes were investigated along a gradient of organic carbon and mineral inputs to the sea floor in the Rhône River delta (France) before the usual flood season in late summer."* We also added a sentence at the beginning of the discussion (line 419): *"The main objectives of this study were to determine the magnitude of the alkalinity source from deltaic regions sediments exposed to large riverine inputs and to identify the biogeochemical processes responsible for the observed benthic TA and DIC net production."*

I strongly appreciated the effort to present this very large set of data that includes Fe(III)-Lorg, sulfide species, and methane in addition to major (or classical) diagenetic species. However, I often lost myself in detail that ultimately brings little. For examples, the section on the role of nitrification/ denitrification and the

section on IAPs are long but their conclusions are not very relevant for the rest of discussion. Overall the discussion should be shortened.

R: We agree with the reviewer that the IAP section is long and we have moved it to the supplemental material to shorten the manuscript as indicated in response to the same comment by Reviewer 1. In turn, we disagree that the role of nitrification/denitrification is not relevant to the discussion. Denitrification is well known to generate TA, though nitrification consumes TA as well such that only exogenous nitrate fed by the River in this case could contribute to TA production (Section 4.2). This information has been provided in the discussion but we kept it concise in the revised manuscript to satisfy this reviewer's comment.

- My second concern is on the role of terrigenous organic matter in this type of sediment. The authors characterized the study site as "deltaic sediments exposed to large riverine inputs of inorganic and organic material". In these sediments, coarse particulate organic matter is deposited during flood events and supports the establishment of sulfidic conditions and the precipitation of Fe-S phases (François Charles et al., 2014; Fagervold et al., 2014; Rassmann et al., 2016). As POM, CPOM is probably a source of DOM and organic alkalinity in pore water. What is the role of organic alkalinity on TA in these sediments? Did you calculate the theoretical TA based on DIC and pH?

R: This issue has been discussed as well by reviewer 1. We calculated theoretical TA in the bottom waters based on the DIC and pH values. The departure between theoretical and measured TA provided bottom water organic alkalinity ranging between 8-28 µmol/l. In the sediment, we only have pH in the first 2 cm on a vertical resolution of 200 µm whereas DIC and TA have been measured with a vertical resolution of 2 cm. As explained earlier in response to the same comment by reviewer 1, this difference in resolution is an issue that is difficult to overcome, and organic alkalinity increases generally in the anoxic zone deeper in the sediment, where no pH data are available. A first guess using an averaged pH value around 0, 1, and 2 cm depth leads to concentrations of organic alkalinity not exceeding 60 µmol/L. These low organic alkalinity concentrations in surface sediments may also prevail in deeper layers as the DIC/TA ratio is close to 1 (Figure 6) which is not the case when large quantities of organic alkalinity accumulate in sediments. The occurrence of large organic alkalinity is definitely an issue that could be resolved in further investigations by measuring pH profiles over longer depths in the sediment. As our dataset is not appropriate for the calculation of organic alkalinity at depth (where it occurs mostly) and the lack of organic alkalinity data does not change the main conclusions of the manuscript.

We added lines 491-495 to briefly discuss the importance of organic alkalinity in our study area. "*Organic alkalinity was estimated in the bottom waters using bottom water TA, pH and DIC concentrations to be less than 1% of TA. In the pore waters, the data set did not allow estimating organic alkalinity directly, but the $r_{AD}$ close to 1 indicates that the organic alkalinity fraction is limited contrarily to previous findings where organic alkalinity plays an important role and $r_{AD}$ ratios > 1.3 have been recorded at similar pH (Lukawska-Matuszewska, 2016).*"

- The production of organic alkalinity should be discussed and the contribution to TA should be estimated (it's rare to have enough data to do it).

R: Yes, but as already explained, we lack data to estimate the amount of organic alkalinity in the sediments.

- In addition, the accumulation of refractory organic carbon in sediments appears intimately associated with the sequestering of iron and sulfides in micro-environments (see the works of François). When the authors discuss about the aggregation of FeS, do they talk about microenvironments?

R: No, the main point of mentioning FeS aggregation (size of superior or inferior to 5nm) was to explain the differences between FeS0 and voltammetrically determined FeSaq, which points to a dynamic system.

- I think the role of terrigenous organic matter on these biogeochemical processes should be clarified.

R: Previous investigations used Delta C14 and Delta C13 isotope analyses to highlight the importance of organic material in the Rhône River prodelta (Tesi et al., 2007; Lansard et al., 2009; Cathalot et al., 2013; Pozzato et al., 2018). These studies demonstrated that terrestrial and riverine organic matter (i.e., produced by riverine primary production) constitute the major fraction of organic matter in the sediments close to the river mouth (station A and Z) and that the terrestrial fraction decreases with distance from the river mouth and is lowest at station E. As the characteristics of natural organic matter might be of interest to the reader, we briefly presented the average oxidation state, POM concentrations, and C/N ratios in section 2.1 of the revised manuscript (line 113-115): *"A large proportion of this terrestrial organic matter (>90%; Lansard et al., 2009; Cathalot et al., 2013) with occasional coarse particles (CPOM, Charles et al., 2014) is mineralized in the spring and summer."*

- My third concern is on the time scale of the explored biogeochemical processes. The deltaic sediments cannot be considered at steady state, specifically in the proximal stations. However, the discussion is based on a steady state view of the different reactions. So, what is the impact of floods on the oxygen demand and DIC/TA release? Are these fluxes constant over the year, with no seasonal variations?

R: The question of temporal and spatial scales is crucial in these environments and we tried to introduce them in the paper although we probably did not include enough material dealing with this topic in the background (1. Introduction) and field site description (2.1: The Rhone River delta) sections. These two sections have been updated with additional sentences on the temporal and spatial scales as explained in response to the same point made by Reviewer 1 (see 2[nd] comment by Reviewer 1). In brief, we added in section 2.1 (line 115-127): *"Although data are scarce, metabolites from carbon remineralization processes probably build-up progressively during winter and spring (Rassmann, unpublished data). This temporal evolution yields similar diagenetic signatures from mid-spring to end of summer, including almost complete sulfate reduction, large concentration of DIC and alkalinity (30-40 mM), 500-800 μM of dissolved iron, and no dissolved sulfide in the pore waters (Rassmann et al., 2016; Pastor et al., 2011). This pattern was observed consistently over several sampling campaigns, including April 2007 (Pastor et al., 2011), April 2013 (Dumoulin et al., 2018), May 2014 (Rassmann et al., 2016), September 2015 (this paper), and May 2018 (unpublished results). Altogether, the pore water data collected over the years in the Rhône prodelta system are consistent and indicate that biogeochemical processes in the critical proximal zone reach a reproducible state on a yearly basis due to the regularity of flood deposition in late fall and maturation of the system in spring and summer. This reproducibility of the spring-summer conditions probably also applies to benthic fluxes."*

- "[. . .] their presence in zones of sulfate reduction suggest these sediments are highly dynamic with periods of intense sulfate reduction alternating with periods during which sulfate reduction is repressed and replaced by microbial iron reduction (line 458-460). Does this sentence mean that there are two different conditions depending of the flood conditions or seasons? What are the consequences on FeS precipitation and on TA release? This sentence is too vague and raises questions on the temporal representativeness of the data (episodic event? Seasonal variations?).

R: Both reviewers were puzzled by this sentence (see specific comment by Reviewer 1 about lines 458-460), and we think the sentence is misleading. We thus decided to remove the sentence from the abstract and to keep these thoughts for the discussion section as (line 561-569): *"Finally, as these organic-Fe(III) complexes are readily reduced by ΣH₂S (Taillefert et al., 2000), their presence in zones of sulfate reduction suggests these sediments are biogeochemically dynamic with periods of microbial iron reduction followed by sulfate reduction and rapid FeS precipitation which culminates in spring and summer.  This dynamics may be temporally controlled by the input of organic and inorganic material from the Rhône River in the proximal domain during major floods in late fall and winter, which generates large DIC and TA concentrations, large iron(II) concentrations, and completely exhaust sulfate at depth in the pore waters (Pastor et al., 2018) in a reproducible manner over the years during the spring and summer."*
Further work will be needed to elucidate these processes and their temporal succession.

- I have the same questions on the spatial representativeness of the data. How do the authors explain the difference between the two replicates Z and Z'? Then, I encourage the authors to discuss about the spatio-temporal representativeness of observations.

R: The difference between both samplings can be due to spatial heterogeneity at the local scale due to differential deposition at these sites. As explained in response to the same comment by Reviewer 1 (see spatial variations in response to the 2$^{nd}$ comment by Reviewer 1), we have already described the different zones of the Rhône delta from the proximal zone to the continental shelf (lines 82-86) and their characteristics (sedimentation rates, depth, organic carbon content; see also Table 2) in the original manuscript. Something surely missing in this paragraph is an appreciation of the spatial heterogeneity at the local scale. As the main deposition of sediment occurs during floods, sediment layers are heterogeneous at the meter scale, and differences in pore water profiles can be detected at that scale (e.g. DIC and TA profiles on Fig. 6 for station A and Z taken from two different cores at the same station). These points has been highlighted in the revised manuscript. However, even with this local variability taken into consideration, the difference between the proximal zone stations and stations in the prodelta or the continental shelf are still obvious as highlighted in the discussion of the original manuscript (see sections 4.5, 4.6, 4.7, and 4.8) (see also Rassmann et al., 2016).

Lines: 331-333 (Results):  *"The relatively high variability between these two measurements is probably due to high spatial heterogeneity of the sediments due to the deposition conditions during floods.*

*Added Lines 432-438 (Discussion): The observed fluxes show some variability between stations in the proximal zone, most probably due to the high inter- (i.e., km scale between stations A and Z) and intra-station (i.e., < 100 m between Z and Z') biogeochemical heterogeneities associated with massive and rapid deposition events during floods. This heterogeneity is also visible in pore water profiles from two different cores at station Z or A (Fig. 5). Despite this subkilometer variability near the river mouth, the biogeochemical gradient from the proximal zone to the continental shelf is large enough to contrast the different zones."*

- My last issue is on the role of bioturbation. I think about bioturbation when I looked at the figure 10. According to the frequency of flood events and to the accumulation rates at the proximal stations, the diffusive transport of the anaerobically-produced alkalinity in the flood deposit to the SWI, takes time no? (see the work of Anschutz and collaborators in natural turbidites (Anschutz et al., 2002; Chaillou et al., 2006) and in experimental turbidites (Chaillou et al., 2007)).

R: As pointed out in response to similar comments by Reviewer 1 (see 2$^{nd}$ main comment and specific comment of L501-504), the spring-summer diagenetic processes described in this paper are the result of the late fall deposition of flood layers and their maturation. The fluxes of alkalinity and DIC certainly vary over

time with the progressive buildup of alkalinity and DIC in pore waters, as sulfate reduction, iron oxide reduction, and FeS precipitation proceed during the spring and summer seasons. The alkalinity flux, however, is linked to the net precipitation of FeS, which is the difference between precipitation and re-oxidation due to transport in the oxic zone. Burial occurs when a new flood layer is deposited (in late fall) which traps the FeS produced during the year below a new sediment layer of 10-30 cm ensuring its preservation. This yearly preservation of FeS ensures the concomitant benthic alkalinity flux to represent a net flux to the bottom waters which is not affected by FeS oxidation, contrarily to sediments exposed to low sedimentation rates where FeS can be entrained by bioturbation to the oxygenated layer and be re-oxidized (therefore consuming alkalinity).

As a result, the net alkalinity flux can arise by diffusion over a depth of 15 cm as the diffusion time over this distance is 6 months (with $D(HCO_3^-)=6\times10^{-6}$ $cm^2/s$). In the first 5 centimeters, transport may be increased by bio-irrigation as can be observed in the A and Z concave DIC/TA profiles. Except for the flood period, advection is limited as sedimentation remains low. Hence, the processes that produce alkalinity (i.e., FeS precipitation) and occur in the first 20 cm of sediment can most probably be linked to the bottom water fluxes by diffusion during the late spring, summer, and early fall. These concepts have been added to the revised manuscript to clarify this point. (lines 600-614)

Added Lines 600-614: "*The connection between alkalinity fluxes at the sediment-water interface and FeS burial at depth is questionable given the low residence time of the sediment near the interface (< 1yr in the first 30 cm) and the temporal variability in deposition processes (see section 2.1). Chemical gradients and thus benthic fluxes are shaped by biogeochemical reactions occurring within the diffusion length, i.e. the distance (d) that can be travelled by diffusion of chemical species over a given time:*

$$d = \sqrt{2*D*t} \qquad\qquad (6)$$

*where d is the diffusive length (cm), $D_s$ the diffusion coefficient in the sediment ($cm^2$ $s^{-1}$) and t the time (s). For a period of 6 months (between fall and spring), and using the diffusion coefficient of $HCO_3^-$ ($D_s = 7.10^{-6}$ $cm^2$ $s^{-1}$ at 20°C), the diffusion distance reaches around 15 cm. This distance represents a minimal estimate as transport is likely enhanced by bioturbation and bioirrigation such that 20 cm of sediment and pore water may be considered connected to the SWI on a semi-annual basis. These findings indicate that biogeochemical processes over that depth interval are able to shape net benthic alkalinity fluxes at the SWI over a 6-month period after the fall floods. The FeS burial effect is strengthened by the episodic but large deposition of new sediment during the following fall floods.*"

Anschutz et al. (2002) on a 4 month-old natural turbidite and Chaillou et al. (2007) on artificial turbidite showed that after a few month steady-state was not reached (10 months, Chaillou et al., 2007) but that a relative buildup of nutrients (i.e., $NH_4^+$ as an integrative diagenetic indicator) occurred. In Anschutz et al (2002), concentration of $NH_4^+$ >400µM were recorded at the bottom of the turbidite layer implying rapid diagenesis after turbidite deposition. Unfortunately, DIC was not measured in these studies, although it is likely that DIC profiles would have been similar to $NH_4^+$ profiles. In Pastor et al. (2011) in the Rhone prodelta, a peculiar flood (occurring in spring) was followed over 6 months: DIC concentrations reached 35mM at the bottom of the flood layer (30 cm) after 6 months, with a shape similar to normal-year spring and summer profiles. We did acknowledge the non steady-state issues in the new manuscript (section 2.1 and 4.7).

Added Lines 115-127: "*Although data are scarce, metabolites from carbon remineralization processes probably build-up progressively during winter and spring (Rassmann, unpublished data). This temporal*

*evolution yields similar diagenetic signatures from mid-spring to end of summer, including almost complete sulfate reduction, large concentration of DIC and alkalinity (30-40 mM), 500-800 µM of dissolved iron, and no dissolved sulfide in the pore waters (Rassmann et al., 2016; Pastor et al., 2011). This pattern was observed consistently over several sampling campaigns, including April 2007 (Pastor et al., 2011), April 2013 (Dumoulin et al., 2018), May 2014 (Rassmann et al., 2016), September 2015 (this paper), and May 2018 (unpublished results). Altogether, the pore water data collected over the years in the Rhône prodelta system are consistent and indicate that biogeochemical processes in the critical proximal zone reach a reproducible state on a yearly basis due to the regularity of flood deposition in late fall and maturation of the system in spring and summer. This reproducibility of the spring-summer conditions probably also applies to benthic fluxes."*

- Bioturbation and biodiffusion could be an efficient mechanism to transport anaerobically-produced metabolites, as TA from the anaerobic zone to the surface. Did the authors measure the bioturbation coefficients in the incubations? Did they consider the macrofauna in the studied sediment? What about the difference between total fluxes and diffusive fluxes of DO, TA and DIC? Are they similar (same magnitude)?

R: Bioturbation or bioirrigation was not measured during this cruise, although macrofauna has been measured in previous cruises (see Charles et al., 2014 or Bonifacio et al., 2014). In addition, bioirrigation is poorly quantified in the study area except the paper by Lansard et al. (2009) in which total oxygen Uptake (TOU) was compared to Diffusive oxygen uptake (DOU) in sediment incubations to estimate the effect of bioirrigation. TOU/DOU ratios of 1.2 +/- 0.4 were found over the offshore transect indicating that bio-irrigation is probably not particularly efficient in this zone. The role of bioturbation on the transport of TA and other pore water constituents can unfortunately not be determined by comparing diffusive fluxes of TA and DIC to benthic TA and DIC fluxes, as the vertical resolution of the rhizons is too low (~ 2 cm) to be able to estimate accurate gradients near the sediment-water interface. TOU fluxes were unfortunately not measured.

We added a sentence at lines 341-343: *In this area, DOU fluxes are quite representative of total oxygen uptake (TOU) by the sediments as TOU:DOU ratios are typically around 1.2 +/- 0.4 (Lansard et al., 2009).*

- The authors are kindly asked to see the attached annotated PDF with my suggestions

R: The comments of the attached annotated pdf document have been taken into account. Most of them are minor comments. Here we only respond to the comments on the content of the manuscript. Finally, we agree with the majority of the specific comments provided below, and these comments has been incorporated in the revised manuscript unless addressed specifically below.

L19: "Not sure to understand the link between organic-Fe(III) and the variability of the organic and inorganic particulate input…"
R: This point was already brought up by Reviewer 1 that some statements need context in the abstract (see response to specific comments of lines 19-21 by Reviewer 1).

L45: "produce?" implied to be better than 'create TA'

L55: reference needed
R: Several references were already provided in the sentences immediately above this one, such that we feel they do not need to be repeated here.

L66: "what is the role of the organic alkalinity produced by the terrestrial OM?  Please discuss about the role of OM source on the production of TA."

R: See the response to the 2[nd] concern of Reviewer 2 above.

L77: "? Fe-rich particles?"

R: We mean mostly iron oxyhydroxides ($FeOH_3$) . This comment has been ignored.

L95: "It 's very high. Detrital IC input from where? geology in the watershed?"

R: The carbonate content quoted here comes from numerous studies of C in sediments. There are many calcareous formations in the drainage basin of the Rhône River especially in the Southern Alps.

Line 101: *"The sedimentary inorganic carbon content ranges between 28 and 38 % (Roussiez et al., 2005) and is mostly composed of calcite (Rassmann et al., 2016) originating from the calcareous belt around the Alps."*

L99: "does it mean that you will performed diffusif and total flux calculations? what about the role of bioturbation to the exfluxes of TA and DIC?"

R: As mentioned above in response to the last main concern of the reviewer, we cannot compute the diffusive DIC and TA fluxes as the vertical resolution of the rhizons is too low (~2 cm) to determine accurate gradients near the sediment-water interface. Concerning the role of bio-irrigation on the fluxes, see remarks above and the low TOU/DOU ratio measured in the area (1.2+-0.4).

L105: "ok, see my comment on the table 2"

L114: "Which ones? please detailled"

R: This information has been provided in the revised manuscript.

L122: "Do you measure the bioturbation / bioirrigation coefficient? "

R: No, we did not measure bioturbation, as indicated earlier.

L125: "with the syringe system?"

R: Yes, with the syringe system described by Jahnke and Christiansen, 1989.

L130: "and oxygen? "

R: Unfortunately not, due to technical issues.

L141: "no TRIS?"

R: We did not use TRIS buffers, but corrected the salinity shift by using pH values of the bottom water measured via spectrophotometry (line 139). This information has been provided in the revised manuscript.

L145: "what is the porewater volume you collected? ~20mL? enough to mesaure all the parameters?"

R: Pore water volumes were between 12 and 15 ml. 3 to 5 ml were used for DIC, 3 ml for TA, 1 ml for $NH_4^+$ and 1 ml for $SO_4^{2-}$, 1-2 ml for $Ca^{2+}$, 1 ml for phosphate and dissolved iron. $CH_4$ was measured on a separate core.

L169: "I know this technique but I am surprise by the volume you collected with the pore size of 0.1um. "

R: According to the manufacturers, this is the pore size of the rhizons.

L171: "Viollier et al. for Fe2+? The ferrozine method revisited: Fe(II)/Fe(III) determination in natural waters."

R: We used Stookey (1970) as a basis for the method. The method of Viollier et al. is based on Stookey's method and only use a 10 minute reaction period with hydroxylamine which we extended to 24 hours, as initially reported in Stookey. We feel this information is not needed in the revised manuscript, as the methods section only needs to report what was conducted.

L181: "these methods seem complicated. Why not use the Rodier method ? I am just curious. Why two different methods ? what is the difference"

R: These two methods are HPLC methods which are more precise than the nephelometric method (quoted as Rodier). HPLC ensures a precision of around 1% for sulfate determination and requires 100 µl whereas the nephelometric method requires larger samples and has an uncertainty of 3-5%. As some of our stations had limited $SO_4^{2-}$ decrease and the amount of sample was small, we preferred using HPLC methods. We used two methods to establish a comparison between the methods for our own purposes. Motivation for the two HPLC methods has been provided in the revised manuscript.

Line 219: %. *"To validate a newly developed high performance liquid chromatography method (Beckler et al., 2014), pore water fractions from a separate core were also frozen at -18 °C for sulfate analysis back in the laboratory."*

L188: "I suppose it is only for the porewater Ca concentrations. not for the determination of total Ca fluxes in the chamber."

R: Yes, it is only for pore waters. Actually, we tried both, but the changes in $Ca^{2+}$ concentrations over time were too small compared to the uncertainty of the used method. This comment will not be addressed in the revised manuscript.

L191: "please add the limits  of detection"

L212: "Did you use the PHREEQC software to do the IAP calculation? please add information Same question for the omega calcite calculations "

R: As mentioned in response to a similar comment by Reviewer 1, IAP were estimated in a spreadsheet. The equilibrium constant was recalculated at the ionic strength of seawater, the measured $Fe^{2+}$ concentrations were used as 'free' available $Fe^{2+}$, as $Fe^{2+}$ does not form strong complexes, and $\sum H_2S$ concentrations were used to calculate the speciation of sulfide species (assuming no elemental sulfur or polysulfide were present in the pore waters). Finally, the activity coefficients provided in the methods section were used for these calculations. These details has been provided in the supplemental material of the revised manuscript. For the OMEGA calcite calculations, the Seacarb package for R was used.

L227: "Whatever the depth? DO you calculate de AVS burial for each depth/layer?"

R: As mentioned in these lines, we estimated an average concentration for the core and multiplied by the sedimentation rate and porosity. We feel the description was clear enough to not warrant any modifications in the revised manuscript.

L235: "this is the data repported in the table 3 (corrected from Diffusion and Ca). Ok I understand now! It is difficult to understand what do you want to do. I think you need to better introduce the goal of these calculations.
What is the avantage to do these calculations compared to use a DIC - TA diagram where you add the theoretical slopes of the different reactions and the corrected data?
Please see the last paper of Pain et al. 2018 and the figure 9 therein"

R: These calculations has been better introduced in the revised manuscript though similar calculations were already reported in Rassmann et al. (2016). We are not sure if adding a new figure would be useful, but we have referenced the paper of Pain et al. 2018 to clarify the methodology used. (Line 279)

L242: "Do Di and Dj corrected from temperature?"

R: Yes, the diffusion coefficients were calculated as a function of temperature, pressure, and salinity. This information has been provided in the revised manuscript.

L242: "Did you use a softawar (Seacarb or PHREEQC?"

R: No, these calculations were conducted in Excel using the functions reported in the literature (Li and Gregory, 1974). This information has been provided in the Supplementary material of the revised manuscript.

L246: "ok but TA is not only carbonate alkalinity
TA = [HCO3-] + 2[CO32-] + [B(OH)4-] + [OH-] + [HPO42-] + 2[PO43-] + [H3SiO4-] + 2[H2SiO42-] + [HS-] + 2[S-] + [NH3+] + [Org-] - [H+] - [H3PO4]
See my previous comment on the role of OM and organic alkalinity on TA"

R: In our case, carbonate alkalinity is the major fraction of TA. As already explained above in response to a similar comment by this reviewer, we are lacking data to estimate the non-carbonate alkalinity fraction accurately. (see line 491)

L247: "You measure TA but you talk about carbonate alkalinity in the text. Please clarify "

R: This comment is not clear, as we did not discuss carbonate alkalinity in the lines identified by the reviewer. We only describe the effect of the precipitation of calcium carbonate on TA and DIC variations in this sections.

L270: "please change the number it is figure 3."

R: The figure number is correct. Oxygen was measured with Clark type electrodes (Figure 2) and by voltammetry (Figure 5). Both methods concur. Figure 3 shows the evolution of DIC and TA concentrations in the benthic chamber over time and has been moved into the supplementary material.

L287: ""relative importance" : not clear, what does "importance" mean ?"

R: Changed

L289: "what about Mn2+? not used? please delete the data from the figure"

R: Although Mn is not discussed in the paper, we decided to leave the data on the graph as it can be useful for further studies.

L292: "mean value over the lenght of the core?"

R: Here we are actually describing maximum concentrations. These details has been added to the revised manuscript. (Line 347-348)

L343: "below the limit of detection or not measured?"

R: As mentioned in this sentence, dissolved phosphate was not measured at station B. As this sentence is clear enough, this comment will not be considered further in the revised manuscript.

L357: "so it's mainly based on a qualitative approach. Despite the lenght of discussion, there is still some questions on:
- the "steady state" approach of the redox processes under sedimentary transient conditions, (especially at the proximal stations A Z and to a lesser extent AK

R: L357 does not address the 'steady-state' approach, and this comment by the reviewer is not clear. However, the problem of comparing profiles obtained at one time period to TA fluxes integrated over several months has already been addressed in response to the 3$^{rd}$ main concern of the reviewer (see above). This comment will not be addressed further at this location in the revised manuscript.

- the role of terrestrial organic matter on benthic TA, mainly in the proximal zone

R: Similarly, we already addressed the role of terrestrial organic matter on the benthic TA flux in response to the 2$^{nd}$ main concern of Reviewer 2. We refer to our above response to address this comment.

- the role of bioturbation to explain the exfluxes of TA and DIC

R: As mentioned above in response to the last main concern of Reviewer 2, bioturbation is poorly constrained in this area, though high resolution electrochemical profiles show no apparent effect on the gradients near the sediment-water interface. In addition, the measured benthic TA and DIC fluxes are in situ fluxes over 30 x 30 cm of enclosed sediment such that bioturbation should be accounted for in these measurements. Besides the changes highlighted in response to the last main concern of the reviewer, this comment will not be addressed further here.

The sections on the formation of iron sulfide species and FeS precipitations are too long and could be shortened. I don't understand what is really new here
"
R: The section has been shortened and the novelties better highlighted to shorten the discussion.

L426: "just the distance between the two fronts ((>5cm)"

This comment is not clear and cannot be addressed easily. Sulfide concentrations are low in each sediment core below the oxygen penetration depth, suggesting not oxidation but FeS precipitation. This comment will not be considered further in the revised manuscript.

L456: "I am not sure this figure is necessary" for Fig. 7

R: We disagree as the correlation between $Fe^{2+}$ and organic-Fe(III) provides strong evidence for the microbial reduction of Fe(III) oxides as discussed in the manuscript. This information appears important to highlight the possible concomitant nature of the microbial processes in these sediments. This comment has not been considered in the revised manuscript.

L488: "so is it necessary to present the PIAP (in supplementary material?). The discussion should be shortened "
R: We agree, as acknowledged in response to comments of both reviewers. The section on pIAP was moved to the Supplementary Material.

L491: "yes, but the accumulation is not constant over the year... not at steady state after the deposition"

R: We recognize that these sedimentation events are extreme and the temporal variations of sedimentation rates has been emphasized in the revised manuscript as explained above in response to the comments of both reviewers.

L511: "For me, it is the first step of the discussion (it is also the first reaction presented in the table 1). I suggest to the authors to move this section to start the discussion."

R: This is an interesting idea by the reviewer, as we could move this section earlier in the discussion (after section 4.1), though the conclusion of this section requires to present the effect of FeS burial on the alkalinity flux for comparison. We included the ideas of this section in the new section 4.2.

L546: "Yes but I suppose this rate is not constant : massive input of  "new" and terrigeneous-rich OM material"

R: We addressed the seasonal variations associated with flood events in the fall and slow build up of alkalinity during the spring and summer seasons in response to comments made by both reviewers. The discussion has been strengthened by such discussion.

L868: "Why do you show the results in red? I would remove these incubations (Z and Z' in red)"

R: As mentioned in response to a similar comment by Reviewer 1, this was the result of an error during data plotting. The correct figure has been provided in the supplement to the revised manuscript.

L891: "below the dl?"

R: No, not measured. We feel this statement is unambiguous here.

Fig 1: "compared to the others figures, this one is "ugly" !"

R: The figure has been modified to improve its clarity.

Fig 2: "change the color between O2 and pH. At this scale,  dotted lines are not visible"

R: The figure has been modified for better visibility.

Fig 3: red triangles??

R: The figure has been reformatted and moved into the supplement section.

Table 2: "Concentrations in the bottom water? please add the information"

R: This information has been provided in the revised caption.

---

## Referee Report (RR1)

**Review of "Benthic alkalinity and DIC fluxes in the Rhone River prodelta generated by decoupled aerobic and anaerobic processes" by Rassmann, Eitel et al.**

This study uses an impressively diverse and high-quality dataset to investigate the source of strong total alkalinity fluxes originating from nearshore sediments in the Rhone River prodelta. The introduction is clear and  provides a very good summary of the diagenetic concepts required to follow this study. The methods section is detailed, even though I missed some description of how organic alkalinity was derived, or of what $CaCO_3$ minerals are actually made of. Overall, I found that the authors adequately communicate their conclusions, that the measured high alkalinity fluxes are caused by iron sulfide precipitation and burial, rather than $CaCO_3$ dissolution or denitrification, as well as the novelty of their approach. Nevertheless, I have some small concerns about the veracity of these conclusions that I will detail below. Unfortunately, this manuscript contains a number of inconsistencies that make it not straightforward to follow and understand. This work should be published upon minor revisions, which would include addressing and justifying the potential methodological flaws highlighted here.

The whole $CaCO_3$ treatment is somewhat vague and appears misleading. Statements such as "porewaters are  supersaturated with respect to calcite, suggesting that carbonate mineral dissolution is not significant" (L464-465) or "such saturation state precludes massive carbonate dissolution at the sediment surface" (L473-474) are inducing unnecessary confusion. The fact that porewaters are supersaturated with respect to calcite means that no calcite dissolution occurs, but does not mean anything regarding the other $CaCO_3$ phases, which could very well be dissolving, because more soluble. Although you have detected the presence of Mg-calcite, you never mention the possibility that it could be dissolving, and never mention a saturation state with respect to these Mg-calcite phases. Similarly, the possibility of aragonite dissolving in "microniches" is mentioned only once (L477). With the large pH decrease in the first millimeters below the SWI (L468 and Fig. 2), the presence of Mg-calcite (L297) and possibly aragonite, and given the [Ca] porewater profiles for stations B, K, E and AK, it seems unjustified to rule out $CaCO_3$ dissolution completely. Given that you apparently have data on the solid composition of the sediment, notably in terms of $CaCO_3$ phases and their Mg content, as well all the necessary porewater data necessary to calculate the saturation state profiles with respect to each of the minerals, why not using all this data to determine with certainty whether $CaCO_3$ minerals are dissolving or not?

In situ chambers may have several drawbacks, including landing disturbances, uncertainties regarding chamber mixing and hydrodynamics, leaks, etc. With porewaters being extremely rich in TA (30-40 mM, line 119 and Fig. 5) and bottom waters having a TA of ~2.5 mM, one might be worried that the landing disturbances release some of this alkalinity in the porewaters and cause the measured fluxes to be overestimated. The same applies for DIC. In fact, TA and DIC fluxes (measured with chambers) are 2 to 8 times larger than DOU rates (diffusive flux inferred from microelectrode concentration gradient), which may lead to suspicion. I see two ways this could be addressed. (1) Since you have microelectrode concentration profiles for both TA and DIC, their diffusive fluxes could be computed using Eq. (3) and compared with the chambers measured fluxes. This would be an easy way to validate chamber fluxes. (2) TA and DIC concentrations from the overlying water sampled from the sediment cores were also measured (L144-145). How do they compare with TA and DIC concentrations from the bottom waters sampled in Niskin bottles? That comparison could

help qualitatively estimating TA and DIC releases upon sediment-water interface disturbance by the landing of an instrument.

It is not always clear to me what the production ratios actually stand for, and what they are bringing to the study. Are the production ratios computed for each station as a function of sediment depth or not? I believe that it would be enlightening to plot these ratios, which are currently not shown anywhere, as a function of depth, and compare that to the porewater concentration profiles of Figures 4 and 5.

Regarding mass transfer through the sediment-water interface (SWI), and the presence of a diffusive boundary layer (DBL), there may be either a lack of documentation or assumptions that are not clearly stated. The authors never mention the presence of a DBL sitting above the seabed throughout this manuscript, which is known to control the diffusive fluxes of certain solutes through the SWI. If the authors are assuming that the DBL has a negligible influence in this system, they should bring evidence supporting this statement, or at least state this assumption explicitly. Benthic chambers alter the fluid flow above the SWI and modify the DBL shape and thickness. By stirring the water within the chamber, they may cause the DBL to be thinner than without a chamber, and enhance the diffusive fluxes. What was the stirring rate in these chambers? How can the authors quantify the effect of stirring on benthic fluxes? More discussion is required on this side. Besides, as the SWI is a plane, there can be no gradient at the interface (line 262). Given, the use of the diffusion coefficient in sediments in Eq. (3), I assume that the concentration gradient used in Eq. (3) is within porewaters only and does not extent on the water-side of the interface. If so, please state it. If, instead, the authors are referring to a concentration gradient between just above and just below the interface, then there is a diffusive boundary layer that they are not acknowledging. See the approach of Hicks et al. (2017, Biogeochemistry, 135, 35-47) who computed DOU rates using $O_2$ concentration gradients within the DBL.

**Specific comments**

**Abstract**

L11-12: Where is the "solid composition data shown in this manuscript"? If not shown, remove that from the abstract.

L13-15: Specify in which direction these fluxes are going.

L20: Make it clear that 12.5 mmol/m2/d is the burial flux of iron sulfides, with [m2] standing for the sediment surface area.

**Introduction**

L33: "account for more than 40% of POC burial in the oceans": it deserves a reference

L48: "carbonate saturation state": "carbonate" is vague. It should be "carbonate minerals", "calcite" or "aragonite"

L49: "supersaturation" instead of "oversaturation". Please correct here and elsewhere.

**Methods**

L95-98: The mentioned accumulation rates do not correspond to what is being presented in Table 2, which does not correspond to what is being presented in Table 3. Please be consistent, or explain better where the accumulation rates from Tables 2 and 3 are coming from.

L100: "total organic carbon content is higher than 2%"at which depth(s) / range of depths ?

L102: "mostly composed of calcite": A more detailed description of these composition is given in section 2.9, but I feel that it should also appear here.

L140: The CO2SYS software can provide very different in-situ pH values depending on which equilibrium constants are used, which carbonate system pair is used as an input (TA, DIC, pH), if silicate and phosphate concentrations are used or not, etc. More description on this is required. Besides, if the authors used the new version of that software described in Orr et al. (2018, Marine Chemistry 207, 84-107), which they should definitely do, as it properly propagates the uncertainties, they should reference it.

L142-143: The sentence "Dissolved oxygen concentrations … 0.5 microM" already appeared in the text a couple of lines above.

L156-157: Please provides more background on the REML approach that is used, mention any software used, and explain how were uncertainties of individual measurements taken into account.

L160: Fig. S2, which presents the concentration as a function of time in the chambers, is only referenced in the results section. It should be mentioned, and more importantly described, in this section as well, as it shows how long chambers were deployed, and the linear dependency between concentration and time.

L167: "their response to variations in oxygen concentrations is linear": it deserves a reference.

L181: This is not the same sediment density than in Table 4. Please be consistent.

L202: Is it proven that the air within the glove bag was indeed anaerobic or is it an assumption? Are there any oxygen concentration data supporting this statement?

L224: "$NH_4^+$" instead of "$NH_4^+$"

Eq. (3): This should be referred to as an "oxygen flux" or a "DOU rate". Please reword this elsewhere in the manuscript too.

Eq. (5): Are the diffusion coefficients for free-water conditions, or for sediments, corrected for tortuosity?

L298-301: I do not understand the logic behind the $CaCO_3$ reactions correction in the TA and DIC changes. If $\Delta Ca$ is positive upon $CaCO_3$ dissolution and negative upon precipitation, in order to correct for $CaCO_3$ reactions, we should subtract [2 $D_{Ca}$ $\Delta Ca$] to [$D_{TA}$ $\Delta TA$] instead of adding it, and subtract [$D_{Ca}$ $\Delta Ca$] to [$D_{DIC}$ $\Delta DIC$]. This would be much easier to understand and would correct for both $CaCO_3$ dissolution and precipitation.

L306: Instead of saying "see below", please indicate the section you are referring to.

**Results**

L400: Porewaters may be supersaturated with respect to calcite but what about Mg calcite and aragonite? A little bit of aragonite and Mg-calcite dissolution near the interface would make a big difference in terms of TA release.

**Discussion**

L491: How was organic alkalinity estimated? No detail is given on the method. The authors say in L305 that they do not have enough data to estimate it.

Eq.(6): "D" should be "$D_S$"

L630: A reference is needed to support the statement of slow pyrite precipitation.

**Figures**

Fig. 9: This figure seems to never be referenced in the text. Describe it in the text, or remove it. Besides, wouldn't it make more sense to plot the fluxes ratios as a function of the horizontal distance from the coast or river mouth, rather than water-column depth?

---

## Author Response (AR2)

Dear Prof. Dr. Jack Middelburg,

Please find attached the second revision of our paper submitted to Biogeosciences accompanied by the response. Most required modifications were made according to the reviewer's comments.

Our answers to the individual comments are given in blue and when the manuscript has been modified, the modifications are given in italic for small changes. When entire paragraphs have been rewritten, only the line numbers are indicated to avoid unnecessary length of this response letter.

All minor comments have been taken into account. When only typos etc. where spotted, no specific comment is given in the response letter.

We hope that our responses are satisfying and that our manuscript has evolved towards a publishable form for Biogeosciences.

Kind regards,
Dr. Christophe Rabouille

**Reviewer #1**
Review of "Benthic alkalinity and DIC fluxes in the Rhone River prodelta generated by decoupled aerobic and anaerobic processes" by Rassmann, Eitel et al.

Reviewer comment (RC): This study uses an impressively diverse and high-quality dataset to investigate the source of strong total alkalinity fluxes originating from nearshore sediments in the Rhone River prodelta. The introduction is clear and provides a very good summary of the diagenetic concepts required to follow this study. The methods section is detailed, even though I missed some description of how organic alkalinity was derived, or of what $CaCO_3$ minerals are actually made of. Overall, I found that the authors adequately communicate their conclusions, that the measured high alkalinity fluxes are caused by iron sulfide precipitation and burial, rather than $CaCO_3$ dissolution or denitrification, as well as the novelty of their approach. Nevertheless, I have some small concerns about the veracity of these conclusions that I will detail below.

Unfortunately, this manuscript contains a number of inconsistencies that make it not straightforward to follow and understand. This work should be published upon minor revisions, which would include addressing and justifying the potential methodological flaws highlighted here.

Author response (AR): We thank the referee for his thoughtful comments that we address within this response letter and the joined revised manuscript. We especially reworked the sections about organic alkalinity and carbonate dissolution and sincerely hope that this revision will help to get rid of these inconsistencies.

RC: The whole $CaCO_3$ treatment is somewhat vague and appears misleading. Statements such as "porewaters are over supersaturated with respect to calcite, suggesting that carbonate mineral dissolution is not significant" (L464-465) or "such saturation state precludes massive carbonate dissolution at the sediment surface" (L473-474) are inducing unnecessary confusion. The fact that porewaters are supersaturated with respect to calcite means that no calcite dissolution occurs, but does not mean anything regarding the other $CaCO_3$ phases, which could very well be dissolving, because more soluble.

AR: The reviewer is right about this remark, a supersaturation regarding calcite does not exclude the dissolution of aragonite or magnesian calcite. In the paper we remind the readers that solid phase analyses have demonstrated, that calcite accounts for more than 95 % of the $CaCO_3$ phases (Rassmann et al., 2016). We recalculated the saturation state with respect to aragonite (shown below). It shows that there is indeed a depth +/- 1 mm around the pH minimum, where aragonite is at saturation or very slightly below. This new figure replaces the ancient figure 6 in the new version of the paper in order to give a representation of other more soluble classes of carbonate minerals.

[Figure]

For magnesian calcite, estimations are more difficult to make, as until today, we do not know the Ca:Mg molar ratio of this mineral in these sediments.

Calculations made from pore water TA and DIC concentrations indicate supersaturation with respect to aragonite in proximal zone stations (A and Z) (see Figure below).

[Figure]

Similarly to the previous figure, this graph shows that the dissolution of calcium carbonates, whether calcite or aragonite, remains unlikely in the proximal zone where the strong TA fluxes were measured.

Section 4.2 has been modified to emphasize these aspects (lines 469-488) and a sentence has been added at the results section (line 406)

RC: Although you have detected the presence of Mg-calcite, you never mention the possibility that it could be dissolving, and never mention a saturation state with respect to these Mg-calcite phases. Similarly, the possibility of aragonite dissolving in "microniches" is mentioned only once (L477). With the large pH decrease in the first millimeters below the SWI (L468 and Fig. 2), the presence of Mg-calcite (L297) and possibly aragonite, and given the [Ca] porewater profiles for stations B, K, E and AK, it seems unjustified to rule out CaCO3 dissolution completely.

AR: Indeed, as shown above with the new Fig. 6, dissolution of aragonite and magnesian calcite could appear around the depth of the pH minimum, if microniches show lower saturation states that the average. Aragonite has been included into the discussion (see comment above). In turn, to estimate saturation states of magnesian calcite, the Mg:Ca molar ratio has to be determined,

something that has yet to be done. This is for sure an interesting topic for a future investigation in the prodelta stations (B, K, and AK). It is noteworthy that stations A and Z where high alkalinity fluxes are measured do not indicate undersaturation with respect to aragonite and show porewater Ca2+ decrease with depth.

RC: Given that you apparently have data on the solid composition of the sediment, notably in terms of CaCO3 phases and their Mg content, as well all the necessary porewater data necessary to calculate the saturation state profiles with respect to each of the minerals, why not using all this data to determine with certainty whether CaCO 3 minerals are dissolving or not?

AR: The only information that we have about the $CaCO_3$ phase relies on a previous cruise (Rassmann et al., 2016) and indicates that it is composed of more than 95 % of calcite and to a minor fraction of aragonite and magnesian calcite of unknown Mg:Ca ratio. As a result, the TA and DIC concentrations of the pore waters have been used to calculate the saturation states of calcite and aragonite. However, we judged useful not to overcharge Fig. 5 with $\Omega_{ar}$ profiles as aragonite is completely minor and is generally supersaturated.

RC: In situ chambers may have several drawbacks, including landing disturbances, uncertainties regarding chamber mixing and hydrodynamics, leaks, etc. With porewaters being extremely rich in TA (30-40 mM, line 119 and Fig. 5) and bottom waters having a TA of ~2.5 mM, one might be worried that the landing disturbances release some of this alkalinity in the porewaters and cause the measured fluxes to be overestimated. The same applies for DIC.

AR: We thank the reviewer for this comment. It has been long debated if benthic chamber accurately estimate benthic fluxes and several complementary approaches have been developed (see Eddy correlation techniques) that have their own drawbacks. Clearly, no technique is perfect but in situ benthic chamber (and core incubations) fluxes have been used for more than 30 years with a fairly good success. In our deployments, we rule out massive disturbance of the sediment and porewater mixing with bottom water  as the starting concentration in the chamber was 2.3 ± 5% mM for DIC and 2.6 ± 8% mM for TA. The graph below is in the Appendix (Fig. S1) of the present paper, and will be in the revised version as well. We added a sentence in the method section to acknowledge the limited impact of landing disturbance based on this argument. Line 334 *"Minimal disturbance of the sediment-water interface during deployments was evidenced by the initial DIC and TA concentrations which were within 5% and 8% respectively of the bottom water concentration (Fig. S1)."*

[Figure]

***Figure S2:*** *Temporal evolution of DIC and total alkalinity concentrations in the benthic chamber at stations A, Z (measured during two deployments), and E. Error bars represent analytical*

*uncertainties determined from triplicate measurements. The benthic fluxes and their standard deviations are provided in the text, in Figure 4 and in Table 2.*

RC: In fact, TA and DIC fluxes (measured with chambers) are 2 to 8 times larger than DOU rates (diffusive flux inferred from microelectrode concentration gradient), which may lead to suspicion.

AR: Contrarily to reviewer 1, we do not believe that the gap between DIC and DOU fluxes should lead to suspicion, because most of the reduced chemical species (Fe, H2S) produced during anaerobic mineralization and production of DIC and TA are combined together in FeS and buried. This creates a large decoupling between mineralization i.e. production of DIC and TA and the oxygen fluxes for which re-oxidation of Fe2+ and H2S plays a major role. This is indicated in our article title "Benthic alkalinity and DIC fluxes in the Rhône River prodelta generated by decoupled aerobic and anaerobic processes"). The effect of large FeS burial on the decoupling between DIC and O2 fluxes had already been postulated in Pastor et al. (2011) for the same stations.

RC: I see two ways this could be addressed.

(1) Since you have microelectrode concentration profiles for both TA and DIC, their diffusive fluxes could be computed using Eq. (3) and compared with the chambers measured fluxes. This would be an easy way to validate chamber fluxes.

AR: We thank the reviewer for this comment but unfortunately we do not have "micro-electrode" DIC and TA profiles at high resolution, but only at low resolution (see Fig. 5; first data point at 1-2 cm and spacing of 2 cm below). This is not enough for calculating a diffusive flux and for comparing it to the benthic chamber DIC and TA fluxes.

RC: (2) TA and DIC concentrations from the overlying water sampled from the sediment cores were also measured (L144-145). How do they compare with TA and DIC concentrations from the bottom waters sampled in Niskin bottles? That comparison could help qualitatively estimating TA and DIC releases upon sediment-water interface disturbance by the landing of an instrument.

AR: As mentioned above, we have directly compared TA and DIC in the chamber to TA and DIC in bottom waters (see answer above). We can rule out massive disturbance of the sediment and porewater mixing with bottom water as the starting concentration in the chamber was $2.3 \pm 5\%$ mM for DIC and $2.6 \pm 8\%$ mM for TA (see also Fig. S1)

RC: It is not always clear to me what the production ratios actually stand for, and what they are bringing to the study. Are the production ratios computed for each station as a function of sediment depth or not? I believe that it would be enlightening to plot these ratios, which are currently not shown anywhere, as a function of depth, and compare that to the porewater concentration profiles of Figures 4 and 5.

AR: We thank the reviewer for his comment on the production ratio which points towards a lack of clarity in the method section concerning the calculation of production ratios (line 282-285). Indeed, the production ratios are a classical way to investigate dominant reaction pathways by comparing the observed ratios (DIC/SO4 or DIC/TA) to theoretical ratios. The best recent example stands in Burdige and Komada (2011) for $r_{C:S}$, the DIC production to sulfate consumption ratio. In the present paper, these ratios show that the proposed set of reaction (sulfate reduction, Fe(OH)3 reduction and FeS or FeS2 precipitation) is compatible with the observed C/S, TA/S and C/TA ratios (see Table 1-theoretical and 3-observed).

The lack of clarity of our paper lies in the description of the method which led the reviewer to think that we calculated one ratio at each depth of the cores and that we can thus plot the ratio's variation with depth in each core. On the contrary, for robustness of the calculation, we calculated only one ratio per station using the entire profile of DIC, TA or SO4 represented in a concentration-concentration plot. It is thus not possible to plot the variations with depth of these ratios.

In the method section, we modified the description of the ratio calculation (line 286-291):

*"For each station, experimental stoichiometric ratios were obtained from the slope and standard deviation of the linear regression of ΔTA, ΔDIC, and ΔSO$_4^{2-}$ property-property plots. The Δs represent concentration changes with respect to bottom water concentrations and were corrected by multiplying their values by their diffusion coefficient (Δi \* Di; Berner, 1980).     The corresponding diffusion coefficients corrected for temperature and salinity (cm$^2$ s$^{-1}$) were adopted from Li and Gregory (1974)."*

RC: Regarding mass transfer through the sediment-water interface (SWI), and the presence of a diffusive boundary layer (DBL), there may be either a lack of documentation or assumptions that are not clearly stated. The authors never mention the presence of a DBL sitting above the seabed throughout this manuscript, which is known to control the diffusive fluxes of certain solutes through the SWI. If the authors are assuming that the DBL has a negligible influence in this system, they should bring evidence supporting this statement, or at least state this assumption explicitly. Benthic chambers alter the fluid flow above the SWI and modify the DBL shape and thickness. By stirring the water within the chamber, they may cause the DBL to be thinner than without a chamber, and enhance the diffusive fluxes. What was the stirring rate in these chambers? How can the authors quantify the effect of stirring on benthic fluxes? More discussion is required on this side.

AR: Indeed, the DBL has to be taken into account when dealing with processes that take place on the same scale than the DBL (Boudreau and Guinasso, 1982). From micro electrode measurements, we estimated a DBL of about 400 μm. We know that the DBL has a strong impact on mass transfer of oxygen across the sediment water interface, but we did not measure oxygen fluxes in the benthic chamber as done by Hicks et al. 2017.

Concerning the DIC and TA fluxes, the pore water profiles and mineralization extend deep in the sediment down to 10-20 cm. Therefore as stated in Boudreau and Guinasso (1982), the impact of the "resistance" caused by the DBL should be minimal on these species, as mineralization kinetics and diffusion in the sediment column dominates. Therefore, in this case, fluxes should not be influenced to a large degree by the change of DBL due to stirring rate. The stirring rate in the chamber was regulated following Jahnke & Christiansen (1989) as written in the methods section (line 153) in order to reproduce a DBL thickness comparable to the in situ situation. We extended the description of the stirring mechanism and provided the stirring rate as well as a reference that calibrated the DBL thickness as a function of the stirring rate for that particular benthic chamber (~260 μm for 10 rpm). Line 155:

"A mechanical stirrer integrated in the chamber lid was run at 10 rpm to homogenize the overlying waters in the chamber without interfering with sediment-water exchange processes (Buchholtz-Ten Brink et al., 1989)."

Concerning DBL, we did not introduce this point of discussion as we think that it is out of the scope of this paper to discuss this point and the paper is already too long.

RC: Besides, as the SWI is a plane, there can be no gradient at the interface (line 262). Given, the use of the diffusion coefficient in sediments in Eq. (3), I assume that the concentration gradient used in Eq. (3) is within porewaters only and does not extent on the water-side of the interface. If so, please state it. If, instead, the authors are referring to a concentration gradient between just above and just below the interface, then there is a diffusive boundary layer that they are not

acknowledging. See the approach of Hicks et al. (2017, Biogeochemistry, 135, 35-47) who computed DOU rates using O2 concentration gradients within the DBL.

AR: In this study, the oxygen gradient used for the calculation was taken "below" the interface, down to 400 µm depth. This information was added to the manuscript (line 268-269)

**Specific comments**

All specific comments of the reviewer have been taken into account.

**Abstract**
L11-12: Where is the "solid composition data shown in this manuscript"? If not shown, remove that from the abstract.

We removed "solid composition" from the abstract, even if nanoparticular solid FeS was measured.

L13-15: Specify in which direction these fluxes are going.

We added "into the water column"

L20: Make it clear that 12.5 mmol/m2/d is the burial flux of iron sulfides, with [m2] standing for the sediment surface area.

We added " with an estimated burial flux of"

**Introduction**
L33: "account for more than 40% of POC burial in the oceans": it deserves a reference

The reference was (Hedges and Keil, 1995; Muller-Karger et al., 2005) and has been moved to line 34-35.

L48: "carbonate saturation state": "carbonate" is vague. It should be "carbonate minerals", "calcite" or "aragonite"

We reworded to: "drives the calcite and aragonite saturation state of the pore waters towards supersaturation, and potentially triggers carbonate mineral precipitation"

L49: "supersaturation" instead of "oversaturation". Please correct here and elsewhere.

We replaced oversaturation by supersaturation in the whole manuscript

**Methods**
L95-98: The mentioned accumulation rates do not correspond to what is being presented in Table 2, which does not correspond to what is being presented in Table 3. Please be consistent, or explain better where the accumulation rates from Tables 2 and 3 are coming from.

We corrected the mistake in the manuscript for the sedimentation rates and are now consistent between the text and table 2. These rates are ranges for the corresponding domains (proximal,

prodelta and distal). In table 4, we used sedimentation rates corresponding to the selected stations estimated according to the equation and figure 8 from Lansard et al., 2009. Our calculation is a first estimation of AVS burial in the aim of seeing if the order of magnitude of FeS burial and TA fluxes match.

L100: "total organic carbon content is higher than 2%"at which depth(s) / range of depths ?

The 2 % refer to surficial sediments (0-0.4 cm). The information has been added to the manuscript (line 102)

L102: "mostly composed of calcite": A more detailed description of these composition is given in section 2.9, but I feel that it should also appear here.

The precisions have been incorporated into lines 103-106 and shortened the corresponding phrase in line 303.

L140: The CO2SYS software can provide very different in-situ pH values depending on which equilibrium constants are used, which carbonate system pair is used as an input (TA, DIC, pH), if silicate and phosphate concentrations are used or not, etc. More description on this is required. Besides, if the authors used the new version of that software described in Orr et al. (2018, Marine Chemistry 207, 84-107), which they should definitely do, as it properly propagates the uncertainties, they should reference it.

We used the equilibrium constants from Luecker et al., 2000 and added the missing information to the manuscript (lines 143-147).
Changing the equilibrium constants and varying slightly the silicate and phosphate concentrations influences the computed carbonate saturation state by a few hundredth of unit. In this paper, we do not try to obtain such accuracy and thus the precision obtained by our calculations is enough for the purpose of this paper.

L142-143: The sentence "Dissolved oxygen concentrations ... 0.5 microM" already appeared in the text a couple of lines above.
We removed this sentence from the manuscript

L156-157: Please provides more background on the REML approach that is used, mention any software used, and explain how were uncertainties of individual measurements taken into account.

We used the "lm" function in R that can make a linear approximation taking into account +/- values of individual data points. The information has been added to the manuscript.

L160: Fig. S2, which presents the concentration as a function of time in the chambers, is only referenced in the results section. It should be mentioned, and more importantly described, in this section as well, as it shows how long chambers were deployed, and the linear dependency between concentration and time.

The figure was removed from the manuscript as requested by previous reviewers. The linearity of the concentrations changes with time was already mentioned (line 332) and the duration of the deployments (10-22 hrs) was added at the same line of the revised manuscript.

L167: "their response to variations in oxygen concentrations is linear": it deserves a reference.

A reference was added.

L181: This is not the same sediment density than in Table 4. Please be consistent.

The density value has been corrected to 2.5 g cm-3

L202: Is it proven that the air within the glove bag was indeed anaerobic or is it an assumption? Are there any oxygen concentration data supporting this statement?

We monitored the oxygen concentration in the glove box with an oxygen sensor and made a test with a 2 % $((NH_4)_2 Fe(SO_4)_2 \cdot 6H_2O)$ solution that changes its colour in the presence of oxygen even in trace concentration. This information has been added to the manuscript (line 207).

L224: "$NH_4^+$ " instead of "$NH^+4$ "
This was changed (line 229)

Eq. (3): This should be referred to as an "oxygen flux" or a "DOU rate". Please reword this elsewhere in the manuscript too.
This was changed throughout the manuscript

Eq. (5): Are the diffusion coefficients for free-water conditions, or for sediments, corrected for tortuosity?

Equation 5 was removed from the paper as we think it was misleading (see above response to major comment). In our calculations, the diffusion coefficients are for sediment and are corrected for tortuosity of the sediments. As production ratio are corrected at the numerator and the denominator by tortuosity, an alternative is to use free-water diffusion coefficient.

L298-301: I do not understand the logic behind the $CaCO_3$ reactions correction in the TA and DIC changes. If $\Delta Ca$ is positive upon $CaCO_3$ dissolution and negative upon precipitation, in order to correct for $CaCO_3$ reactions, we should subtract [2 D Ca $\Delta Ca$] to [D TA $\Delta TA$] instead of adding it, and subtract [D Ca $\Delta Ca$] to [D DIC $\Delta DIC$]. This would be much easier to understand and would correct for both $CaCO_3$ dissolution and precipitation.

If one unit of $CaCO_3$ precipitates, 2 units of TA are consumed. As we want to know how much TA has been produced by reactions other than precipitation, we have to add this (absolute) precipitation value to the measured concentration as the measured concentration equals the TA produced by OM mineralization reduced by the TA consumed to precipitate $CaCO_3$. The same goes for DIC.

L306: Instead of saying "see below", please indicate the section you are referring to.

We referenced section 4.2 where organic alkalinity is discussed.

**Results**
L400: Porewaters may be supersaturated with respect to calcite but what about Mg calcite and aragonite? A little bit of aragonite and Mg-calcite dissolution near the interface would make a big difference in terms of TA release.

We calculated the pore water saturation state with respect to aragonite as explained in response to the main comment of the reviewer above. The problem with magnesian calcite is that we do not know its Ca:Mg ratio.

**Discussion**

L491: How was organic alkalinity estimated? No detail is given on the method. The authors say in L305 that they do not have enough data to estimate it.

Organic alkalinity is classically computed as the difference between observed alkalinity and the alkalinity computed using DIC and pH. We adopted this method. For the bottom waters, organic alkalinity could be estimated using DIC and pH to calculate carbonate alkalinity and compare it to the measured total alkalinity. In the pore waters, only TA and DIC concentrations are known, but not pH except at the very surface which is not enough. As a consequence, we cannot estimate organic alkalinity in the pore waters. We rephrased: "In the bottom waters, organic alkalinity was estimated from bottom water TA, pH and DIC concentrations to represent less than 1% of TA"

Eq.(6): "D" should be "D S"
This was modified in the text

L630: A reference is needed to support the statement of slow pyrite precipitation.

Two references were added, one for pyrite precipitation rates and one for FeS precipitation rates. The differences in overall second order rate constants is 12 orders of magnitude at 25°C. Line 636

**Figures**

Fig. 9: This figure seems to never be referenced in the text. Describe it in the text, or remove it. Besides, wouldn't it make more sense to plot the fluxes ratios as a function of the horizontal distance from the coast or river mouth, rather than water-column depth

Figure 9 was referenced in the text at line 667 (now line 673). We chose to show the $F_{TA}/F_{DIC}$ with respect to the water depth because it is expressed that way in Hu and Cai (2011).

---

## Author Response (AR3)

LABORATOIRE DES SCIENCES DU CLIMAT & DE L'ENVIRONNEMENT

Gif sur Yvette, 19 novembre 2019

Dear Biogeosciences Editor, Dear Jack,

Thank you for your positive answer to our paper. We have carefully checked our bibliography (5 papers were missing plus some typos) and made all the required changes (in blue in the new version of the article).

Best regards,

Christophe.

Editor's comments:
Check your reference list: it is incomplete.    Done, five references were added in addition to Pain et al., 2019
l. 209: A sample volume of 1 ml was immediately...   Done
l. 286: Pain et al. is not in reference list         Added
l. 288: The delta(subscript i) values represent..... concentrations for component i and ....Berner, 1980), where Di is the sediment diffusion coefficient for component i.        Done
l. 498: add space before TA           Done
Luecker et al not Lueker et al. (p. 35).      Done

Sincerely,

C. RABOUILLE

Unité Mixte de Recherche CEA-CNRS-UVSQ
LSCE-Orme - Bât. 701 - Orme des Merisiers - 91191 Gif-sur-Yvette Cedex      Tél. : 01 69 08 77 11 - Fax : 01 69 08 77 16
LSCE-Vallée - Bât. 12 - avenue de la Terrasse - 91198 Gif-sur-Yvette Cedex      Tél. : 01 69 82 35 23 - Fax : 01 69 82 35 68